# Resurfacing the Instance-only Dependent Label Noise Model through Loss Correction

**Mustafa Enes Aydın**[1]**, Maarten De Vos**[1 2] **& Alexander Bertrand**[1]
[1]Department of Electrical Engineering, KU Leuven, Belgium
[2]Department of Development and Regeneration, KU Leuven, Belgium
{mustafaenes.aydin,maarten.devos,alexander.bertrand}@kuleuven.be

## Abstract

We investigate the label noise problem in supervised binary classification settings and resurface the underutilized instance-*only* dependent noise model through loss correction. On the one hand, based on risk equivalence, the instance-aware loss correction scheme completes the bridge from *empirical noisy risk minimization* to *true clean risk minimization* provided the base loss is classification calibrated (e.g., cross-entropy). On the other hand, the instance-only dependent modeling of the label noise at the core of the correction enables us to estimate a single value per instance instead of a matrix. Furthermore, the estimation of the transition rates becomes a very flexible process, for which we offer several computationally efficient ways. Empirical findings over different dataset domains (image, audio, tabular) with different learners (neural networks, gradient-boosted machines) validate the promised generalization ability of the method.

## 1 Introduction

Label noise is one of the culprits for low-quality data in supervised classification settings where some of the labels deviate from their true values. While it is relatively easy to collect data, it is as laborious to label them correctly in a time-efficient manner, which inevitably results in errors in annotations. The machines are prone to overfitting these mislabels which results in poor generalization (Arpit et al., 2017; Belkin et al., 2018). The need for tackling noisy labels has thereby arisen.

Surrogate loss functions are one prominent way of handling label noise in a machine-agnostic manner: given a possibly *noise intolerant* (Manwani & Sastry, 2013) loss function, "correct" it by devising a new loss which is robust to label noise. For example, Ma et al. (2020) suggest a normalization trick to symmetrize any loss function, which is a sufficient condition for noise tolerance (Ghosh et al., 2015). A main advantage of these loss correction methods is their ease in implementation: usually the correction of the base loss takes a few lines of code with very little computational overhead, compared to, say, modifying an entire machinery.

To this end, we direct our attention to the ultimate aim in machine learning: generalization (Mohri et al., 2012). In classification with label noise, one seeks a classifier trained on the noisy data that generalizes well with respect to the clean data. *True risk* quantifies the generalization error of a hypothesis under a loss function; the mathematical formulation of the desideratum is therefore the following risk equivalence:

$$\mathbb{E}_{X,\widetilde{Y}}\big[\widetilde{\ell}(h(X),\widetilde{Y})\big] = \mathbb{E}_{X,Y}\big[\ell_{01}(h(X),Y)\big], \tag{1}$$

where $X \in \mathcal{X}$ is the instance random variable, $Y, \widetilde{Y} \in \{\pm 1\}$ are the (unknown) clean and (observed) noisy label random variables, respectively; $\mathcal{H} \ni h : \mathcal{X} \to \mathbb{R}$ is any scorer function from the hypothesis space $\mathcal{H}$; $\ell_{01}(h(\boldsymbol{x}),y) = (1 - \mathrm{sgn}(y \cdot h(\boldsymbol{x})))/2$ is the 0-1 loss function and $\widetilde{\ell} : \mathbb{R} \times \{\pm 1\} \to \mathbb{R}$ is our target loss function that is minimized during the training of $h$. Ideally, one wants $\widetilde{\ell} = \ell_{01}$, which is in fact a noise-tolerant loss function (Manwani & Sastry, 2013). Nonetheless, optimization against it is intractable in general (Ben-David et al., 2003) and the proposed direct optimizers constrain the hypothesis space to be linear functions (Sastry et al., 2010; Nguyen & Sanner, 2013). Therefore, *classification calibrated* (Bartlett et al., 2006) convex surrogate loss functions, e.g., logistic and hinge losses, are used instead. These loss functions (here denoted by $\ell$) are not necessarily

noise tolerant, however. In fact, Manwani & Sastry (2013) showed that neither the logistic loss nor the hinge loss is label-noise tolerant whether the noise is instance-dependent or not. This eliminates the possibility $\widetilde{\ell} = \ell$, and devising a noise tolerant loss function is in order: noise tolerance, combined with classification calibratedness, form a path from *noisy empirical risk w.r.t.* $\widetilde{\ell}$ to *clean true risk w.r.t.* $\ell_{01}$.

Risk equivalence in equation 1 involves the transition probabilities to make a connection between $Y$ and $\widetilde{Y}$. This transition is an unknown process in general whose modeling roughly falls into four categories (Menon et al., 2018): RCN (random classification noise), CCN (class conditional noise), IDN (instance dependent noise) and ILDN (instance and label dependent noise). They are essentially distinguished by what they put on the right side of the conditional probability $\mathbb{P}(\widetilde{Y} \neq Y \mid \cdot)$: RCN leaves it empty; CCN puts $Y$; IDN puts $X$; and ILDN puts $X, Y$. RCN is rather unrealistic as it assumes a uniform rate of transition independent of $X$ and $Y$. CCN is the most widely studied noise model that assumes $\widetilde{Y} \perp\!\!\!\perp X \mid Y$, i.e., the label noise depends *only* on the class but not on the specific instance. The more natural (Chen et al., 2021) and powerful yet challenging assumption involves a conditioning on $X$ by assuming IDN or ILDN, which continues to attract attention in recent research (Bae et al., 2024; Li et al., 2025).

The distinction between IDN and ILDN is worth emphasizing. When "instance dependent noise" is used, what is usually meant is instance *and* label dependent noise, i.e., ILDN and not IDN. IDN is driven by the plausible assumption that $Y$, being an aggregate statistic of $X$, does not convey any more information than $X$ on the transition rate. That IDN involves less parameters might imply underfitting in modeling the noise; however, as empirically shown, it is not less powerful than ILDN while being much more computationally efficient. Interestingly, there are only a few works that model the noise in this instance-only dependent form (Bylander, 1998; Du & Cai, 2015), which both focused on linear machines. We generalize this underutilized noise model to be employable with any learning machine and provide practical ways to estimate its value per instance. This model complements the risk equivalence starting point to devise a noise tolerant surrogate loss function.

We summarize our main contributions as follows.

- We introduce an instance-aware loss correction mechanism based on risk equivalence to fortify a given loss function to combat instance-dependent label noise effectively (theory-wise) and efficiently (computation-wise).

- We resurface the underutilized instance-*only* dependent noise model to estimate the transition probability of each instance to complement the surrogate loss function design. We also provide computationally efficient ways to model the transition rates.

- Empirically, we validate our approach with moderate and high noise rates over four datasets of different domains with neural networks and tree-based models, showing comparable/better results against the current approaches while being dataset- and machine-agnostic.

## 2 RELATED WORK

**Noise models.** Angluin & Laird (1988) formalized the label noise problem and showed learnability under RCN where the transition rate is the same for all $x, y$ pairs. The more realistic CCN model appoints a distinct rate for each possible transition among classes, which is the most widely studied noise model: Liu & Tao (2016) employed data reweighting; Zhang & Sabuncu (2018), Patrini et al. (2017) and Wang et al. (2019) focused on loss correction; and Han et al. (2018), Yu et al. (2019) and Li et al. (2020) introduced effective neural network machineries based on the difference of loss suffered among samples. The more natural noise model where labeling errors depend on the instance itself has also been heavily studied: Xia et al. (2020) and Yang et al. (2022) learn the transition matrix for ILDN via three stage training where the former learns partial matrices to be combined and the latter uses distillation to acquire Bayes-optimal labels; Zhang et al. (2021) and Cheng et al. (2021) progressively correct suspicious data pairs; Bae et al. (2024) perform Dirichlet-based resampling using the transition matrix; Du & Cai (2015) reinterpret linear classifiers from a distance-to-boundary perspective under IDN, which Bylander (1998) also studied under monotonic noise for perceptron learning; and Yao et al. (2021) and Li et al. (2025) took a causal graph viewpoint in ILDN modeling.

**Loss correction.** Many machine-agnostic loss correction methods have been proposed to deal with label noise. Natarajan et al. (2013) proposed a CCN-based surrogate loss. Zhang & Sabuncu (2018) and Wang et al. (2019) specifically modified the cross-entropy loss for robust learning based on CCN; the former used a Box-Cox-like transformation and the latter introduced an additive term based on the (reverse) KL divergence. Goldberger & Ben-Reuven (2017) introduced a noise-adaptation layer to a given neural network to make cross entropy more tolerant to label noise. Ma et al. (2020) proposed a normalization trick to symmetrize any loss function for noise tolerance and further improved it by weighted combinations of losses, which was generalized by Zhou et al. (2021) as asymmetric loss functions. Xu et al. (2019) introduced an information theory-centric loss involving the determinant of the transition matrix of a CCN noise model. Rooyen et al. (2015) proposed a modified hinged loss that is provably noise tolerant to combat RCN. Patrini et al. (2017) introduced two ways to correct a given loss function via CCN-based transition rates. Liu & Guo (2020) introduced peer losses without transition rates directly involved, which achieves risk equivalence up to a constant involving the transition rates.

Perhaps the closest studies to our work are Natarajan et al. (2013), Patrini et al. (2017); and Bylander (1998), Du & Cai (2015). The critical difference from Natarajan et al. (2013) is their lack of expectation over the latent label variable $Y$, i.e., we would argue that it does not properly achieve risk equivalence (in fact, the expectation is present in Patrini et al. (2017)). Furthermore, it is a CCN-based model, i.e., the label noise is not instance-dependent, and the transition rates are assumed known or otherwise estimated via validation which is not scalable to IDN. Patrini et al. (2017) is also based on CCN and the estimation of the transition rates assumes the existence of anchor points, i.e., points whose annotations are almost surely correct, and involves a separate training stage, which incurs computational load and disconnection in learning. As for the latter IDN-based works of Bylander (1998) and Du & Cai (2015), they are focused on linear machines (former perceptron; latter logistic/probit regressors) with one way of estimating the noise (distance-based); we, however, not only generalize the learning to nonlinear machines with a generic loss correction mechanism but also formalize and propose several computationally efficient ways to estimate the noise rates.

## 3 METHODOLOGY

In this section, we first present the notation used, then introduce the instance-aware loss correction and lastly propose estimation approaches for the instance-only dependent noise.

### 3.1 PRELIMINARIES

We consider the supervised binary classification problem. $\mathcal{X}$ denotes the feature space, $\mathcal{Y} = \widetilde{\mathcal{Y}} = \{\pm 1\}$ the label space (clean and noisy, respectively). $X, Y, \widetilde{Y}$ denote the random variables and $\boldsymbol{x}, y, \widetilde{y}$ their specific samples associated with $\mathcal{X}$, $\mathcal{Y}$ and $\widetilde{\mathcal{Y}}$, respectively. $\mathcal{H} \ni h : \mathcal{X} \to \mathbb{R}$ is any scorer function from the hypothesis space $\mathcal{H}$; 0-1 loss is given by $\ell_{01}(h(\boldsymbol{x}), y) = (1 - \operatorname{sgn}(y \cdot h(\boldsymbol{x})))/2$. Given a loss function $\ell : \mathbb{R} \times \mathcal{Y} \to \mathbb{R}$, $R_\ell(h) = \mathbb{E}_{X,Y}\big[\ell(h(X), Y)\big]$ is the true clean risk; $\widetilde{R}_\ell(h) = \mathbb{E}_{X,\widetilde{Y}}\big[\ell(h(X), \widetilde{Y})\big]$ is the true noisy risk; and $\widehat{\widetilde{R}}_\ell(h) = \frac{1}{|S|}\sum_{i=1}^{|S|}\ell(h(\boldsymbol{x}_i), \widetilde{y}_i)$ is the empirical noisy risk, where $S = \{(\boldsymbol{x}_i, \widetilde{y}_i)\}$ is a realization of the noisy data pairs of cardinality $|S|$. When the subscript $\ell$ on the risk is omitted, it implies $\ell_{01}$. $\rho_{\boldsymbol{x}} = \mathbb{P}(Y \neq \widetilde{Y} \mid X = \boldsymbol{x})$ denotes the label flip probability of a given instance $\boldsymbol{x}$. We assume $\mathbb{E}_X[\rho_{\boldsymbol{x}}] < 0.5$ for learning to take place [1].

### 3.2 LOSS CORRECTION VIA RISK EQUIVALENCE

We aim to form a path from the *empirical noisy risk minimization* w.r.t. a surrogate loss $\widetilde{\ell}$ to the *true clean risk minimization* w.r.t. $\ell_{01}$. Since $\ell_{01}$ is hard to optimize, a classification calibrated loss function is used instead, e.g., logistic loss: $\ell_{\log}(h(\boldsymbol{x}), y) = \log(1 + e^{-y \cdot h(\boldsymbol{x})})$; it is well-known that risk minimization w.r.t. the logistic loss parallels the minimization of the true risk w.r.t. $\ell_{01}$, i.e., $R(h) - R^*(h) \leq \sqrt{2(R_{\ell_{\log}}(h) - R^*_{\ell_{\log}}(h))}$, where $R^*$ denotes the minimum possible risk, i.e.,

---

[1]The machine would be indifferent to this assumption being invalidated, which amounts to the semantic flip of the positive and negative labels; its predictions on unseen data would simply need to be inverted.

the Bayes-optimal one. As the true distribution of $\boldsymbol{x}$'s are never known, empirical/structural risk minimization is instead used as a proxy backed up by the law of large numbers. Once the label noise comes into the picture, however, the connection between the $\widehat{\widetilde{R}}_{\ell_{\log}}(h)$ and $R(h)$ is lost: logistic loss is not noise tolerant even when the noise rate is uniform, i.e., $\rho_{\boldsymbol{x}} = \rho \, \forall \boldsymbol{x}$ (Ghosh et al., 2015). To this end, we propose to devise an $\widetilde{\ell}$ based on $\ell$, which can be any classification calibrated loss function (e.g., logistic, hinge), such that the path is complete:

$$R_{\ell_{01}}(h) \underset{\substack{\text{Classification} \\ \text{Calibratedness}}}{\longleftarrow} R_\ell(h) \underset{\text{Our } \widetilde{\ell} \text{ design}}{\longleftarrow} \widetilde{R}_{\widetilde{\ell}}(h) \underset{\substack{\text{Law of} \\ \text{Large Numbers}}}{\longleftarrow} \widehat{\widetilde{R}}_{\widetilde{\ell}}(h). \qquad (2)$$

We now propose our loss correction scheme. (All proofs are in Appendix A.1.)

**Proposition 1.** *Let $\ell$ be any loss function. Define*

$$\widetilde{\ell}(h(\boldsymbol{x}), \widetilde{y}) = \frac{\mathbb{P}(Y = \widetilde{y} \mid \boldsymbol{x})(\mathbb{P}(\widetilde{Y} = -\widetilde{y} \mid \boldsymbol{x}) - \rho_{\boldsymbol{x}})\ell(h(\boldsymbol{x}), \widetilde{y}) - \mathbb{P}(Y = -\widetilde{y} \mid \boldsymbol{x})\rho_{\boldsymbol{x}}\ell(h(\boldsymbol{x}), -\widetilde{y})}{\mathbb{P}(\widetilde{Y} = \widetilde{y} \mid \boldsymbol{x})\mathbb{P}(\widetilde{Y} = -\widetilde{y} \mid \boldsymbol{x}) - \rho_{\boldsymbol{x}}}.$$
$$(3)$$

*Then we have $\widetilde{R}_{\widetilde{\ell}}(h) = R_\ell(h)$, i.e., $\mathbb{E}_{X,\widetilde{Y}}\big[\widetilde{\ell}(h(X), \widetilde{Y})\big] = \mathbb{E}_{X,Y}\big[\ell(h(X), Y)\big]$.*

(The proof of Proposition 1 in Appendix A.1 also explains the rationale behind this choice for a modified loss function.)

In equation 3, there are three unknown probabilities: $\mathbb{P}(Y \mid \boldsymbol{x})$, $\mathbb{P}(\widetilde{Y} \mid \boldsymbol{x})$ and $\rho_{\boldsymbol{x}}$. We now explain how we estimate them in practice.

### 3.2.1 MODELING $\mathbb{P}(Y \mid \boldsymbol{x})$ AND $\mathbb{P}(\widetilde{Y} \mid \boldsymbol{x})$

$\mathbb{P}(\widetilde{Y} \mid \boldsymbol{x})$ can be estimated directly from the data using any learning machine as we have direct access to $(\boldsymbol{x}, \widetilde{y})$ pairs. This is not the case for $\mathbb{P}(Y \mid \boldsymbol{x})$ as we have no access to the clean labels. However, since the scorer $h$ also models $\mathbb{P}(Y \mid \boldsymbol{x})$, we can set $\mathbb{P}(Y \mid \boldsymbol{x})$ in equation 3 to the standard logistic over the scorer, i.e., $\mathbb{P}(Y = z \mid \boldsymbol{x}) \approx \sigma(z \cdot h(\boldsymbol{x})) = (1 + e^{-z \cdot h(\boldsymbol{x})})^{-1}$. The rationale is that, ideally, when the training with $\widetilde{\ell}$ finishes, the machine has converged to learn $\mathbb{P}(Y \mid \boldsymbol{x})$, i.e., the clean label probability for generalization. Since clean data pairs (despite being unknown) dominate the learning in the initial stage (Han et al., 2025), we spare a few epochs at the beginning (warm-up) where we only use the (unmodified) base loss $\ell$ to form a strong baseline for $\mathbb{P}(Y \mid \boldsymbol{x})$. As for $\mathbb{P}(\widetilde{Y} \mid \boldsymbol{x})$, one might train a normal model for $\widetilde{Y}$, and then use its predictions inside the modified loss to train $h$ (and hence implicitly also $\mathbb{P}(Y \mid \boldsymbol{x})$). This disjoint training for $\mathbb{P}(\widetilde{Y} \mid \boldsymbol{x})$ and $\mathbb{P}(Y \mid \boldsymbol{x})\,/\,h$, however, is not only time consuming but turns out to also lack in performance (as empirically shown in Appendix A.4). Therefore, in practice, we use $\sigma(z \cdot h(\boldsymbol{x}))$ for $\mathbb{P}(\widetilde{Y} \mid \boldsymbol{x})$ also; as the machine is trained with the noisy labels to mimic the annotator's brain, it is expected to model $\mathbb{P}(\widetilde{Y} \mid \boldsymbol{x})$ in the process (including and after warm-up). We provide a detailed argument for this modeling choice in Appendix A.4.1, and an analysis on the effect of the warm-up period in Appendix A.5.

With these estimates in place, the loss in equation 3 becomes

$$\widetilde{\ell}(h(\boldsymbol{x}), \widetilde{y}) = \frac{\sigma(\widetilde{y} \cdot h(\boldsymbol{x}))(\sigma(-\widetilde{y} \cdot h(\boldsymbol{x})) - \rho_{\boldsymbol{x}})\ell(h(\boldsymbol{x}), \widetilde{y}) - \sigma(-\widetilde{y} \cdot h(\boldsymbol{x}))\rho_{\boldsymbol{x}}\ell(h(\boldsymbol{x}), -\widetilde{y})}{\sigma(\widetilde{y} \cdot h(\boldsymbol{x}))\sigma(-\widetilde{y} \cdot h(\boldsymbol{x})) - \rho_{\boldsymbol{x}}} \qquad (4)$$

and we move to the modeling of the last unknown, $\rho_{\boldsymbol{x}}$.

### 3.2.2 MODELING $\rho_{\boldsymbol{x}}$

The transition rate $\rho_{\boldsymbol{x}} = \mathbb{P}(Y \neq \widetilde{Y} \mid X = \boldsymbol{x})$ is only conditioned on $X$ and not on $Y$. Therefore, per instance, we do not estimate a transition matrix but instead a single value. Furthermore, as $\rho_{\boldsymbol{x}}$ does not depend on the true label, there is nothing preventing the estimation of $\rho_{\boldsymbol{x}}$'s in an unsupervised manner, giving flexibility. Before providing ways to estimate $\rho_{\boldsymbol{x}}$, we formalize it over the notion of *difficulty* – the more difficult an instance $\boldsymbol{x}$ is, the higher the label flip probability $\rho_{\boldsymbol{x}}$.

**Definition 1** ($\rho_{\boldsymbol{x}}$). *Let $z : \mathcal{X} \to \mathbb{R}$ be a "difficulty" mapping such that the more difficult (to label) $\boldsymbol{x}$, the higher label flip probability $\rho_{\boldsymbol{x}} := \mathbb{P}(Y \neq \widetilde{Y} \mid X = \boldsymbol{x})$, i.e.,*

$$\rho_{\boldsymbol{x}} = \phi(z(\boldsymbol{x})),$$

*where $\phi : \mathbb{R} \to [0, 1]$ is a monotonically increasing function (not necessarily differentiable), which, together with z, also satisfies $\mathbb{E}_X[\phi(z(\boldsymbol{x}))] < 0.5$ so that there is more signal than noise for learning to take place.*

An obvious choice would be to use the model $h$ itself within the difficulty map $z(.)$, as the model has some notion of which labels are harder to classify than others. As such, $z(.)$ can be learned online during training of $h$ (see below). In this case, we impose that $z(.)$ is

- increasing when $h(\boldsymbol{x}) < 0$ and decreasing when $h(\boldsymbol{x}) \geq 0$, and
- differentiable and Lipschitz[2] such that $-\infty < \inf_{\boldsymbol{x} \in \mathcal{X}} \frac{\partial \rho_{\boldsymbol{x}}}{\partial h(\boldsymbol{x})} < 0 < \sup_{\boldsymbol{x} \in \mathcal{X}} \frac{\partial \rho_{\boldsymbol{x}}}{\partial h(\boldsymbol{x})} < \infty.$

The definition necessitates the design of two functions: $z$ the difficulty metric and $\phi$ that turns that difficulty into a probability. We now provide several ways to model them (a nonexhaustive list).

---

**How to find the difficulty of an instance, i.e., $z(\boldsymbol{x})$?**

1. Offline (unsupervised)
   - (a) Clustering, e.g., K-means (Lloyd, 1982): $z(\boldsymbol{x})$ is the (inverted) distance of $\boldsymbol{x}$ to its cluster center.
   - (b) Representation learning, e.g., sparse autoencoder (Lee et al., 2007): $z(\boldsymbol{x})$ is the reconstruction error. The learned representations are then used for the downstream classification task.
2. Online (while training $h$)
   - (a) Distance to decision boundary: the further away the instance from the decision boundary formed by $h(\boldsymbol{x})$, the less error prone it is for label flips.
   - (b) Proximity to uniform distribution of number of votes in an ensemble: the closer the vote distribution to 50/50, the higher the probability of label flip. KL divergence can be used to compute the desired proximity (inverted), for example.

**How to turn difficulties into probabilities, i.e., $\phi$?**

1. $\beta$-logistic function: $(1 + e^{-\beta z})^{-1}$ with $\beta > 0$.

2. Distribution functions: exponential PDF: $\frac{1}{\beta} e^{-z/\beta}$ with $\beta \geq 1$ (when the range of $z(\cdot)$ is $\mathbb{R}_+$); Gaussian CDF: $\frac{1}{2} + \frac{1}{2}\text{erf}\left(\frac{z-\mu}{\sigma\sqrt{2}}\right)$ with $\mu \in \mathbb{R}, \sigma > 0$.

---

We found empirically that *distance to decision boundary* for difficulty modeling and *$\beta$-logistic* function for turning them into probabilities work well (these are also used in the experiments in Section 4). Finding the distance to the decision boundary is straightforward if $h$ is linear, i.e., $h(\boldsymbol{x}) = \boldsymbol{w}^T \boldsymbol{x}$ (it is $|h(\boldsymbol{x})|/\|\boldsymbol{w}\|_2$). For nonlinear models, however, this no longer applies. One can perform clever random perturbations on the sample until the decision under $h$ changes, e.g., with DeepFool (Moosavi-Dezfooli et al., 2016) to approximate this distance. However, these approaches are time consuming given the distance needs calculating every iteration. Instead, we observe that if the network was a perfect classifier, then the embedded data in its last layer must be linearly separable. In fact, Li et al. (2019) showed that the last layer is solving an SVM problem in such a network. Building on this observation, we use $|h(\boldsymbol{x})|$ as a proxy for the distance of $\boldsymbol{x}$ to the decision boundary of $h$, whether $h$ is linear or not (note that this choice satisfies the conditions imposed on $\rho_{\boldsymbol{x}}$ earlier). As $h(\boldsymbol{x})$ already gets computed while learning, this approach is computationally highly efficient (a sample implementation in code is in Appendix A.10[3]). We also observed that this proxy also works well for nonlinear machines that are not neural networks, e.g., LightGBM (Ke et al., 2017), as shown in Section 4.3.

On the theoretical side, we provide a mathematical ground for this design choice, due to Menon et al. (2018). They show that a "boundary-consistent noise" model, of which our distance-to-decision-boundary design is a special case, is not only consistent between noisy and clean domains for AU-ROC maximization (Proposition 1 in Menon et al. (2018)) but also lends itself to an explicit excess AUROC risk bound to quantify the said consistency (Theorem 2 in Menon et al. (2018)). Here, we restate the theorem by adapting to our notation for convenience:

---

[2]The reason for this requirement will be clear in Section 3.2.4 (Lemma 1 in particular).
[3]Code is also available at https://github.com/mustafaaydn/NDX.

**Theorem 1** (Menon et al. (2018), adapted). *Given $\rho_x = \phi(d_h(x))$ where $\phi\colon \mathbb{R}_0^+ \to [0,1]$ is a monotonically decreasing function and $d_h(x)$ is the distance of $x$ to the decision boundary of $h$, suppose that $\rho_{\max} := \max_{x \in \mathcal{X}} \rho_{\boldsymbol{x}} < \frac{1}{2}$. Then, for any scorer $h$,*

$$R_{\mathrm{rank}}(h) - R_{\mathrm{rank}}^* \leq \frac{\widetilde{\pi} \cdot (1 - \widetilde{\pi})}{\pi \cdot (1 - \pi)} \cdot \frac{1}{1 - 2 \cdot \rho_{\max}} \cdot (\widetilde{R}_{\mathrm{rank}}(h) - \widetilde{R}_{\mathrm{rank}}^*),$$

*where $\widetilde{\pi} = P(\widetilde{Y} = +1), \pi = P(Y = +1)$; $R_{\mathrm{rank}}(h)$ denotes the true clean ranking risk of $h$, i.e., $\mathbb{E}_{X|Y=+1, X'|Y=-1}[\ell_{01}(h(X) - h(X'), 1)]$ and $\widetilde{R}_{\mathrm{rank}}(h)$ the true noisy ranking risk of $h$, defined similarly, and starred risks represent the Bayes optimal ones in the respective domains.*

Theorem 1 lays down a theoretical ground for this particular choice of $\rho_{\boldsymbol{x}}$ along with its empirical support. We note that the assumption $\rho_{\max} < \frac{1}{2}$ is satisfiable trivially via, e.g., halving the output of the sigmoid or using an exponential PDF with a rate $< 1/2$.

### 3.2.3 Insights into the Modified Loss Function

With our models for the unknown probabilities in place, we can analyze several limit cases, which give some insight in the behavior of the modified loss function:

- $\rho_{\boldsymbol{x}} \to 0$, i.e., label flip seems highly unlikely: $\widetilde{\ell}(\cdot, \widetilde{y}) \to \ell(\cdot, \widetilde{y})$, i.e, it suffers what it would normally suffer as the given label is (most likely) the true label.

- $\rho_{\boldsymbol{x}} \in [0.5, 1]$, i.e., high probability for label flip: $\widetilde{\ell}(\cdot, \widetilde{y})$ is a (nontrivial) weighted combination of $\ell(\cdot, \widetilde{y})$ and $\ell(\cdot, -\widetilde{y})$, which is reasonable as it is these "gray" areas that the machine should be careful about not leaning towards one side, which in turn is making it more robust for generalization.

- $\widetilde{y} \cdot h(\boldsymbol{x}) \to \infty$, i.e., the machine is strongly agreeing with the given annotation: $\widetilde{\ell}(\cdot, \widetilde{y}) \to \ell(\cdot, \widetilde{y}) = 0$, i.e., it suffers no loss as it would not normally.

- $\widetilde{y} \cdot h(\boldsymbol{x}) \to -\infty$, i.e., the machine is strongly disagreeing with the given annotation: $\widetilde{\ell}(\cdot, \widetilde{y}) \to \ell(\cdot, -\widetilde{y}) = 0$; this is a significantly different behavior from a label noise-intolerant loss function, e.g., $\ell_{\log}$, which would make the network suffer the maximal loss. Here, though, we are facing an "obvious" mislabel. Instead of insisting on blindly agreeing with the annotation, the modified loss trusts the machine and moves on.

### 3.2.4 Stabilizing $\widetilde{\ell}$

The division in the formulation of $\widetilde{\ell}$ in equation 3 poses a danger – we have no mathematical control over the denominator and for some instances $\boldsymbol{x}$, it can become arbitrarily close to 0 leading to instability in practice. The same worry carries to the gradient, possibly exploding it. Therefore, we employ a regularization trick to address this: we approximate the division with repeated subtraction, and propose the *regularized* form of $\widetilde{\ell}$, called $\widetilde{\ell}^R$, as follows:

$$
\begin{aligned}
\widetilde{\ell}^R(h(\boldsymbol{x}), \widetilde{y}) &:= \widetilde{\ell}_{\text{numerator}} - \lambda \widetilde{\ell}_{\text{denominator}} \\
&= \sigma(\widetilde{y} \cdot h(\boldsymbol{x}))(\sigma(-\widetilde{y} \cdot h(\boldsymbol{x})) - \rho_{\boldsymbol{x}})\ell(h(\boldsymbol{x}), \widetilde{y}) - \sigma(-\widetilde{y} \cdot h(\boldsymbol{x}))\rho_{\boldsymbol{x}}\ell(h(\boldsymbol{x}), -\widetilde{y}) \quad (5) \\
&\quad - \lambda(\sigma(\widetilde{y} \cdot h(\boldsymbol{x}))\sigma(-\widetilde{y} \cdot h(\boldsymbol{x})) - \rho_{\boldsymbol{x}}),
\end{aligned}
$$

where $\lambda > 0$ is a hyperparameter. Even though the risk equivalence is hurt, the generalization ability with $\widetilde{\ell}^R$ remains intact under sufficient conditions on $\lambda$, which we show next with a high probability generalization bound. To this end, we first establish the Lipschitz continuity of $\widetilde{\ell}^R$.

**Lemma 1.** *Let $\ell(h(\boldsymbol{x}), y)$ be an L-Lipschitz (w.r.t. $h(\boldsymbol{x})$) loss function with a finite upper and lower bound, i.e., there exists an $\ell_\infty$ such that $|\ell(\cdot, \cdot)| \leq \ell_\infty < \infty$. Assume $-\infty < \inf_{\boldsymbol{x} \in \mathcal{X}} \frac{\partial \rho_{\boldsymbol{x}}}{\partial h(\boldsymbol{x})} =: \alpha_{min} \leq 0 \leq \alpha_{max} := \sup_{\boldsymbol{x} \in \mathcal{X}} \frac{\partial \rho_{\boldsymbol{x}}}{\partial h(\boldsymbol{x})} < \infty$. Then, $\widetilde{\ell}^R(h(\boldsymbol{x}), \widetilde{y})$ is Lipschitz w.r.t. $h(\boldsymbol{x})$ with the constant*

$$\widetilde{L}_R := \big(1 + \frac{3}{2}(\alpha_{max} - \alpha_{min})\big)\ell_\infty + \frac{3}{2}L + \big(\frac{3}{8} + \alpha_{max} - \alpha_{min}\big)\lambda.$$

We note that the assumption $\alpha_{\min} \leq 0 \leq \alpha_{\max}$ is reasonable as $\alpha_{\min}\alpha_{\max} > 0$ would imply $\rho_{\boldsymbol{x}}$ as a monotonic function of $h(\boldsymbol{x})$, which is not realistic since the confidence (and the distance of $\boldsymbol{x}$ to the

decision boundary) of the machine is at its maximum when both $h(\boldsymbol{x}) \to +\infty$ and $h(\boldsymbol{x}) \to -\infty$. We address the boundedness imposition on $\ell$ (and $\widetilde{\ell}$) after the next proposition.

**Proposition 2.** *Given a noisy training sample $S = \{(\boldsymbol{x}_i, \widetilde{y}_i)\}$, a base loss function $\ell$, any hypothesis $\mathcal{H} \ni h : \mathcal{X} \to \mathbb{R}$, let the vectors $\widetilde{\boldsymbol{n}}_S$ and $\widetilde{\boldsymbol{d}}_S$ denote the evaluations of the numerator and the denominator of $\widetilde{\ell}$ over $S$, respectively. Now define $\Delta_S \coloneqq \langle \widetilde{\boldsymbol{n}}_S, \widetilde{\boldsymbol{d}}_S \rangle^2 - ||\widetilde{\boldsymbol{n}}_S||_2^2 ||\widetilde{\boldsymbol{d}}_S||_2^2$ and assume $|\widetilde{\ell}(\cdot, \cdot)| \leq \widetilde{\ell}_\infty < \infty$. For $\lambda > 0$, provided that $\lambda \widetilde{\boldsymbol{d}}_S[i] \leq \widetilde{\boldsymbol{n}}_S[i] \cdot (1 - 1/\widetilde{\boldsymbol{d}}_S[i]) \, \forall i \in [1, |S|]$, $\left| \lambda - \frac{\langle \widetilde{\boldsymbol{n}}_S, \widetilde{\boldsymbol{d}}_S \rangle}{||\widetilde{\boldsymbol{d}}_S||_2^2} \right| \leq \frac{\sqrt{\Delta_S + 4|S| ||\widetilde{\boldsymbol{d}}_S||_2^2}}{||\widetilde{\boldsymbol{d}}_S||_2^2}$ and $2|S|\widetilde{\ell}_\infty < \frac{-\Delta_S}{||\widetilde{\boldsymbol{d}}_S||_2^2} < 4|S|$, the following inequality holds with probability at least $1 - \delta$ for any $\delta \in (0, 1)$:*

$$R_\ell(h) \leq \widehat{\widetilde{R}}_{\widetilde{\ell}^R}(h) + 2\widetilde{L}_R \widehat{\mathcal{R}}_S(\mathcal{H}) + 3\widetilde{\ell}_\infty \sqrt{\frac{\log 2/\delta}{2|S|}}, \tag{6}$$

*where $\widetilde{L}_R$ is given by Lemma 1 and $\widehat{\mathcal{R}}_S(\mathcal{H})$ is the empirical Rademacher complexity of $\mathcal{H}$ over $S$.*

Although the practical utility of this bound is limited, it serves as a theoretical sanity check for replacing the numerically unstable modified loss in equation 4 with the more stable regularized loss $\widetilde{\ell}^R$. Indeed, Proposition 2 implies there exist settings for which there is a learning guarantee through empirical noisy risk minimization w.r.t. the regularized loss $\widetilde{\ell}^R$ towards true clean risk minimization w.r.t. $\ell$, i.e., as if the training was performed using the clean labels. If the base loss $\ell$ is classification calibrated (Bartlett et al., 2006), e.g., $\ell_{\log}$, the path in 2 is completed to reach $R(h)$. We note that while no common loss function is bounded (e.g., logistic, hinge, squared), it is not uncommon to clip the losses (see, e.g., Rooyen et al. (2015); Wang et al. (2019)), which does not necessarily hinder the classification calibratedness (Rooyen et al., 2015). Then, if the base loss $\ell$ is upper-bounded (say by $\ell_\infty$ by thresholding and then possibly downscaling), then $\widetilde{\ell}$ can be upper-bounded as well; an example is given in Appendix A.6.

## 4 EXPERIMENTS

Here, we first validate the proposed theory using synthetic noise over datasets from the image, audio and tabular domains using neural networks and decision tree-based models. We then experiment with a real-world dataset, i.e., a naturally noisy one. We note that we also perform a sanity check study to show the effectiveness of the loss correction scheme in Appendix A.2.

### 4.1 SETTINGS

We use the following datasets: on the synthetic noise side, CIFAR-10 (Krizhevsky, 2009) for images; Speakers (Rimi, 2023) for audio signals (to feature a time series dataset); and *Adult* (Becker & Kohavi, 1996), *Diabetes* (Bennett et al., 1971), *Heart* (Janosi et al., 1989), *Splice* (Towell et al., 1991) and *Segmentation* (Brodley, 1990) for tabular (to use a machine other than a neural network). For a naturally noisy dataset, we use Clothing1M (Xiao et al., 2015). As our method is aimed at binary classification, we split the multiclass-aimed datasets into arbitrary 2-class sub-datasets. The ratio of the positive class in sub-datasets is provided in Appendix A.7; most of the datasets are balanced. We use six-layer ReLU CNNs for CIFAR-10 and Clothing1M, and a three-layer ReLU MLP for the other datasets. For CIFAR-10 and Clothing1M, the provided training-test splits are used (no clean training data is used for the latter); for others, 80%-20% split is done, where only in the test set are the clean labels used. In all datasets, 10% of the (noisy) training split is spared for validation. In all the experiments, the base loss function to correct is the logistic loss $\ell_{\log}$ and the noise rates refer to the fraction of labels that are actually flipped. Details of the datasets, networks, preprocessing, optimization and hyperparameter tuning can be found in Appendix A.8; a hyperparameter sensitivity study of our method is presented in Appendix A.3.

### 4.2 METHODS FROM THE LITERATURE

We compare our model with methods of different characteristics from the literature, on both synthetic and real-world experiments. While the details of these methods are given in Appendix A.9,

| Method | CIFAR-10 | | | | | | | | | |
|---|---|---|---|---|---|---|---|---|---|---|
| | 0v1 | | 2v3 | | 4v5 | | 6v7 | | 8v9 | |
| | 28% | 44% | 28% | 44% | 28% | 44% | 28% | 44% | 28% | 44% |
| Normal | 76.12 | 64.23 | 53.42 | 50.00 | 61.58 | 50.00 | 64.50 | 52.73 | 68.38 | 57.00 |
| BCN | 56.82 | 50.00 | 56.43 | 51.33 | 55.92 | 52.58 | 70.28 | 55.78 | 57.98 | 52.73 |
| UB | 76.72 | 62.62 | 66.93 | 55.23 | 67.15 | 62.63 | 76.52 | 62.33 | 73.72 | 61.43 |
| DMI | 68.43 | 61.87 | 56.05 | 55.87 | 60.73 | 60.00 | 69.52 | **69.97** | 65.72 | 65.82 |
| Peer | 66.18 | 69.35 | 61.88 | **57.90** | 57.22 | 59.02 | 66.45 | 63.42 | 65.97 | 67.22 |
| APL | 79.82 | 70.47 | 70.37 | 54.48 | 75.02 | 64.77 | **82.13** | **70.42** | 78.85 | 67.53 |
| PTD | 76.88 | 56.85 | 66.98 | 55.43 | 70.17 | 57.55 | 78.07 | 62.90 | 76.65 | **68.98** |
| BLTM | 77.13 | 33.00 | 55.00 | 33.00 | 67.90 | 21.72 | 78.58 | 19.67 | 70.28 | 21.28 |
| GCE | 78.63 | 54.42 | 54.65 | 50.00 | 66.40 | 50.00 | 78.45 | 50.00 | 75.77 | 53.23 |
| Coteaching+ | 80.22 | 67.23 | 64.58 | 55.00 | 73.57 | 65.17 | **82.43** | 65.53 | 80.57 | 65.40 |
| Forward | 73.00 | 53.40 | 63.85 | 50.00 | 71.90 | 50.00 | 80.77 | 50.98 | 81.00 | 50.00 |
| PLC | 74.87 | 66.48 | 54.73 | 52.97 | 68.88 | **65.52** | 70.75 | 68.78 | 69.18 | 63.98 |
| NDX | **83.43** | **77.33** | **72.38** | 56.45 | **76.77** | **66.70** | **83.27** | **70.23** | **82.98** | 64.78 |

Table 1: Mean test accuracy (%) comparisons on CIFAR-10's five different binary sub-datasets with varying noise levels over three trials. Scores within 2% of the maximum (relative) are highlighted in bold.

we list their names here: *BCN* (Du & Cai, 2015); *UB* (Natarajan et al., 2013); *DMI* (Xu et al., 2019); *Peer* (Liu & Guo, 2020); *APL* (Ma et al., 2020); *PTD* (Xia et al., 2020); *BLTM* (Yang et al., 2022); *Coteaching+* (Yu et al., 2019); *Forward* & *Backward* (Patrini et al., 2017); *GCE* (Zhang & Sabuncu, 2018) and *PLC* (Zhang et al., 2021). We also compare against the *Normal* model trained with the logistic loss, which does nothing special for label noise.

## 4.3 SYNTHETIC LABEL NOISE EXPERIMENTS

We artificially inject label noise to the training sets of CIFAR-10, Speakers and Tabular datasets to simulate the problem. The injection is done by following the procedure in Xia et al. (2020) in an instance-dependent manner (that is *not* matched with the distance-to-decision boundary model we are using for $\rho_{\boldsymbol{x}}$). We experiment with two noise rates: 28% and 44%, which are representatives of moderate and high noise rates for binary classification, respectively. Three independent trials per noise rate are made such that possibly different labels are flipped in each trial.

**Image Results.** We form 5 binary sub-datasets of CIFAR-10. Results in Table 1 suggest that not only the test set accuracy of NDX is overall better or comparable to other methods, it is also more "stable" – the performance does not sweep much across datasets as much as other models. It is also featuring in high noise regime – while some models go astray under 44% noise, NDX manages to learn from the signal present. The rationale for achieving these can be attributed to the promised risk equivalence accompanying better generalization provided that the noise model is fine. While knowing the exact flip rates is practically impossible, the flexibility in modeling it via the instance-only dependent model opens doors for good approximations, combined with the risk equalizer loss correction.

**Audio Results.** Over 5 binary sub-datasets of the Speakers dataset, Table 2 shows that NDX is always a top-2 performing method regardless of the sub-dataset or the noise level, which speaks for its robustness. Furthermore, since the method has no dataset-specific assumptions (e.g., image-based), it is effectively applicable to a dataset of audio signals.

**Tabular Results.** Since a loss correction approach is generally machine-agnostic by design, here we experiment with several tabular datasets with LightGBM (Ke et al., 2017), a decision tree-based gradient-boosting machine, being the underlying learner $h$ for our method. Since $\widetilde{\ell}_{\log}^{R}$ with $\rho_{\boldsymbol{x}} = \sigma(-\beta|h(\boldsymbol{x})|)$ is twice-differentiable w.r.t. $h(\boldsymbol{x})$, the gradient and Hessian values are available to customize the loss function used for LightGBM. We still use a neural network for the other methods since some of them are not compatible with machines other than neural networks (e.g., *PTD*, *Coteaching+*), and others only exposed implementations with neural networks (e.g., *GCE*, *PLC*).

The results are shown in Table 3, where *NDX (NN)* and *NDX (LGBM)* are our models with the learning machine being a neural network and a LightGBM model, respectively. We observe that NDX models perform quite well compared to the other methods across datasets and noise levels. GBM models are known for their effectiveness in tabular datasets (Shwartz-Ziv & Armon, 2022);

| Method | Speakers | | | | | | | | | |
| --- | --- | --- | --- | --- | --- | --- | --- | --- | --- | --- |
| | 0v1 | | 2v3 | | 4v5 | | 6v7 | | 8v9 | |
| | 28% | 44% | 28% | 44% | 28% | 44% | 28% | 44% | 28% | 44% |
| Normal | 69.29 | 56.68 | 62.94 | 55.13 | 69.49 | 54.17 | 58.47 | 51.85 | 60.55 | 52.83 |
| BCN | 81.40 | 62.30 | 62.70 | 44.41 | 45.22 | 45.22 | 67.90 | 58.25 | 52.25 | 52.25 |
| UB | 69.54 | 60.80 | 64.92 | 56.64 | 57.48 | 57.35 | 67.34 | 56.79 | 60.32 | 55.71 |
| DMI | 50.31 | 61.55 | 58.39 | 53.26 | 51.59 | 51.59 | 61.50 | 63.08 | 53.75 | 52.71 |
| Peer | 74.16 | **72.16** | 63.52 | **69.70** | 58.33 | 62.38 | 71.16 | **69.25** | 58.02 | **67.01** |
| APL | **82.65** | 62.55 | 78.09 | 56.99 | 69.98 | 60.05 | **79.69** | 58.92 | 69.78 | 57.55 |
| PTD | **83.65** | 67.42 | **79.14** | 51.63 | 60.91 | 50.00 | 76.99 | 60.72 | **73.01** | 59.63 |
| BLTM | 79.78 | 63.67 | 77.04 | 55.94 | 69.85 | 51.35 | 72.73 | 53.42 | 56.63 | 57.21 |
| GCE | 77.40 | 59.05 | 75.06 | 53.38 | 70.34 | 55.51 | **78.23** | 53.87 | 68.97 | 56.40 |
| Coteaching+ | 79.03 | 67.04 | 72.14 | 49.65 | 68.26 | 60.78 | 77.55 | 56.12 | 65.97 | 62.86 |
| Backward | 77.15 | 60.67 | 71.10 | 52.10 | 66.54 | 55.02 | 71.16 | 52.30 | 66.67 | 54.90 |
| PLC | 75.03 | 64.29 | 75.29 | 58.16 | 68.50 | 61.52 | 73.85 | 54.10 | 61.48 | 56.52 |
| NDX | **83.77** | **72.16** | **80.30** | 64.10 | **74.51** | **66.91** | 79.35 | **69.70** | **72.32** | 65.40 |

Table 2: Mean test accuracy (%) comparisons on the Speakers dataset's five different binary sub-datasets with varying noise levels over three trials. Scores within 2% of the maximum (relative) are highlighted in bold.

| Method | Tabular Datasets | | | | | | | | | |
| --- | --- | --- | --- | --- | --- | --- | --- | --- | --- | --- |
| | Adult | | Diabetes | | Heart | | Segmentation | | Splice | |
| | 28% | 44% | 28% | 44% | 28% | 44% | 28% | 44% | 28% | 44% |
| Normal | 82.72 | 66.06 | 64.94 | 65.58 | 51.37 | 49.18 | 57.94 | 65.01 | 52.02 | 46.19 |
| BCN | **83.86** | 80.05 | 38.96 | 38.83 | 63.39 | 58.47 | 81.89 | 68.18 | 81.84 | 67.45 |
| UB | **84.20** | **80.89** | 71.10 | 65.97 | 73.77 | 60.66 | 88.53 | 74.39 | 82.41 | 69.03 |
| DMI | 76.38 | 75.64 | 65.58 | **68.40** | 59.02 | 49.73 | 66.88 | 71.50 | 70.24 | 67.72 |
| Peer | 78.91 | 79.89 | 68.18 | **68.83** | 59.56 | 59.02 | 85.50 | **76.26** | 73.28 | 72.07 |
| APL | **83.64** | 80.20 | 66.88 | **68.61** | **81.42** | 51.91 | **90.04** | **74.82** | 84.04 | 67.14 |
| PTD | 78.00 | 76.39 | 66.45 | 67.75 | 59.56 | 54.10 | 78.07 | 47.11 | 82.31 | 55.33 |
| BLTM | **84.05** | 69.57 | 64.94 | 64.94 | 49.18 | 49.18 | 65.01 | 59.60 | 55.85 | 45.83 |
| GCE | **84.38** | 76.38 | 64.94 | 64.94 | 50.82 | 50.82 | 57.79 | 57.58 | 54.17 | 54.17 |
| Coteaching+ | **83.65** | 78.22 | 71.00 | 67.75 | 68.85 | 46.99 | 87.81 | 66.23 | 78.79 | 62.36 |
| Forward | **83.17** | 78.39 | 65.37 | 54.98 | 52.46 | 52.46 | 74.39 | 57.79 | 83.62 | 55.22 |
| PLC | 76.38 | 77.85 | 64.94 | 66.23 | 74.86 | 50.82 | 72.87 | 70.35 | 78.79 | 68.66 |
| NDX (NN) | **84.59** | 80.33 | **73.16** | **69.48** | **81.97** | 61.20 | **91.20** | 71.28 | 82.73 | 67.30 |
| NDX (GBM) | 82.63 | **82.23** | **73.34** | 68.83 | 77.05 | **73.77** | 85.06 | 72.73 | **88.35** | **78.90** |

Table 3: Mean test accuracy (%) comparisons on five different tabular datasets with varying noise levels over three trials. Scores within 2% of the maximum (relative) are highlighted in bold.

here we also see it in action: in the *Heart* dataset with 44% label noise, for example, LightGBM performed more than 10% better than the runner-up in absolute terms. Overall, we were able to use an entirely different learning machine, a decision tree-based one, instead of a neural network thanks to the machine-agnostic nature of the loss correction.

## 4.4 REAL-WORLD LABEL NOISE EXPERIMENT

Here we experiment with a naturally noisy dataset, Clothing1M (Xiao et al., 2015). We use the $\sim$1,000,000 noisy training pairs for training, and $\sim$10,000 clean testing pairs for testing the models. Note that the dataset also comes with clean training and validation splits but we discard them for all models. We form 10 binary sub-datasets of Clothing1M, e.g., "6v8" in the header represents the task "Wind-breaker versus Down coat". Results in Table 4 suggest that our model has also competitive performance in a real-world scenario by achieving comparable or mostly better testing accuracy metrics in comparison to the baseline models.

## 5 CONCLUSION

We introduced a risk-equivalence based instance-aware loss correction approach to address label noise in supervised binary classification settings with the underutilized instance-*only* dependent noise model. We showed that when the base loss $\ell$ is classification calibrated, the bridge from empirical noisy risk minimization to true clean risk minimization is complete, i.e., training a machine with the new loss $\widetilde{\ell}$ on the noisy labels promises a generalization accuracy as if it was trained on the clean labels. The instance-only dependent modeling of the transition rates is at the core of the correction scheme, which is highly flexible in how it is approximated, for which we offered several computationally efficient ways. The performance of the corrected loss is empirically validated over

| Method | Clothing1M | | | | | | | | | |
|---|---|---|---|---|---|---|---|---|---|---|
| | 6v8 | 6v7 | 6v9 | 1v6 | 2v6 | 0v2 | 2v9 | 2v11 | 1v7 | 0v11 |
| Normal | 62.79 | 74.85 | 69.03 | 73.53 | 67.10 | 65.67 | 68.42 | 71.97 | 72.88 | 71.46 |
| BCN | 63.24 | 73.10 | 61.56 | 69.60 | 60.46 | 61.59 | 55.42 | 69.50 | 70.20 | 67.21 |
| UB | 65.65 | 70.29 | 64.90 | 65.33 | 58.06 | 63.56 | 57.74 | 57.71 | 71.06 | 62.77 |
| DMI | 64.63 | 67.95 | 64.44 | 66.33 | 58.06 | 61.59 | 57.89 | 59.27 | 68.70 | 62.31 |
| Peer | 65.21 | 71.11 | 67.39 | 70.94 | 63.69 | 64.87 | 65.39 | 69.91 | 72.88 | 68.52 |
| APL | 76.06 | 78.95 | 76.73 | 79.73 | **80.37** | 69.97 | **77.35** | **79.23** | **79.42** | **77.34** |
| PTD | 66.35 | 78.95 | 68.40 | 77.39 | 71.89 | 58.82 | 62.60 | 77.49 | 75.35 | 72.24 |
| BLTM | 64.57 | 73.80 | 59.14 | 67.50 | 53.27 | 67.06 | 58.13 | 56.55 | 72.78 | 65.97 |
| GCE | 70.54 | 75.20 | 71.13 | 71.78 | 72.63 | 67.64 | 72.49 | 72.46 | 75.78 | 75.31 |
| Coteaching+ | 75.17 | 78.36 | 66.54 | **82.83** | 78.25 | 67.78 | 75.12 | 77.82 | **79.53** | **77.20** |
| Backward | 69.59 | **80.70** | **77.43** | **83.08** | **80.83** | 65.96 | 74.80 | 76.83 | **78.56** | **76.88** |
| PLC | 66.22 | 74.50 | 64.75 | 70.69 | 61.29 | 65.89 | 67.07 | 75.52 | 73.95 | 68.13 |
| NDX | **78.10** | **79.88** | **78.37** | **81.57** | **80.92** | **72.96** | **78.71** | **79.80** | **78.78** | **78.25** |

Table 4: Test accuracy (%) comparisons on Clothing1M's ten different binary sub-datasets. Scores within 2% of the maximum (relative) are highlighted in bold.

a variety of datasets of different domains as well as different underlying machines. A natural step forward is the multi-class/label generalization of the instance-only dependent noise model.

ACKNOWLEDGMENTS

The authors acknowledge the financial support of the Flemish Government (AI Research Program), METHUSALEM: METH/26/003; the FWO (Research Foundation Flanders) for projects G026026N, G081722N and G046925N; Internal Funds KU Leuven (projects C3/25/017, IDN/23/006 and C14/25/108); and HORIZON-HLTH-2022-STAYHLTH under EU Grant Agreement #101080581.

A. Bertrand, M. De Vos and Mustafa Enes Aydın are affiliated to Leuven.AI - KU Leuven institute for AI, B-3000, Leuven, Belgium.

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

# A  APPENDIX

## A.1  PROOFS

*Proof of Proposition 1.*  We start from the desired risk equivalence:

$$\mathbb{E}_{X,\widetilde{Y}}[\widetilde{\ell}(\cdot,\widetilde{Y})] = \mathbb{E}_{X,Y}[\ell(\cdot,Y)] \tag{1}$$

$$\mathbb{E}_X[\mathbb{E}_{\widetilde{Y}|X}[\widetilde{\ell}(\cdot,\widetilde{Y}) \mid X]] = \mathbb{E}_X[\mathbb{E}_{Y|X}[\ell(\cdot,Y) \mid X]] \tag{2}$$

$$\int_{\mathcal{X}} \mathbb{E}_{\widetilde{Y}|X}[\widetilde{\ell}(\cdot,\widetilde{Y}) \mid X = \boldsymbol{x}]\, dp_X(\boldsymbol{x}) = \int_{\mathcal{X}} \mathbb{E}_{Y|X}[\ell(\cdot,Y) \mid X = \boldsymbol{x}]\, dp_X(\boldsymbol{x}). \tag{3}$$

We now expand the integrands and require them to be equal for all $\boldsymbol{x} \in \mathcal{X}$, which is a sufficient condition for equation 3:

$$\mathbb{P}(\widetilde{Y} = +1 \mid X = \boldsymbol{x})\,\widetilde{\ell}(\cdot,+1) + \mathbb{P}(\widetilde{Y} = -1 \mid X = \boldsymbol{x})\,\widetilde{\ell}(\cdot,-1)$$
$$= \mathbb{P}(Y = +1 \mid X = \boldsymbol{x})\,\ell(\cdot,+1) + \mathbb{P}(Y = -1 \mid X = \boldsymbol{x})\,\ell(\cdot,-1). \tag{4}$$

To make use of $\rho_{\boldsymbol{x}} := \mathbb{P}(\widetilde{Y} \neq Y \mid X = \boldsymbol{x})$, we employ the following trick based on a (trivial) equality: $\mathbb{P}(\widetilde{Y} = +1 \mid \boldsymbol{x}) = (\mathbb{P}(\widetilde{Y} = +1 \mid \boldsymbol{x}) - \rho_{\boldsymbol{x}}) + \rho_{\boldsymbol{x}}$, and similar for the $\widetilde{Y} = -1$ case. Substituting these into equation 4, we get

$$\big(\mathbb{P}(\widetilde{Y} = +1 \mid X = \boldsymbol{x}) - \rho_{\boldsymbol{x}}\big)\widetilde{\ell}(\cdot,+1) + \rho_{\boldsymbol{x}}\widetilde{\ell}(\cdot,+1)$$
$$+\big(\mathbb{P}(\widetilde{Y} = -1 \mid X = \boldsymbol{x}) - \rho_{\boldsymbol{x}}\big)\widetilde{\ell}(\cdot,-1) + \rho_{\boldsymbol{x}}\widetilde{\ell}(\cdot,-1)$$
$$= \mathbb{P}(Y = +1 \mid X = \boldsymbol{x})\,\ell(\cdot,+1) + \mathbb{P}(Y = -1 \mid X = \boldsymbol{x})\,\ell(\cdot,-1). \tag{5}$$

We now split equation 5 into two parts and form the following linear system, which is a sufficient condition for equation 5:

$$\big(\mathbb{P}(\widetilde{Y} = +1 \mid X = \boldsymbol{x}) - \rho_{\boldsymbol{x}}\big)\widetilde{\ell}(\cdot,+1) + \rho_{\boldsymbol{x}}\widetilde{\ell}(\cdot,-1) = \mathbb{P}(Y = +1 \mid X = \boldsymbol{x})\,\ell(\cdot,+1)$$
$$\rho_{\boldsymbol{x}}\widetilde{\ell}(\cdot,+1) + \big(\mathbb{P}(\widetilde{Y} = -1 \mid X = \boldsymbol{x}) - \rho_{\boldsymbol{x}}\big)\widetilde{\ell}(\cdot,-1) = \mathbb{P}(Y = -1 \mid X = \boldsymbol{x})\,\ell(\cdot,-1). \tag{6}$$

Assuming a unique solution, equation 6 yields

$$\widetilde{\ell}(\cdot,+1) = \frac{\mathbb{P}(Y = +1 \mid \boldsymbol{x})\,(\mathbb{P}(\widetilde{Y} = -1 \mid \boldsymbol{x}) - \rho_{\boldsymbol{x}})\,\ell(\cdot,+1) - \mathbb{P}(Y = -1 \mid \boldsymbol{x})\rho_{\boldsymbol{x}}\,\ell(\cdot,-1)}{\mathbb{P}(\widetilde{Y} = +1 \mid \boldsymbol{x})\mathbb{P}(\widetilde{Y} = -1 \mid \boldsymbol{x}) - \rho_{\boldsymbol{x}}}$$

$$\widetilde{\ell}(\cdot,-1) = \frac{\mathbb{P}(Y = -1 \mid \boldsymbol{x})\,(\mathbb{P}(\widetilde{Y} = +1 \mid \boldsymbol{x}) - \rho_{\boldsymbol{x}})\,\ell(\cdot,-1) - \mathbb{P}(Y = +1 \mid \boldsymbol{x})\rho_{\boldsymbol{x}}\,\ell(\cdot,+1)}{\mathbb{P}(\widetilde{Y} = +1 \mid \boldsymbol{x})\mathbb{P}(\widetilde{Y} = -1 \mid \boldsymbol{x}) - \rho_{\boldsymbol{x}}}. \tag{7}$$

We unite the individual loss terms in equation 7 to arrive at

$$\widetilde{\ell}(\cdot,\widetilde{y}) = \frac{\mathbb{P}(Y = \widetilde{y} \mid \boldsymbol{x})(\mathbb{P}(\widetilde{Y} = -\widetilde{y} \mid \boldsymbol{x}) - \rho_{\boldsymbol{x}})\ell(\cdot,\widetilde{y}) - \mathbb{P}(Y = -\widetilde{y} \mid \boldsymbol{x})\rho_{\boldsymbol{x}}\ell(\cdot,-\widetilde{y})}{\mathbb{P}(\widetilde{Y} = \widetilde{y} \mid \boldsymbol{x})\mathbb{P}(\widetilde{Y} = -\widetilde{y} \mid \boldsymbol{x}) - \rho_{\boldsymbol{x}}}. \tag{8}$$

$\square$

We make the following remarks about the proof.
*Why subtract $\rho_{\boldsymbol{x}}$ in the trick and not anything else?*  In a CCN label noise model for a binary classification, we would have the two noise rates as $\rho_+ := \mathbb{P}(\widetilde{Y} = -1 \mid Y = +1)$ and $\rho_- := \mathbb{P}(\widetilde{Y} = +1 \mid Y = -1)$. In Natarajan et al. (2013), the following relationship was then proposed to construct a modified loss for the CCN setting:

$$(1 - \rho_+)\widetilde{\ell}(\cdot,+1) + \rho_+\widetilde{\ell}(\cdot,-1) = \ell(\cdot,+1)$$
$$\rho_-\widetilde{\ell}(\cdot,+1) + (1 - \rho_-)\widetilde{\ell}(\cdot,-1) = \ell(\cdot,-1). \tag{9}$$

We argue that equation 6 forms a natural extension of equation 9 to the IDN setting, including some additional corrections to ensure risk equivalence in equation 1. First, on the RHS we add probabilities on $Y$ to take into account the expectation on $Y$ in the RHS of equation 1. Similarly,

on the LHS, the non-flip rates (set to 1 in equation 9) should take the uncertainty on $\widetilde{Y}$ into account according to the LHS of equation 1. Finally, the single probability of flip $\rho_{\boldsymbol{x}}$ is in lieu of two different ones in the CCN. The resulting system hosts a weighted combination of $\widetilde{\ell}$ for a "robust" loss correction in the label noise sense, i.e., the weights are the label flip and non-flip probabilities in each of the $\pm 1$ cases.

*What if there is no solution to the linear system?* That happens if and only if $\mathbb{P}(\widetilde{Y} = \widetilde{y} \mid \boldsymbol{x})\mathbb{P}(\widetilde{Y} = -\widetilde{y} \mid \boldsymbol{x}) = \rho_{\boldsymbol{x}}$ for a given $\boldsymbol{x}, \widetilde{y}$ pair. Mathematically, there is nothing preventing this (though, certain measures can be taken in generating $\rho_{\boldsymbol{x}}$ to satisfy, e.g., $\mathbb{P}(\widetilde{Y} = \widetilde{y} \mid \boldsymbol{x})\mathbb{P}(\widetilde{Y} = -\widetilde{y} \mid \boldsymbol{x}) > \rho_{\boldsymbol{x}}$ for any $\boldsymbol{x}, \widetilde{y}$). However, since the existence of the division leads to numeric issues to begin with (e.g., being close to 0 for some instances $\boldsymbol{x}$), we go for the regularized version of the loss function and transfer the worry to the regularization coefficient's selection, which is an easier issue to handle.

*Proof of Lemma 1.* Since $\widetilde{\ell}^R$ is differentiable w.r.t. $h(\boldsymbol{x})$, it suffices to seek an upper bound to (the absolute value of) its derivative w.r.t. $h(\boldsymbol{x})$. We consider four cases of $\{\widetilde{y} = \pm 1\} \times \{h(\boldsymbol{x}) \lessgtr 0\}$ and take the maximum of the bounds found therein.

$$
\left|\frac{\partial \widetilde{\ell}^R(h(\boldsymbol{x}), \widetilde{y})}{\partial h(\boldsymbol{x})}\right| = \left|\sigma(h(\boldsymbol{x}))\widetilde{y}\sigma(-h(\boldsymbol{x}))(\sigma(-\widetilde{y}h(\boldsymbol{x})) - \rho_{\boldsymbol{x}})\ell(h(\boldsymbol{x}), \widetilde{y})\right.
$$
$$
+ \sigma(\widetilde{y}h(\boldsymbol{x}))\left((\sigma(h(\boldsymbol{x}))(-\widetilde{y}\sigma(-h(\boldsymbol{x}))) - \frac{\partial \rho_{\boldsymbol{x}}}{\partial h(\boldsymbol{x})})\ell(h(\boldsymbol{x}), \widetilde{y}) + (\widetilde{y}\sigma(-h(\boldsymbol{x})) - \rho_{\boldsymbol{x}})\frac{\partial \ell(h(\boldsymbol{x}), \widetilde{y})}{\partial h(\boldsymbol{x})}\right)
$$
$$
- \left(\sigma(h(\boldsymbol{x}))(-\widetilde{y}\sigma(-h(\boldsymbol{x})))\rho_{\boldsymbol{x}}\ell(h(\boldsymbol{x}), -\widetilde{y})\right.
$$
$$
\left. + (1 - \sigma(-\widetilde{y}h(\boldsymbol{x})))(\frac{\partial \rho_{\boldsymbol{x}}}{\partial h(\boldsymbol{x})}\ell(h(\boldsymbol{x}), -\widetilde{y}) + \rho_{\boldsymbol{x}}\frac{\partial \ell(h(\boldsymbol{x}), -\widetilde{y})}{\partial h(\boldsymbol{x})})\right)
$$
$$
\left. - \lambda\left(\sigma(h(\boldsymbol{x}))(1 - \sigma(h(\boldsymbol{x})))^2 + \sigma(h(\boldsymbol{x}))^2(\sigma(h(\boldsymbol{x})) - \rho_{\boldsymbol{x}}) - \frac{\partial \rho_{\boldsymbol{x}}}{\partial h(\boldsymbol{x})}\right)\right|.
$$

(1)

*Case 1:* $\widetilde{y} = +1, h(\boldsymbol{x}) > 0.$

$$
\left|\frac{\partial \widetilde{\ell}^R(h(\boldsymbol{x}), +1)}{\partial h(\boldsymbol{x})}\right| \leq \max\left\{\left(\frac{3}{4} - \alpha_{\min}\right)\ell_\infty + \frac{1}{2}L + \left(\frac{3}{8} + \alpha_{\max}\right)\lambda, \right.
$$
$$
\left. \left(\frac{1}{2} + \frac{3}{2}\alpha_{\max}\right)\ell_\infty + \frac{3}{2}L + \left(\frac{1}{8} - \alpha_{\min}\right)\lambda\right\}.
$$

(2)

*Case 2:* $\widetilde{y} = -1, h(\boldsymbol{x}) > 0.$

$$
\left|\frac{\partial \widetilde{\ell}^R(h(\boldsymbol{x}), -1)}{\partial h(\boldsymbol{x})}\right| \leq \max\left\{\left(\frac{1}{4} - \frac{3}{2}\alpha_{\min}\right)\ell_\infty + \frac{1}{2}L + \left(\frac{3}{8} + \alpha_{\max}\right)\lambda, \right.
$$
$$
\left. \left(1 + \alpha_{\max}\right)\ell_\infty + L + \left(\frac{1}{4} - \alpha_{\min}\right)\lambda\right\}.
$$

(3)

*Case 3:* $\widetilde{y} = +1, h(\boldsymbol{x}) < 0.$

$$
\left|\frac{\partial \widetilde{\ell}^R(h(\boldsymbol{x}), +1)}{\partial h(\boldsymbol{x})}\right| \leq \max\left\{\left(\frac{7}{8} - \frac{1}{2}\alpha_{\min} + \alpha_{\max}\right)\ell_\infty + \frac{3}{2}L + \left(\frac{1}{4} - \alpha_{\min}\right)\lambda, \right.
$$
$$
\left. \left(\frac{1}{4} + \frac{3}{2}\alpha_{\max}\right)\ell_\infty + \frac{3}{2}L + \left(\frac{3}{8} - \alpha_{\min}\right)\lambda\right\}.
$$

(4)

*Case 4:* $\widetilde{y} = -1, h(\boldsymbol{x}) < 0.$

$$
\left|\frac{\partial \widetilde{\ell}^R(h(\boldsymbol{x}), +1)}{\partial h(\boldsymbol{x})}\right| \leq \max\left\{\left(\frac{1}{2} - \alpha_{\min}\right)\ell_\infty + \frac{1}{2}L + \left(\frac{1}{4} + \alpha_{\max}\right)\lambda, \right.
$$
$$
\left. \left(\frac{1}{2} + \frac{3}{2}\alpha_{\max}\right)\ell_\infty + \frac{3}{2}L + \left(\frac{3}{8} - \alpha_{\min}\right)\lambda\right\}.
$$

(5)

Combining the upper bounds in 2–5 with "element-wise" maximums yields the given $\widetilde{L}_R$. □

*Proof of Proposition 2.* The generic empirical Rademacher complexity-based generalization bound is (Mohri et al., 2012)

$$\widetilde{R}_{\widetilde{\ell}}(h) \leq \widehat{\widetilde{R}}_{\widetilde{\ell}}(h) + 2\widehat{\mathcal{R}}_S(\widetilde{\ell} \circ \mathcal{H}) + 3\widetilde{\ell}_\infty \sqrt{\frac{\log 2/\delta}{2|S|}}. \tag{1}$$

Since $\widetilde{\ell}$ is derived from the risk equivalence between noisy and clean domains, we have $\widetilde{R}_{\widetilde{\ell}}(h) = R_\ell(h)$. To replace the empirical risk w.r.t. $\widetilde{\ell}$ with that w.r.t. $\widetilde{\ell}^R$, we need $\lambda$ to satisfy $\widehat{\widetilde{R}}_{\widetilde{\ell}}(h) \leq \widehat{\widetilde{R}}_{\widetilde{\ell}^R}(h) \forall h \in \mathcal{H}$, i.e., $\frac{1}{|S|}\sum_{i=1}^{|S|}\widetilde{\ell}(h(\boldsymbol{x}_i), \widetilde{y}_i) \leq \frac{1}{|S|}\sum_{i=1}^{|S|}\widetilde{\ell}^R(h(\boldsymbol{x}_i), \widetilde{y}_i)$. We note that element-wise inequality is a sufficient condition for this to hold, i.e., $\widetilde{\ell}(h(\boldsymbol{x}_i), \widetilde{y}_i) \leq \widetilde{\ell}^R(h(\boldsymbol{x}_i), \widetilde{y}_i) \forall i \in [1, |S|]$. Writing the loss terms in terms of the numerator and denominator evaluations of $\widetilde{\ell}$, we have $\widetilde{\boldsymbol{n}}_S[i]/\widetilde{\boldsymbol{d}}_S[i] \leq \widetilde{\boldsymbol{n}}_S[i] - \lambda\widetilde{\boldsymbol{d}}_S[i] \forall i$. Rearranging this gives the first condition on $\lambda$ as $\lambda\widetilde{\boldsymbol{d}}_S[i] \leq \widetilde{\boldsymbol{n}}_S[i] \cdot (1 - 1/\widetilde{\boldsymbol{d}}_S[i]) \forall i \in [1, |S|]$. To "denoise" $\widehat{\mathcal{R}}_S(\widetilde{\ell} \circ \mathcal{H})$, i.e., be left with the complexity of the function space $\mathcal{H}$ only, we first show that it is upper-bounded by $\widehat{\mathcal{R}}_S(\widetilde{\ell}^R \circ \mathcal{H})$ under some sufficient conditions on $\lambda$, which is further upper-bounded by a constant multiple of $\widehat{\mathcal{R}}_S(\mathcal{H})$ using the Lipschitz composition property of the Rademacher complexity. To this end, recall the definition of the empirical Rademacher complexity:

$$\widehat{\mathcal{R}}_S(\widetilde{\ell} \circ \mathcal{H}) = \frac{1}{|S|}\mathbb{E}_\sigma\left[\sup_{h \in \mathcal{H}}\sum_{i=1}^{|S|}\sigma_i\widetilde{\ell}(h(\boldsymbol{x}_i), \widetilde{y}_i)\right] \tag{2}$$

$$= \frac{1}{|S|}\mathbb{E}_\sigma\left[\sup_{h \in \mathcal{H}}\langle\boldsymbol{\sigma}, \widetilde{\boldsymbol{\ell}}_{h,S}\rangle\right], \tag{3}$$

where $\sigma_i$ are i.i.d. Rademacher random variables, $\boldsymbol{\sigma} = [(\sigma_i)]_i^T$ and $\widetilde{\boldsymbol{\ell}}_{h,S} = [(\widetilde{\ell}(h(\boldsymbol{x}_i), \widetilde{y}_i))]_i^T$. Note that $\sigma_i$ and $\boldsymbol{x}_i$ are mutually independent. We first lower-bound $\widehat{\mathcal{R}}_S(\widetilde{\ell}^R \circ \mathcal{H})$ as follows:

$$\widehat{\mathcal{R}}_S(\widetilde{\ell}^R \circ \mathcal{H}) = \frac{1}{|S|}\mathbb{E}_\sigma\left[\sup_{h \in \mathcal{H}}\langle\boldsymbol{\sigma}, \widetilde{\boldsymbol{\ell}}_{h,S}^R\rangle\right] \geq \frac{1}{|S|}\mathbb{E}_\sigma\left[\langle\boldsymbol{\sigma}, \widetilde{\boldsymbol{\ell}}_{h,S}^R\rangle\right] \tag{4}$$

$$= \frac{1}{2|S|}\mathbb{E}_\sigma\left[||\boldsymbol{\sigma}||_2^2 + ||\widetilde{\boldsymbol{\ell}}_{h,S}^R||_2^2 - ||\boldsymbol{\sigma} - \widetilde{\boldsymbol{\ell}}_{h,S}^R||_2^2\right] \tag{5}$$

$$\geq \frac{1}{2|S|}||\widetilde{\boldsymbol{\ell}}_{h,S}^R||_2^2 \tag{6}$$

for any given $h \in \mathcal{H}$, where we used the definition of the supremum, polarization identity, independence of $\boldsymbol{\sigma}$ from $\widetilde{\boldsymbol{\ell}}_{h,S}^R$, monotonicity of expectation, the fact $||\boldsymbol{\sigma}||_2^2 = |S|$, and imposed $||\widetilde{\boldsymbol{\ell}}_{h,S}^R||_2^2 \leq 4|S|$. We now upper-bound $\widehat{\mathcal{R}}_S(\widetilde{\ell} \circ \mathcal{H})$ as follows:

$$\widehat{\mathcal{R}}_S(\widetilde{\ell} \circ \mathcal{H}) = \frac{1}{|S|}\mathbb{E}_\sigma\left[\sup_{h \in \mathcal{H}}\langle\boldsymbol{\sigma}, \widetilde{\boldsymbol{\ell}}_{h,S}\rangle\right] \leq \frac{1}{|S|}\mathbb{E}_\sigma\left[||\boldsymbol{\sigma}||_2 \sup_{h \in \mathcal{H}}||\widetilde{\boldsymbol{\ell}}_{h,S}||_2\right] \tag{7}$$

$$= \frac{1}{\sqrt{|S|}}\sup_{h \in \mathcal{H}}||\widetilde{\boldsymbol{\ell}}_{h,S}||_2 \leq \widetilde{\ell}_\infty, \tag{8}$$

where we used the Cauchy-Schwarz inequality and the independence assumption. Now we further impose $\lambda$ such that the upper-bound in equation 8 is to bound the lower-bound in equation 6 from below for any $h \in \mathcal{H}$, i.e.,

$$\widetilde{\ell}_\infty \leq \frac{1}{2|S|}||\widetilde{\boldsymbol{n}}_S - \lambda\widetilde{\boldsymbol{d}}_S||_2^2. \tag{9}$$

Combining this with the imposed assumption for equation 6, we have $2|S|\widetilde{\ell}_\infty \leq ||\widetilde{\boldsymbol{n}}_S - \lambda\widetilde{\boldsymbol{d}}_S||_2^2 \leq 4|S|$. This is a set of second degree polynomial inequalities over $\lambda$ and the following are sufficient for it to hold:

$$\left|\lambda - \frac{\langle\widetilde{\boldsymbol{n}}_S, \widetilde{\boldsymbol{d}}_S\rangle}{||\widetilde{\boldsymbol{d}}_S||_2^2}\right| \leq \frac{\sqrt{\Delta_S + 4|S|||\widetilde{\boldsymbol{d}}_S||_2^2}}{||\widetilde{\boldsymbol{d}}_S||_2^2}, 2|S|\widetilde{\ell}_\infty < \frac{-\Delta_S}{||\widetilde{\boldsymbol{d}}_S||_2^2} < 4|S|. \tag{10}$$

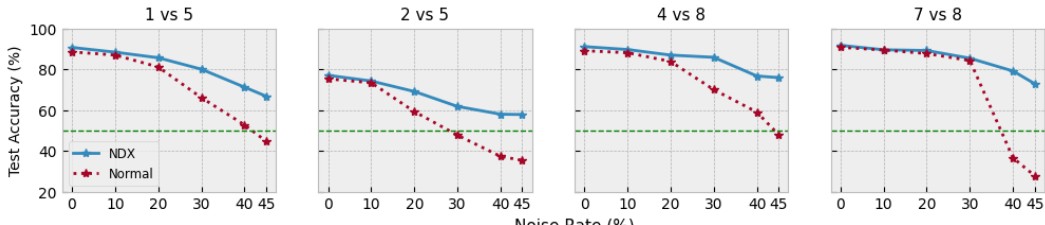

Figure 1: Test accuracy comparisons between the proposed method (NDX; solid) and the Normal model (dashed), i.e., $\widetilde{\ell}_{\log}^R$ versus $\ell_{\log}$ on 4 random sub-datasets of the CIFAR-10 dataset when the noise process is favorable. The horizontal dashed line is the 50% level, i.e., the test accuracy of a random guess model since the datasets are all balanced. The proposed method is quite stable and performs better than the model with the noise intolerant loss even when the noise rate is close to 50%, where the Normal model performs worse than the random guess.

With the Lipschitz composition property of Rademacher complexity, we have $\widehat{\mathcal{R}}_S(\widetilde{\ell}^R \circ \mathcal{H}) \leq \widetilde{L}_R \widehat{\mathcal{R}}_S(\mathcal{H})$. The chain is then completed, i.e.,

$$\widehat{\mathcal{R}}_S(\widetilde{\ell} \circ \mathcal{H}) \leq \widetilde{\ell}_\infty \leq \frac{1}{2|S|} \|\widetilde{\boldsymbol{n}}_S - \lambda \widetilde{\boldsymbol{d}}_S\|_2^2 \leq \widehat{\mathcal{R}}_S(\widetilde{\ell}^R \circ \mathcal{H}) \leq \widetilde{L}_R \widehat{\mathcal{R}}_S(\mathcal{H}), \tag{11}$$

and the proposition follows. $\qquad\square$

## A.2 SANITY CHECK OF THE LOSS CORRECTION WITH MATCHED NOISE MODEL

To demonstrate the effectiveness of the loss correction when the flip rates are well-modeled, we purposefully inject synthetic label noise to a given dataset the way we are estimating it, i.e., based on distance-to-decision-boundary as proposed in the last paragraph of Section 3.2.2. To this end, we first train a three-layer ReLU MLP over the dataset with $\ell_{\log}$ without any consideration of the label noise and compute the transition rates based on the distances. We then flip the labels with these rates and train MLPs with $\ell_{\log}$ (named *Normal*) and $\widetilde{\ell}_{\log}^R$ (named *NDX* for "noise depending only on X") for comparison.

We pick four binary classification sub-datasets of CIFAR-10 randomly. Means of the test set accuracy over three trials are plotted in Figure 1 against the noise rates ranging from 0% to 45%. Across all four datasets, while the network trained with $\ell_{\log}$ experiences a dramatic decrease in performance as the noise rate goes up (so much to go below 50% at times, i.e., worse than random guessing for a balanced dataset such as the ones used here), the one with $\widetilde{\ell}_{\log}^R$ is much more stable, better at each rate and manages to learn from the signal even when the flip rate is close to 50%, which underlines the claimed generalization ability when the noise process is well-modeled.

Furthermore, the leftmost points of subfigures in Figure 1 correspond to 0% noise rate experiments on CIFAR-10. We observe that our model's performance is not unnecessarily degraded and practically the same as the Normal model's on all four subdatasets.

## A.3 HYPERPARAMETER SENSITIVITY

The proposed method has two hyperparameters to tune: $\lambda > 0$ the regularization coefficient that helps approximate the original division in the loss function with repeated subtraction:

$$\widetilde{\ell}^R = \widetilde{\ell}_{\text{numerator}} - \lambda \widetilde{\ell}_{\text{denominator}},$$

and $\beta > 0$ the logistic function's scale parameter, which function we use to turn distances-to-decision-boundary into (pseudo-)probabilities $\rho_{\boldsymbol{x}} = \sigma(-\beta|h(\boldsymbol{x})|)$, where $h : \mathcal{X} \to \mathbb{R}$ is the learning machine. To check the sensitivity of the method with respect to the hyperparameters, we present the frequency table of the selected $\lambda - \beta$ combinations over the 5 sub-datasets of the CIFAR-10 dataset and the Speakers dataset (results of which were both presented in Section 4.2). Note that there

| $\lambda$ | $\beta$ | Frequency |
|------|------|-----------|
| 0.01 | 20.0 | 10 |
| 0.50 | 20.0 | 3 |
| 0.50 | 7.0 | 3 |

Table 5: Frequency of selected hyperparameters of NDX over sub-datasets of CIFAR-10 over 30 sub-experiments.

| $\lambda$ | $\beta$ | Frequency |
|-------|------|-----------|
| 10.00 | 1.0 | 5 |
| 1.00 | 1.0 | 4 |
| 0.01 | 1.0 | 3 |
| 0.01 | 7.0 | 3 |

Table 6: Frequency of selected hyperparameters of NDX over sub-datasets of the Speakers dataset over 30 sub-experiments.

were 30 sub-experiments for each dataset: 5 binary sub-datasets $\times$ 2 noise levels $\times$ 3 trials, and the validation set is 10% of the noisy training set as mentioned in Section 4. Tables 5 and 6 show that 3 and 4 $\lambda - \beta$ combinations, respectively, already account for 50% of the experiments done, i.e., the distribution to the chosen $\lambda - \beta$ values are far from uniform and instead cluster around 2-3 unique values of the parameters, showing the rather insensitive nature of the method with respect to its hyperparameters. Note that the need for validation is not relinquished as the chosen values are different across the image and audio datasets.

## A.4 Disjoint Learning of $\mathbb{P}(Y \mid \boldsymbol{x})$ and $\mathbb{P}(\widetilde{Y} \mid \boldsymbol{x})$

Here, we compare two setups for estimating $\mathbb{P}(Y \mid X)$ and $\mathbb{P}(\widetilde{Y} \mid X)$:

1. The disjoint estimation where we first train a network with $\ell_{\log}$ (and no label noise specific adjustment) to gather $\mathbb{P}(\widetilde{Y} \mid X)$ values per instance, and then train a different network with $\widehat{\ell}_{\log}^{R}$ while using the frozen $\mathbb{P}(\widetilde{Y} \mid X)$ values in the loss correction.

2. Joint training where both $\mathbb{P}(Y \mid X)$ and $\mathbb{P}(\widetilde{Y} \mid X)$ are approximated the way described in Section 3.2.1.

| Subset | Noise Level | Disjoint | Joint |
|--------|-------------|----------|-------|
| 0v4 | 20% | 79.08 | **92.34** |
|     | 40% | 64.42 | **68.25** |
| 2v7 | 20% | 68.38 | **79.90** |
|     | 40% | 60.11 | **62.87** |
| 5v6 | 20% | 67.11 | **73.09** |
|     | 40% | 61.30 | **61.79** |

Table 7: Test accuracy (%) comparisons on three different sub-datasets of Speakers dataset with varying noise levels.

We perform the experiment on three random sub-datasets of the Speakers dataset of Section 4.3; the experimental setup is the same as described therein. As shown in Table 7, somewhat surprisingly, the disjoint training performs considerably worse than the proposed approximation. It also takes more time to train while being memory heavy as it requires saving $N$ floating point numbers into memory. We attribute the empirical success to the warm-up period we employ before the loss correction kicks in, during which the "clean" $\widetilde{Y}$s dominate and later, the correction refrains the learner from overfitting to the noisy ones such that two disjoint stages are blended into one in that sense. We thereby have an empirical support for our way of modeling clean and noisy label probabilities.

### A.4.1 On Using the Same Approximation for Both $\mathbb{P}(Y \mid \boldsymbol{x})$ and $\mathbb{P}(\widetilde{Y} \mid \boldsymbol{x})$

We first note that $\mathbb{P}(Y \mid \boldsymbol{x})$'s estimation in statistically consistent (i.e., training as if with the clean labels as the sample size grows) or probabilistic models in general requires design choices. We exemplify from the literature as follows.

- "Learning from Massive Noisy Labeled Data for Image Classification" (Xiao et al., 2015): they concurrently fit two models to model $Y$ and $Z$ (the label noise *kind* latent variable)

and maximize the likelihood of $\widetilde{Y}$ with Expectation-Maximization. That is, they exploit the factorization (of $\widetilde{Y}$ over $Y$) the class-only dependent label noise model allows, and maximize the (incomplete) likelihood. However, the resulting framework has 3 models in it (one for $Y$, one for $Z$ and one for the transition probabilities) and the optimization via EM gets complex (e.g., requires careful initialization) and requires an identified set of clean labels.

- "Learning with Noisy Labels" (Natarajan et al., 2013): when they develop their "Method of Unbiased Estimators", they require $\mathbb{E}[\ell(\widetilde{Y}, f(\boldsymbol{x}))] = \ell(Y, f(\boldsymbol{x}))$ for all $Y$, $f(\cdot)$ values. Please note that there is no expectation on the right hand side. Therefore $\mathbb{P}(Y \mid \boldsymbol{x})$ is nonexistent in their formulation, making it not achieve risk equivalence (we also point to this fact in the manuscript (end of related work)).

- "Making Deep Neural Networks Robust to Label Noise: a Loss Correction Approach" (Patrini et al., 2017): This is a multi-class generalization of the above work of Natarajan et al. (2013); this time we see the expectation on the right hand side. Building on the factorization $\mathbb{P}(Y \mid \boldsymbol{x}) = \mathbb{P}(Y \mid \widetilde{Y}, \boldsymbol{x})\mathbb{P}(\widetilde{Y} \mid \boldsymbol{x})$ and their assumption of y-only dependent label noise, they aim to estimate $\mathbb{P}(Y \mid \widetilde{Y})$, i.e., a $K \times K$ transition matrix ($K$: number of classes). Their (one) design choice is to train a separate neural network to model this matrix by assuming the existence of perfect samples (i.e., those having almost surely clean labels). Therefore, they first learn a transition matrix, freeze it and then use it in the second phase of learning a new machine on $\widetilde{Y}$s to uncover $\mathbb{P}(Y \mid \boldsymbol{x})$. This modeling of the transition matrix turned out to perform really well, as Forward & Backward loss correction methods from this paper is still of high relevance in label noise research (theory- and performance-wise).

Our instance-dependent label noise model, while more natural/powerful (Chen et al., 2021), does not lend itself to the factorization above because in $\rho_{\boldsymbol{x}} = \mathbb{P}(\widetilde{Y} \neq Y \mid \boldsymbol{x})$, $Y$ and $\widetilde{Y}$ are on the left side of the condition together. Nevertheless, we could still learn the "annotator's brain" in a separate machine, i.e., model $\mathbb{P}(\widetilde{Y} \mid \boldsymbol{x})$, then freeze its in-sample predictions. Then, in a second phase, while training an $h$ with $\tilde{\ell}$ (on the same dataset still), use the frozen predictions of the former machine as a proxy for $\mathbb{P}(\widetilde{Y} \mid X)$, and the current output of $h$ as a proxy for $\mathbb{P}(Y \mid \boldsymbol{x})$ (since with $\tilde{\ell}$, this phase's machine $h$ is expected to uncover $\mathbb{P}(Y \mid \boldsymbol{x})$ by Proposition 2). In the previous section (Appendix A.4.1), we present a detailed empirical comparison of this disjoint approach with what we instead do – the difference in performance (and naturally also the computation time) was significantly worse in this disjoint way of modeling those probabilities, providing an empirical evidence for our design choice.

While that empirical evidence suggests one can do better than disjoint modeling, we directly justify our assumption as follows. As noted above, in Patrini et al.'s loss correction design (Patrini et al., 2017), they assume the existence of perfectly clean examples, on/with which they first train a network to model $\mathbb{P}(Y \mid \widetilde{Y})$ (and similarly, also in Xiao et al. (2015)). In our setup (or in any noisy label learning setup for that matter), while the *identification* of the clean labels is not assumed, their existence is, e.g., our $\mathbb{E}[\rho_{\boldsymbol{x}}] < 0.5$ assumption. In fact, in a $K$-class scenario, one needs at least $1/K$ clean samples to exist so that learning is even possible (Menon et al., 2018). (Actually, training is possible either way, e.g., as an extreme case of 100% label noise in binary classification, a (good enough) model's predictions on the unseen data will always be worse than random *until* they are flipped, at which point the semantic labels are matched. So the assumption of high signal-to-noise ratio is to preserve consistency of the semantic meaning of the labels between training and testing sets.) What's more, it has been demonstrated by Han et al. (2025) that the neural network first "focuses" on these clean samples in the early stages of the training. But this means we can elevate the disjoint modeling idea of Patrini et al. (2017) of $\mathbb{P}(Y \mid \boldsymbol{x})$ and $\mathbb{P}(\widetilde{Y} \mid \boldsymbol{x})$ with the dominance of clean samples in early stages of training to make the modeling joint: a number of warm-up epochs at start with the normal (uncorrected) loss. Once the "groundwork" of establishing a decision boundary by the machine in the warm-up period is done with the usual loss function through the dominant clean samples, our loss correction mechanism kicks in to make the machine more aware of the pitfalls due to label noise (this "awareness" is examined mathematically on several edge cases in Section 3.2.3, where we present similarities and differences between $\tilde{\ell}$ and $\ell$).

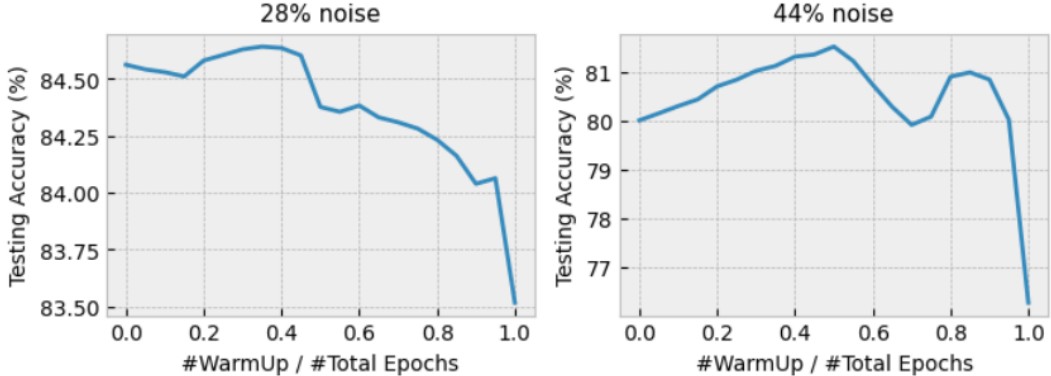

Figure 2: Observing (approximately) U-shapes with rather flat maxima while sweeping the rate of number of warm-up epochs from 0 to 1, against test accuracy. This supports the need for the warm-up period as well as the (in)sensitivity against it. Note that the rightmost points represent full warm-up without any loss correction, i.e., the Normal model.

## A.5 EFFECT OF THE NUMBER OF WARM-UP PERIODS

Here, we perform an experiment on the effect of the number of warm-up epochs to model's generalization ability by sweeping it to see the need for it as well as the sensitivity against it. We sweep the ratio "#Warm-Up Epochs / #Total Epochs" from 0 to 1, both ends inclusive. Ideally, we expect a reverse U-shape with a rather flat maxima w.r.t. test accuracy to represent:

1. The trade-off: when the ratio is 1, model reduces to the "Normal" model, i.e., does nothing special about the label noise and has an inferior performance than the other rates; when the ratio is 0, dominance of the clean labels is not sufficiently utilized to form a reasonable baseline to correct;

2. The insensitivity against the number of warm-up epochs in the middle region as a rather flat maxima.

We use the Tabular Adult dataset (Becker & Kohavi, 1996) with 28% and 44% label noise as in Section 4.3. We train and validate our model as done in the usual experiments while sweeping the warm-up rate. We plot the warm-up rate versus test accuracy with two different noise rates in Figure 2. We roughly observe the mentioned trade-off and insensitivity over the number of warm-up epochs (while not being perfect U-shapes), supporting the warm-up strategy. The insensitivity also allows for not treating the number of warm-up epochs as a hyperparameter that needs heavy tuning (also the case in, e.g., Li et al. (2020), Yang et al. (2022) and Zhang et al. (2021)).

## A.6 BOUNDING $\widetilde{\ell}$

Even though there is nothing mathematically restraining $\widetilde{\ell}$ from going to infinity, it can still be a bounded loss function provided that the base loss $\ell$ is also bounded and $\rho_{\boldsymbol{x}}$ satisfies a certain condition. For example, if, for a given $h \in \mathcal{H}$ and a finite $M$,

$$\rho_{\boldsymbol{x}} \leq \sigma(h(\boldsymbol{x}))\sigma(-h(\boldsymbol{x})) \cdot \min \left\{ 1, \right.$$

$$\frac{\ell(h(\boldsymbol{x}), \widetilde{y})}{\sigma(h(\boldsymbol{x}))\ell(h(\boldsymbol{x}), \widetilde{y}) + \sigma(-h(\boldsymbol{x}))\ell(h(\boldsymbol{x}), -\widetilde{y})}, \tag{1}$$

$$\left. \frac{\ell(h(\boldsymbol{x}), \widetilde{y}) - M}{\sigma(h(\boldsymbol{x}))\ell(h(\boldsymbol{x}), \widetilde{y}) + \sigma(-h(\boldsymbol{x}))\ell(h(\boldsymbol{x}), -\widetilde{y}) - M} \right\} \forall \boldsymbol{x}, \widetilde{y},$$

then $|\widetilde{\ell}(\cdot, \cdot)| \leq \max\{M, A\} =: \widetilde{\ell}_{\infty}$. An exemplary $\rho_{\boldsymbol{x}}$ would be $\max(0, \sigma(h(\boldsymbol{x}))\sigma(-h(\boldsymbol{x})) - \varepsilon)$ for small $\varepsilon > 0$ with $\widetilde{\ell}_{\infty} = 50/40$ and $M = 1.5 = \widetilde{\ell}_{\infty}$, i.e., there exists a $\rho_{\boldsymbol{x}}$ configuration that preserves boundedness of $\ell$.

Note that the threshold on the original unbounded loss function for a finite upper bound cannot be chosen arbitrarily small, as the gradient will vanish for even small activation values close to 0, severely hindering learning. For example, while the raw input in absolute value to the logistic loss barely crosses 10 in practice (making it a candidate upper bound), it would be too stringent to, say, clamp the loss at 1.5 only to get a lower gap on the RHS of the bound. Furthermore, while scaling down any loss function has no effect for the optimization problem (all other things being equal), again, in practice, heavy downscaling to get a lower gap is a matter of trade-off since i) numerical issues may arise (e.g., underflow) and ii) it requires extra care in tuning the optimization related hyperparameters, e.g., the learning rate in case of gradient-based methods. More importantly, though, the risk values themselves in the bound (equation 6) are scaled down since they involve averaging the losses suffered, so the interpretation of the bound stays the same up to a constant factor.

### A.7 TEST SET DISTRIBUTION OF THE DATASETS

Since we report accuracy scores on the (clean) test sets, here we present the ratios of the positive (latter) class in each (sub-)dataset used in the experiments:

| Subset | 0v1 | 2v3 | 4v5 | 6v7 | 8v9 |
|---|---|---|---|---|---|
| CIFAR-10 | 50% | 50% | 50% | 50% | 50% |
| Speakers | 43% | 56% | 55% | 49% | 48% |

| | Adult | Diabetes | Heart | Segmentation | Splice |
|---|---|---|---|---|---|
| Tabular | 24% | 35% | 49% | 58% | 46% |

| Subset | 6v8 | 6v7 | 6v9 | 1v6 | 2v6 | 0v2 | 2v9 | 2v11 | 1v7 | 0v11 |
|---|---|---|---|---|---|---|---|---|---|---|
| Clothing1M | 48% | 50% | 52% | 48% | 48% | 50% | 51% | 51% | 48% | 51% |

Table 8: Ratios of the positive class in the testing sets of the datasets.

We observe that the test sets are mostly balanced with the exception of *Adult* and *Diabetes* tabular datasets, for which a baseline (a model predicting 0 regardless of the instance) would achieve 76% and 65% testing accuracy, respectively. We see from the results in Section 4.3 that the best performing models considerably pass these thresholds (e.g., 84.53% and 73.34%, respectively (under 28% noise)).

### A.8 DETAILS OF THE SETUP OF THE EXPERIMENTS

**Datasets.**

- CIFAR-10 (Krizhevsky, 2009): The well-known 10-class classification of 32x32x3 RGB images; 50,000 training and 10,000 test samples.
- Speakers (Rimi, 2023): This is a times series-based dataset aimed for 11-class classification of YouTube clips of famous motivational speakers. Each clip is a five second signal and the corresponding label is the name of the speaker. 6,204 training and 1,551 test samples.
- Tabular datasets.
  1. *Adult* (Becker & Kohavi, 1996): predict whether the income of an individual exceeds a certain threshold; 48,842 instances (11,687 positive), 14 features.
  2. *Diabetes* (Bennett et al., 1971): predict whether a given patient has diabetes; 768 instances (268 positive), 8 features.
  3. *Heart* (Janosi et al., 1989): predict whether a given patient has a heart disease; 303 instances (165 positive), 13 features.
  4. *Splice* (Towell et al., 1991): predict whether a given boundary at the splice junction of a DNA is an acceptor or a donor; 3,175 instances (1,527 positive), 60 features.

5. *Segmentation* (Brodley, 1990): assign a group number to hand-segmented 3x3 areas of various outdoor images; 2,310 instances (1,320 positive), 18 features.

- Clothing1M (Xiao et al., 2015): This dataset is a large-scale real-world dataset of cloth images that is naturally noisy. It provides $\sim$1,000,000 (noisy) training pairs and $\sim$10,000 (clean) testing pairs. There are 14 classes and the images are 64x64x3 RGB. We note that we do not use the clean training and validation sets the dataset also provides.

**Machines.** We use neural networks and LightGBM as the underlying machines in the experiments.

- MLP: a three-layer feed-forward neural network with ReLU as the hidden activations. Number of hidden neurons are 128, 64 and 32 towards the (single) output. Output activation is identity.
- CNN (for CIFAR-10): a two convolution-ReLU-maxpool layers (kernel sizes 6x6 and 16x16, respectively with a stride of 5 and no padding for both layers) followed by a 128-64-32 fully connected layers towards the (single) output. Output activation is identity.
- CNN (for Clothing1M): a four convolution-ReLU-maxpool layers (kernel sizes 32x32, 32x32, 64x64 and 64x64 respectively with a stride of 1 and no padding for all layers) followed by a 128-sized fully connected layer towards the (single) output. Output activation is identity.
- LightGBM: default parameters are used except for the number of trees and the learning rate, which are tuned.

**Preprocessing.** All datasets undergo standardization (subtract the mean, divide by the standard deviation; both statistics are obtained from the training split) after the following specific preprocessing steps are applied:

- CIFAR-10: random crops and random horizontal flips.
- Speakers: Mel-frequency cepstrum with 128 mel bands such that inputs are akin to a 2D image of shape 128 x 157.
- Tabular datasets: For the *Adult* dataset, categorical features are one-hot encoded yielding 100 features in effect.

**Optimization.** For all experiment runs, 20 epochs of SGD with a batch size of 128 (32 for Speakers) is used (first 4 epochs are for warm-up in NDX). 10% of the training set is spared for the validation of hyperparameters with grid search: learning rate, regularization coefficient ($\lambda$) and sigmoid scale ($\beta$) for NDX; learning rate for other models. Noncritical hyperparameters, if any, are taken from the respective papers/source code as is, against which most of the methods assert to be rather insensitive anyway (e.g., "iteration_nmf" of PTD (Xia et al., 2020) is taken to be 20; "forget_rate" of Coteaching+ (Yu et al., 2019) (which is actually not mentioned in the paper) is set to 0.2). Validation is done for more prominent hyperparameters, e.g., $\rho_+$ and $\rho_-$ of UB (Natarajan et al., 2013) are searched in $[0.1, 0.2, 0.3, 0.4]$; $\alpha$ of Peer (Liu & Guo, 2020) in $[0.1, 1., 5.]$ and $\alpha$ and $\beta$ of APL (Ma et al., 2020) in $[0.1, 1., 10.]$. The center learning rate (call $\eta$) is 0.01 for all experiments; $\eta/100, \eta/10, 10\eta, 100\eta$ is the learning rate search space. For NDX, $\lambda \in (0.01, 0.10, 0.50, 1, 10)$ and $\beta \in (1, 3, 7, 20, 50)$ are searched. As an exception, Adam optimizer is used in Coteaching+'s training as advised in its paper (Yu et al., 2019). For LightGBM, number of trees are searched in $(10, 25, 50, 100, 150)$ and the learning rate in $(0.01, 0.05, 0.10, 0.50, 1)$.

A.9    DETAILS OF THE METHODS FROM THE LITERATURE USED IN THE EXPERIMENTS

Here we give brief details on the methods used in the Experiments in Section 4.2 for comparison over the real-life datasets.

*BCN* (Du & Cai, 2015) is a modified logistic regression model with boundary-consistent instance-only dependent noise model. *UB* is the "method of unbiased estimators" of Natarajan et al. (2013) with a CCN-based noise model. *DMI* (Xu et al., 2019) is a CCN-based loss correction method from an information theoretic view that involves the determinant of the transition matrix. *Peer* (Liu & Guo, 2020) represents the peer loss that achieves risk equivalence up to a constant involving noise

rates. *APL* (Ma et al., 2020) introduced a normalization trick to make any loss function noise tolerant and uses combinations of un/normalized (active/passive) loss functions for better generalization. *PTD* (Xia et al., 2020) is based on part-dependent ILDN modeling of the noise which, for each part of a given instance, first learns representations and then a transition matrix by utilizing almost surely non-noisy samples. *BLTM* (Yang et al., 2022) is also an ILDN-based model which brings in the Bayes-optimal labels through which a transition matrix is learned and a revision is lastly made as in PTD. *Coteaching+* (Yu et al., 2019) is an improvement over the Coteaching model (Han et al., 2018) that uses two neural networks in parallel and proposes a small-loss and disagreement based cross update rule to deal with the noise. *Forward* and *Backward* (Patrini et al., 2017) are CCN-based loss correction mechanisms where first a normal network is trained to estimate the transition matrix and it is then fed to another one to correct predictions or the loss itself, respectively. *GCE* is the generalized cross entropy loss function proposed by Zhang & Sabuncu (2018), which uses a Box-Cox-like transformation to correct the logistic loss. *PLC* (Zhang et al., 2021) progressively corrects suspicious data pairs with an ILDN-based noise model. Lastly, we note that we also experimented with DivideMix (Li et al., 2020), a neural network machinery with distillation of small-loss examples via a Gaussian mixture model; however, it did not attain a reasonable score with validation (considerably above 50% test accuracy) for any of the binary classification tasks presented in Section 4, and therefore is excluded from the results.

## A.10 CODE FOR THE LOSS CORRECTION

As mentioned in Section 1, loss correction methods are generally very easy to implement. Our method with probabilities calculated through distances to the decision boundary requires little extra computation thanks to the observation mentioned at the end of Section 3.2.2, i.e., distances are approximated with $|h(\boldsymbol{x})|$ where $h(\boldsymbol{x})$ is already calculated in the training loop as an input gets feed-forwarded. Here, we share a sample implementation of this loss correction mechanism in code using PyTorch (Paszke et al., 2019) where we correct $\ell_{\log}$:

```python
import torch
from torch import exp, log, sigmoid

def loss_fun(self, X, y):
    """
    Given (batch) pairs X and y (noisy), return ˜l^R
    """
    ## Feed-forward
    h_x = self(X).squeeze(-1)
    f_x = sigmoid(h_x)

    ## Calculate distance-based probabilities of label flips
    # 'self.sigmoid_scale' is \beta
    distances = h_x.abs()
    rho_x     = 1 / (1 + exp(self.sigmoid_scale * distances))

    ## Compute the corrected loss
    # Base loss (\ell; logistic) w.r.t. ˜y and -˜y
    normal_loss   = torch.where(y == +1, -log(f_x), -log(1 - f_x))
    opposite_loss = torch.where(y == +1, -log(1 - f_x), -log(f_x))

    # Approximate P(Y | X) and P(˜Y | X), form the modified loss
    pyy_x       = torch.where(y == +1, f_x, 1 - f_x)
    numerator   = ((1 - pyy_x - rho_x) * pyy_x * normal_loss
                - rho_x * (1 - pyy_x) * opposite_loss)
    denominator = pyy_x * (1 - pyy_x) - rho_x

    # 'self.regularization_scale' is \lambda
    loss = numerator - self.regularization_scale * denominator

    return loss.mean()
```

