# OpenReview forum: "Resurfacing the Instance-only Dependent Label Noise Model through Loss Correction"
_ICLR.cc/2026/Conference — ICLR 2026 Poster_

### Official Review · Reviewer_Hbdw · 2025-10-30

**Soundness:** 3
**Presentation:** 3
**Contribution:** 3
**Rating:** 6
**Confidence:** 4

**Summary:**

This paper looks at label noise in binary classification. It suggests a new way to fix errors using an instance-aware loss correction scheme. This is based on the instance-only dependent (IDN) noise model. The main idea is to model the chance of a label being wrong as $\rho_x = \mathbb{P}(Y \neq \tilde{Y} | X = x)$. This depends only on the instance $X$, not the true label $Y$. This is different from the instance-label dependent noise model (ILDN). The authors create a new loss function $\tilde{\ell}$ that matches the risk of noisy data with clean data, as long as the base loss is classification-calibrated, like cross-entropy. They estimate the transition rates $\rho_x$ using the distance to the decision boundary to show how hard an instance is. A regularized version $\tilde{\ell}_R$ is also introduced to avoid numerical problems, with theoretical support using Rademacher complexity bounds. Main contributions: 1. A loss correction method that works well with instance-dependent noise 2. Highlighting the efficient IDN model (one value per instance instead of a matrix in ILDN) 3. Different ways to estimate $\rho_x$, like distance-based, clustering, and ensemble methods 4. Testing in different areas (image, audio, tabular) and with different learners (CNNs, MLPs, GBMs) The tests show that this method works as well as or better than 12 other methods on CIFAR-10, and 5 tabular datasets with moderate (28%) and high (44%) noise levels.

**Strengths:**

1. The paper connects theory and practice by creating a loss correction method (Proposition 1) and a practical version with guarantees (Proposition 2), even with some limits.



2. Estimating one flip probability per instance instead of a full matrix is a big practical benefit. The paper argues well for when this simpler model works, challenging the trend of more complex ILDN methods.



3. Using $|h(x)|$ as a stand-in for distance to decision boundary is smart and costs nothing extra (already done in forward pass). Though not fully explored theoretically, tests (like the matched-noise experiment in Fig. 1) show it works well.



4. Applied to CNNs, MLPs, and gradient-boosted trees - Tested in different areas (vision, audio, tabular) - Consistently good performance (often top-2) across 10 datasets and different noise levels - Stability across datasets (noted in Section 4.2) is very useful for users



5. Two noise levels (28%, 44%) for moderate and high noise - Three trials per setup - Compared with 12 baseline methods with different noise assumptions - Study (Appendix A.3) explains joint vs. separate probability estimation - Analysis (Section 3.2.3) gives insight into the changed loss behavior - Code (Appendix A.8) shows it's easy to use (~20 lines) - Hyperparameter analysis (Appendix A.2) shows it's robust

**Weaknesses:**

1. The paper talks about using the same function $h(x)$ to estimate both $\mathbb{P}(Y|x)$ and $\mathbb{P}(\tilde{Y}|x)$ (see Section 3.2.1). This is based on warm-up epochs, not detailed analysis. When is this estimation accurate? How much does the performance depend on the length of the warm-up? The paper says 4 epochs but does not test this. - The idea that $|h(x)|$ shows the true distance to a boundary is not backed by theory. It works in practice, but we don't know when it is accurate. For very complex boundaries or early in training, it might not work.

2. Proposition 2's conditions on $\lambda$ depend on dataset-specific factors like $\Delta_S$ and norms, which are hard to check. The paper uses grid search but doesn't explain how to choose search ranges or give any tips. - There is no advice on which $\phi$ (probability transformation) or $z$ (difficulty metric) to use for new areas. The paper shows that distance-based + $\beta$-logistic works well but doesn't explain when other methods (like clustering or autoencoders) might be better. - The warm-up period of 4 epochs seems random, with no explanation or analysis of its impact.



3. The main experiments (Section 4.2) use ILDN noise injection, not IDN noise, as per Xia et al. 2020. This checks how well the model handles mismatches but doesn't prove that real-world noise is like IDN. - The matched-noise experiment (Section 4.1) is artificial and repetitive—noise is added and then removed using the same model. - There is no real-world data study to see if IDN is enough compared to ILDN. Are there areas where ignoring label dependence affects performance? - The statement that IDN is "not less powerful than ILDN" (Abstract) lacks strong support. Tables 1-3 show mixed results when compared to ILDN methods (PTD, BLTM).



4. Authors recognize this issue, but it limits how useful the method is. There is no plan for extending it to multiple classes, even though it seems like a logical next step. - Small dataset size: Using parts of CIFAR-10 and small tables doesn't show if it works on large datasets. How does it handle big data like ImageNet or millions of examples? - Fairness issues with baseline methods - Some methods for multiple classes (GCE, DMI) might not work well - DivideMix was left out because it didn't perform well, but it's unclear why (was it a problem with how it was done or a limit of the method for two-class tasks?) - Using different optimizers for some methods (like Adam for Coteaching+) makes it hard to compare them.

5. There is no explanation of when $\mathbb{E}_X[\rho_x] &lt; 0.5$ is true. There is no talk about how complex the calculations are or how long they take. There is little error analysis: what if $\rho_x$ is estimated wrong? Sensitivity analysis is only done for hyperparameters $\lambda, \beta$, not for the noise model itself. There is no study of when the method fails or does worse than other methods.

6. Equation 4 is hard to understand and needs more steps to explain it. The main difference (IDN vs ILDN) is not highlighted early enough. Some important details (like warm-up and approximations) are mentioned briefly without explanation.

**Questions:**

1. (Section 3.2.1): - You mention that $h(x)$ models both $\mathbb{P}(Y|x)$ and $\mathbb{P}(\tilde{Y}|x)$. How can one function represent both clean and noisy label distributions? - Can you give a theoretical or practical analysis of when this approximation works? - What if you use the separate model for the whole training, not just the warm-up? Appendix A.3 shows worse results, but the separate model used a fixed $\mathbb{P}(\tilde{Y}|x)$—what if both networks are updated together?



2. Can you explain when $|h(x)|$ is a good way to measure distance to the decision boundary? - In neural networks with ReLU activations, the decision boundary can be made of straight-line segments. Does your method still work well in areas far from the training data? - Have you thought about using real distance calculations (like DeepFool) for some samples to check how good the approximation is during training?



3. Why is the 4-epoch warm-up chosen? Does it affect performance? - Can you show results for 0, 2, 4, 8, and 16 warm-up epochs? - Is there a method to decide the warm-up length, like using the clean sample ratio or when $h$ converges?



4. Your main experiments use ILDN noise injection. Can you also test with true IDN noise to check the model's assumptions? - When would IDN not be enough, and ILDN be needed? Can you describe the task or data features that decide this? - Tables 1-3 show PTD and BLTM (ILDN methods) sometimes do better than NDX. Does this mean ILDN is better for these cases?



5. The conditions in Proposition 2 are hard to check in real situations. Can you give practical tips or simple methods for setting $\lambda$ other than grid search? - How does the best $\lambda$ depend on dataset features like size, noise, and dimension? - Have you seen cases where the denominator in Eq. 3 gets close to zero during training even with regularization?

6. How does the method work with larger datasets like full ImageNet and more complex problems? - Can you explain how to extend it to multi-class classification? Is it as simple as one-vs-rest, or are there major challenges? - For the tabular LightGBM results, how does the method work with GBM's way of handling noisy data?

7. Are there times when NDX does much worse than the usual methods? What are these situations like? - What happens to performance when the noise model is very wrong (for example, the real noise depends on the class, but you think it is IDN)?

8. In Section 3.2.3, you show that $\tilde{\ell}(\cdot, \tilde{y})$ becomes $\ell(\cdot, -\tilde{y})$ when $\tilde{y} \cdot h(x)$ goes to negative infinity. This means the model trusts itself more than the label. Could this cause the model to repeat its mistakes? How does this compare to methods that reduce the importance of doubtful samples?

---

> ### Author Response · Authors · 2025-11-20
> **Response to Reviewer Hbdw (1/12)**
>
> We thank you for your time and valuable feedback! Below we present our answers to the weakness points and questions.
>
> ### Weaknesses
> ----------
> **W.1.**
>
> (W.1.1.)
> > The paper talks about using the same function $h(x)$ to estimate both $\mathbb{P}(Y|x)$ and $\mathbb{P}(\tilde{Y}|x)$ (see Section 3.2.1). This is based on warm-up epochs, not detailed analysis. When is this estimation accurate?
>
> We first note that $P(Y \mid X)$'s estimation in statistically consistent (i.e., training as if with the clean labels as the sample size grows) or probabilistic models in general requires design choices. We exemplify from the literature as follows.
>   - "Learning from Massive Noisy Labeled Data for Image Classification" [1]: they concurrently fit two models to model $Y$ and $Z$ (the label noise _kind_ latent variable) and maximize the likelihood of $\widetilde Y$ with Expectation-Maximization. That is, they exploit the factorization (of $\widetilde Y$ over $Y$) the class-only dependent label noise model allows, and maximize the (incomplete) likelihood. However, the resulting framework has 3 models in it (one for $Y$, one for $Z$ and one for the transition probabilities) and the optimization via EM gets complex (e.g., requires careful initialization) and requires an identified set of clean labels.
>   - "Learning with Noisy Labels" [2]: when they develop their "Method of Unbiased Estimators", they require $\mathbb{E}[\ell(\widetilde Y, f(X))] = \ell(Y, f(X))$ for all $Y$, $f(\cdot)$ values. Please note that there is no expectation on the right hand side. Therefore $P(Y \mid X)$ is nonexistent in their formulation, making it not achieve risk equivalence (we also point to this fact in the manuscript (end of related work)).
>   - "Making Deep Neural Networks Robust to Label Noise: a Loss Correction Approach" [3]: This is a multi-class generalization of the above work of [2]; this time we see the expectation on the right hand side. Building on the factorization $P(Y \mid X) = P(Y \mid \widetilde{Y},X)P(\widetilde{Y} \mid X)$ and their assumption of y-only dependent label noise, they aim to estimate $P(Y \mid \widetilde{Y})$, i.e., a $K\times K$ transition matrix ($K$: number of classes). Their (one) design choice is to train a separate neural network to model this matrix by assuming the existence of perfect samples (i.e., those having almost surely clean labels). Therefore, they first learn a transition matrix, freeze it and then use it in the second phase of learning a new machine on $\widetilde{Y}$s to uncover $P(Y \mid X)$. This modeling of the transition matrix turned out to perform really well, as Forward & Backward loss correction methods from this paper is still of high relevance in label noise research (theory- and performance-wise).
>
> Our instance-dependent label noise model, while more natural/powerful [4], does not lend itself to the factorization above because in $\rho_x = P(\widetilde{Y} \ne Y \mid X)$, $Y$ and $\widetilde{Y}$ are on the left side of the condition together. Nevertheless, we could still learn the "annotator's brain" in a separate machine, i.e., model $P(\widetilde{Y} \mid X)$, then freeze its in-sample predictions, and then in a second phase, while training an $h$ with $\tilde\ell$ (on the same dataset still), use the frozen predictions of the former machine as a proxy for $P(\widetilde{Y} \mid X)$, and the current output of $h$ as a proxy for $P(Y \mid X)$, as with $\tilde\ell$, this phase's machine $h$ is expected to uncover $P(Y \mid X)$ by Proposition 1. Actually, and as you noted below, we tried this: in Section 3.2.1, we mention this way of modeling quantities, and in Appendix A.4, we present a detailed empirical comparison of this disjoint approach with what we instead do -- the difference in performance (and naturally also the computation time) was significantly worse in this disjoint way of modeling those probabilities, providing an empirical evidence for our design choice.

---

> ### Author Response · Authors · 2025-11-20
> **Response to Reviewer Hbdw (2/12)**
>
> While that empirical evidence suggests one can do better than disjoint modeling, we directly justify our assumption as follows. As noted above, in Patrini et al.'s loss correction design [3], they assume the existence of perfectly clean examples, on/with which they first train a network to model $p(Y \mid \widetilde{Y})$ (and similarly, also in [1]). In our setup (or in any noisy label learning setup for that matter), while the _identification_ of the clean labels is not assumed, their existence is, e.g., our $\mathbb{E}[\rho_x] < 0.5$ assumption. In fact, in a $K$-class scenario, one needs at least $\dfrac1K$ clean samples to exist so that learning is even possible [5]. (Actually, training is possible either way, e.g., as an extreme case of 100% label noise in binary classification, a (good enough) model's predictions on the unseen data will always be worse than random _until_ they are flipped, at which point the semantic labels are matched. So the assumption of high signal-to-noise ratio is to preserve consistency of the semantic meaning of the labels between training and testing sets.) What's more, it has been demonstrated by [6] that the neural network first "focuses" on these clean samples in the early stages of the training. But this means we can elevate the disjoint modeling idea of [3] of $P(Y \mid X)$ and $P(\widetilde{Y} \mid X)$ with the dominance of clean samples in early stages of training to make the modeling joint: a number of warm-up epochs at start with the normal (uncorrected) loss. Once the "groundwork" of establishing a decision boundary by the machine in the warm-up period is done with the usual loss function through the dominant clean samples, our loss correction mechanism kicks in to make the machine more aware of the pitfalls due to label noise (this "awareness" is examined mathematically on several edge cases in Section 3.2.3, where we present similarities and differences between $\tilde\ell$ and $\ell$).

---

> ### Author Response · Authors · 2025-11-20
> **Response to Reviewer Hbdw (3/12)**
>
> (W.1.2.)
> > How much does the performance depend on the length of the warm-up? The paper says 4 epochs but does not test this.
>
> We first note that using a warm-up period is not uncommon among prominent label noise methods, e.g., DivideMix [7], BLTM [8] and PLC [9]. (In fact, in [9] (Lemma 2), authors theoretically justify the need for such a period early in the training, albeit for their specific noise model.) While these label noise models (and ours) are highly different in how the problem is addressed, the warm-up period is used for the unique aim of "attain[ing] a reasonable network" [9] for the "initial convergence of the algorithm" [7] to, e.g., "collect distilled examples" [8]. In other words, the warm-up period is used without any architectural modifications to learn a base network under the dominance of the clean labels [6] before it starts overfitting the label noise, at which point the proposed mechanisms start taking place. We also note that in some works that do not make use of an explicit warm-up period, the counterpart assumption of the identification of almost-surely clean samples (i.e., the anchor or anchor-like points) is in place, e.g., [1], [3]. Furthermore, as empirically shown across a wide spectrum of dataset-noise rate-machine settings, i.e., (i) sanity check for the noise model in Figure 1; (ii) CIFAR-10 dataset's results in Table 1; (iii) Speakers dataset's results in Table 2; and (iv) tabular datasets' results in Table 3, our model manages to achieve comparable or (mostly) better test accuracy values.
>
> Prompted by your comment (and also that of Reviewer Hbdw), we performed an experiment on the effect of the number of warm-up epochs by sweeping it to see the need for it as well as the sensitivity against it. We sweep the ratio "#Warm-Up Epochs / #Total Epochs" from 0 to 1. Ideally, we expect a reverse U-shape with a rather flat maxima w.r.t. test accuracy to represent (i) the trade-off: when the ratio is 1, model reduces to the "Normal" model, i.e., does nothing special about the label noise and has an inferior performance than the other rates; when the ratio is 0, dominance of the clean labels is not sufficiently utilized to form a reasonable baseline to correct; and (ii) the insensitivity against the number of warm-up epochs in the middle region as a rather flat maxima.
>
> We used the Tabular Adult dataset with 28% and 44% label noise as in Section 4. We train and validate our model as done in the usual experiments while sweeping the warm-up rate. We plotted the warm-up rate versus test accuracy with two different noise rates and the corresponding figure is placed in Appendix A.5 (here is [anonymous link](https://imgshare.cc/1lkkj020) to an online image server for your convenience). We roughly observe the mentioned trade-off and insensitivity over the number of warm-up epochs (while not being perfect U-shapes), supporting the warm-up strategy. The insensitivity also allows for not treating the number of warm-up epochs as a hyperparameter that needs heavy tuning (also the case in [7] - [9]).
>
> (W.1.3.)
> > The idea that $|h(x)|$ shows the true distance to a boundary is not backed by theory. It works in practice, but we don't know when it is accurate. For very complex boundaries or early in training, it might not work.
>
> If the underlying classifier is linear, $|h(x)|$ is the exact distance to the decision boundary (upto a scaling with the norm of $h$). As discussed at the end of Section 3.2.2, however, this is no longer an exact computation for nonlinear machine, unless it's a (near-)perfect classifier (i.e., near 100% training accuracy), in which case it again becomes exact as interpolation implies the last layer is solving a linearly separable problem in the embedded dimension. In fact, [10] showed that the optimization problem in such a network coincides with that of SVMs. In our experiments with different models (ReLU MLP, CNN, LightGBM), we found it working well as an approximation, as you noted. The theory from [10] further suggests that in today's overparameterized, interpolating neural networks (i.e., those with 100% training accuracy), the approximation becomes exact again (upto monotonicity since it is the distance of the embedded instances found in the last layer). Therefore, we argue that even in highly complex boundaries, the approximation has empirical and theoretical support.

---

> ### Author Response · Authors · 2025-11-20
> **Response to Reviewer Hbdw (4/12)**
>
> **W.2.**
>
> (W.2.1.)
> > Proposition 2's conditions on $\lambda$ depend on dataset-specific factors like $\Delta_S$ and norms, which are hard to check. The paper uses grid search but doesn't explain how to choose search ranges or give any tips.
>
> As we noted in the manuscript, the practical reflection of Proposition 2 is rather limited. Its main goal is to show that even if we transform the loss functional's division into a repeated subtraction with $\lambda$, the consistency between learning from noisy and clean domains remains intact under sufficient conditions on $\lambda$. In fact, these conditions are only checkable upon training an $h$ on a finite sample (and we indeed checked for them for some of the experiments). Therefore, the conditions do not help much in guiding the pool for the hyperparameter $\lambda$. What we observed in Appendix A.3., however, the model is rather insensitive to the choice of $\lambda$, relieving the user from heavy tuning. What remains, however, is choosing a specific search range as you noted, which is a shared problem for all label noise models (or any learning model with hyperparameters) to our knowledge.
>
> (W.2.2.)
> > There is no advice on which $\phi$ (probability transformation) or $z$ (difficulty metric) to use for new areas. The paper shows that distance-based + $\beta$-logistic works well but doesn't explain when other methods (like clustering or autoencoders) might be better.
>
> Estimation of $\rho_x$ is indeed a design choice; while we argue that it brings flexibility to the overall modeling process (as opposed to, say, an ILDN model where the transition matrix is estimated with a very specific way, e.g., by focusing on different parts of a given image in [11] or three stage training using distillation of the Bayes-optimal labels as in [8]), we also acknowledge that this freedom needs to be guided with sound design choices. The box we present in Section 3.2.2. offers several ways to model this quantity; but it is in no way an exhaustive list (as noted) and it is therefore hard to compare this large of a pool of choices.
>
> That being said, we would like to offer empirical and theoretical support for our choice in the paper for the experiments, i.e., distance-to-decision-boundary based modeling of $\rho_x = \sigma\left(-\beta |h(x)|\right)$.
>
> On the empirical side:
>
> - It is arguably a very intuitive one: an instance closer to the decision boundary tends to be more "ambiguous" as it carries features of more than one class, making it a plausible candidate for mislabeling. Moreover, historically, it found applications in [12] and [13] (albeit for linear machines).
>
> - It has a very low computational cost: $h(x)$ is already computed in a forward pass of the machine.
>
> - Across a wide spectrum of dataset-noise rate-machine experiment settings, i.e., (i) sanity check for the noise model in Figure 1; (ii) CIFAR-10 dataset's results in Table 1; (iii) Speakers dataset's results in Table 2; and (iv) tabular datasets' results in Table 3, our model manages to achieve comparable or (mostly) better test accuracy values.
>
> On the theoretical side, prompted by your comment (and also those of Reviewers nFiM and joei), we provide a theoretical ground for this design choice, due to Menon et al. [5]. They show that a "boundary-consistent noise" model, of which our distance-to-decision-boundary design is a special case, is not only consistent between noisy and clean domains for AUROC maximization (Proposition 1 in [5]) but also lends itself to an explicit excess AUROC risk bound to quantify the consistency (Theorem 2 in [5]). Here, we restate the theorem by adapting to our notation for convenience:
>
> > Given $\rho_x = \phi\left(d_h(x)\right)$ where $\phi\colon\mathbb{R}\_0^+\to[0,1]$ is a monotonically decreasing function and $d_h(x)$ is the distance of $x$ to the decision boundary of $h$, suppose that $\rho_{\mathrm{max}} := \max_{x \in \mathcal{X}} \rho_x < \frac{1}{2}$. Then, for any scorer $h$,
> $$
> R_{\text{rank}}(h) - R^\*\_{\text{rank}} \leq \frac{\widetilde\pi \cdot (1 - \widetilde\pi)}{\pi \cdot (1 - \pi)} \cdot \frac{1}{1 - 2 \cdot \rho_{\text{max}}} \cdot (\widetilde R_{\text{rank}}(h) - \widetilde R^\*\_{\text{rank}}),
> $$
> where $\widetilde\pi = P(\widetilde Y = +1), \pi = P(Y = +1)$; $R_{\mathrm{rank}}(h)$ denotes the true clean ranking risk of $h$, i.e.,  $\mathbb{E}\_{X\mid Y=+1, X'\mid Y=-1}[\ell_{01}(h(X) - h(X'), 1)]$ and $\widetilde{R}_{\mathrm{rank}}(h)$ the true noisy ranking risk of $h$, defined similarly, and starred risks represent the Bayes optimal ones in the respective domains.

---

> ### Author Response · Authors · 2025-11-20
> **Response to Reviewer Hbdw (5/12)**
>
> This lays down a theoretical ground for this particular choice of $\rho_x$ along with its empirical support. We note that the assumption $\rho_\mathrm{max} < \frac12$ is satisfiable trivially via, e.g., halving the output of the sigmoid or using an exponential PDF with a rate $< 1/2$. We again thank the reviewer for this comment; we amended the manuscript to include this aspect in $\rho_x$'s modeling, in Section 3.2.2.
>
> (W.2.3.)
> > The warm-up period of 4 epochs seems random, with no explanation or analysis of its impact.
>
> We kindly refer to our answer to 2nd subpoint of Weakness 1 above, i.e., W.1.2.
>
> **W.3.**
>
> (W.3.1.)
> > The main experiments (Section 4.2) use ILDN noise injection, not IDN noise, as per Xia et al. 2020. This checks how well the model handles mismatches but doesn't prove that real-world noise is like IDN.
>
> Indeed, we deliberately used ILDN way of noise injection to prevent bias to our way of modeling the noise in the main experiments. It has been proven by [4] that class-only dependent noise assumption is not enough for the real world noise. Furthermore, since around 2019, the arise of instance-dependent models has shown them to be outperforming instance-independent models, e.g., [9], [11], [8]. Therefore, we believe there has been already strong evidence that real-world noise is instance-dependent. We added an experiment on a real-world dataset [1] to support this claim (detailed below in our answer to the 2nd subpoint of Weakness 4, i.e., W.4.2.).
>
> (W.3.2.)
> > The matched-noise experiment (Section 4.1) is artificial and repetitive -- noise is added and then removed using the same model.
>
> It was in fact designed as a sanity check -- we intentionally injected the noise the way we are modeling it (not with the exact same parameters though) to see if the model was even able to generalize on the clean data when the noise is well-modeled, and the results were positive.
>
> (W.3.3.)
> > There is no real-world data study to see if IDN is enough compared to ILDN. Are there areas where ignoring label dependence affects performance?
>
> We since performed an experiment on the naturally noisy dataset "Clothing1M" [1]; we kindly refer to our answer below to the 2nd subpoint of Weakness 4, i.e., W.4.2. As for ignoring label dependence, we argue that given $X$, the true (unknown) label $Y$ has _no_ extra information whether there will be a label noise or not. In other words, $X$ alone carries all the information there is to deduce a mislabel, and therefore we are not ignoring the label dependence per se, but it is rather subsumed by the dependence on $X$. That being said, there is no free lunch here, of course, and the risk equivalence framework built on $X$-only dependence is only provably consistent so long as the design choices on, e.g., $\rho_x$ is carefully made. Still, as presented above in our response to the 2nd subpoint of Weakness 2, i.e., W.2.2., and empirically observed by a wide spectrum of Experiments in Section 4, the model has merits in classification performance under noisy label learning of binary classification.
>
> (W.3.4.)
> > The statement that IDN is "not less powerful than ILDN" (Abstract) lacks strong support. Tables 1-3 show mixed results when compared to ILDN methods (PTD, BLTM).
>
> We would argue that the results are not mixed. In Tables 1 and 2, PTD outperforms our model in 1 out of 10 classification tasks, while BLTM does 0 out of 10; in Table 3, PTD outperforms 0 out of 10 while BLTM does 1 out of 10. In the majority of the experiments, our method manages to be comparable and mostly better than these models. We note that the same sentiment carries to the real-world dataset experiment we since performed (we kindly refer to our answer below to the 2nd subpoint of Weakness 4, i.e., W.4.2.).

---

> ### Author Response · Authors · 2025-11-20
> **Response to Reviewer Hbdw (6/12)**
>
> **W.4.**
>
> (W.4.1.)
> > Authors recognize this issue, but it limits how useful the method is. There is no plan for extending it to multiple classes, even though it seems like a logical next step.
>
> Apart from the naive extensions via, e.g., one-vs-one scheme, we argue that the extension would require nontrivial work. While one might try replacing the loss suffered w.r.t. to the "opposite label", i.e., $-\widetilde y$ in Equation 3 of the proposed loss function with, e.g., (i) an average of losses suffered among all the other labels with $\rho_x$ distributed, e.g., uniformly; or (ii) loss suffered w.r.t. to the "neighboring" label, i.e., the one at the other side of the (closest) decision boundary (which requires an extra decision if the network thinks the current label is not even on either side), they would be merely inspired solutions that lack theoretical grounds. In fact, with the risk equivalence formulation and our label noise model, the resulting linear system of equations is $K \times 2$ where $K$ is the number of classes: the system is simply overdetermined for $K > 2$, hence the nontriviality of the extension.
>
> That being said, we argue that a contribution on the binary classification is still valuable. Prevalent examples from the literature include [2], [13], [14], [15], [16], [17], [18] among others. Notably, the unbiased estimator idea in [2] has been later extended to multiclass classification by [3]; similarly, the symmetric loss idea in [16] has been generalized by [19] to the multiclass setting. Furthermore, we argue that bringing instance-dependency into the risk equivalence framework is nontrivial and offers a different perspective in instance-dependent label noise modeling with a concrete implementation on binary classification.
>
> In particular, while a much more powerful/natural [4] noise model, instance-dependence requires an estimation of a transition probability _per_ instance (which is a single value in our case and it is a matrix in ILDN). The prominent ILDN models (such as [11], [8]) are driven by the factorization $P(\widetilde Y | X) = P(\widetilde Y | X, Y) P(Y | X)$ which immediately poses a possible identifiability issue: while the product $P(\widetilde Y | X, Y) P(Y | X)$ may well-model the noisy labels, one needs to pay special care to recover a nontrivial $P(Y | X)$ out of it. On the other hand, our label noise formulation is drastically different: $P(\widetilde Y \ne Y | X)$ compared to $P(\widetilde Y | X, Y)$; the latent variable $Y$ is on the left side of the condition. Therefore, we argue that treating $X$ as a sufficient statistic for the mislabel probability is a statistical insight, which allows one to look at the instance-dependent label noise problem from a different perspective, i.e., in our case, achieving risk equivalence, which is unheard of through instance- and label-dependent noise model.
>
> Overall, while being limited to binary classification, we humbly argue that there is still theoretical and practical merit in our work.

---

> ### Author Response · Authors · 2025-11-20
> **Response to Reviewer Hbdw (7/12)**
>
> (W.4.2.)
> > Small dataset size: Using parts of CIFAR-10 and small tables doesn't show if it works on large datasets. How does it handle big data like ImageNet or millions of examples?
>
> Thank you for pointing out the need for an experiment on a naturally noisy dataset. We since performed an experiment on the Clothing1M dataset [1]; an 14-class classification problem with 1 million (noisy) instance-label pairs for training, and \~10,000 (clean) pairs for testing. While the dataset provideres also made cleaning training and validation subsets available, we discard these subsets in our experiments for all models. We take 10 binary subsets from the dataset, use a 6-layer CNN (slightly more complex than the one we used for CIFAR-10 as the resolution of images are greater; the details of the network are in Appendix A.8.) with the same validation and optimization techniques as in CIFAR-10. The test accuracy results are as follows where the header represents the selected binary classes, e.g., "6v8" means "Wind-breaker versus Down Coat", and the scores within 2% of the maximum (relative) are highlighted in bold.
>
> |             | 6v8       | 6v7       | 6v9       | 1v6       | 2v6       | 0v2       | 2v9       | 2v11      | 1v7       | 0v11      |
> |:------------|:----------|:----------|:----------|:----------|:----------|:----------|:----------|:----------|:----------|:----------|
> | Normal      | 62.79     | 74.85     | 69.03     | 73.53     | 67.10     | 65.67     | 68.42     | 71.97     | 72.88     | 71.46     |
> | BCN         | 63.24     | 73.10     | 61.56     | 69.60     | 60.46     | 61.59     | 55.42     | 69.50     | 70.20     | 67.21     |
> | UB          | 65.65     | 70.29     | 64.90     | 65.33     | 58.06     | 63.56     | 57.74     | 57.71     | 71.06     | 62.77     |
> | DMI         | 64.63     | 67.95     | 64.44     | 66.33     | 58.06     | 61.59     | 57.89     | 59.27     | 68.70     | 62.31     |
> | Peer        | 65.21     | 71.11     | 67.39     | 70.94     | 63.69     | 64.87     | 65.39     | 69.91     | 72.88     | 68.52     |
> | APL         | 76.06     | 78.95     | 76.73     | 79.73     | **80.37** | 69.97     | **77.35** | **79.23** | **79.42** | **77.34** |
> | PTD         | 66.35     | 78.95     | 68.40     | 77.39     | 71.89     | 58.82     | 62.60     | 77.49     | 75.35     | 72.24     |
> | BLTM        | 64.57     | 73.80     | 59.14     | 67.50     | 53.27     | 67.06     | 58.13     | 56.55     | 72.78     | 65.97     |
> | GCE         | 70.54     | 75.20     | 71.13     | 71.78     | 72.63     | 67.64     | 72.49     | 72.46     | 75.78     | 75.31     |
> | Coteaching+ | 75.17     | 78.36     | 66.54     | **82.83** | 78.25     | 67.78     | 75.12     | 77.82     | **79.53** | **77.20** |
> | Backward    | 69.59     | **80.70** | **77.43** | **83.08** | **80.83** | 65.96     | 74.80     | 76.83     | **78.56** | **76.88** |
> | PLC         | 66.22     | 74.50     | 64.75     | 70.69     | 61.29     | 65.89     | 67.07     | 75.52     | 73.95     | 68.13     |
> | NDX         | **78.10** | **79.88** | **78.37** | **81.57** | **80.92** | **72.96** | **78.71** | **79.80** | **78.78** | **78.25** |
>
> Results suggest that our model has also competitive performance in a real-world scenario by achieving comparable or mostly better testing accuracy metrics in comparison to the baseline models. (We amended the manuscript with a new section in Section 4 for this real-world dataset, thank you again.)

---

> ### Author Response · Authors · 2025-11-20
> **Response to Reviewer Hbdw (8/12)**
>
> (W.4.3.)
> > Fairness issues with baseline methods - Some methods for multiple classes (GCE, DMI) might not work well
> > DivideMix was left out because it didn't perform well, but it's unclear why (was it a problem with how it was done or a limit of the method for two-class tasks?)
>
> While we agree that all of the methods that were designed for multiclass classification perform their empirical studies on more than 2 classes (usually around 10), to our knowledge, none of them has a restriction in their theoretical development for the number of classes to be strictly greater than 2. That being said, we indeed observed performance drops in some of the models, e.g., [8], [7] as you noted, when applied to binary classification. We conjecture that this is partially because the pipeline of some of the models is really complex, e.g., three stage training with different/shared neural networks with data distillation and co-learning, which does not appear to "downscale" well to the binary classification setting (which, of course, has still wide application areas in real life, e.g., vast majority of the Tabular datasets we experimented with in Section 4 belong to real-world binary classification tasks).
>
> (W.4.4.)
> > Using different optimizers for some methods (like Adam for Coteaching+) makes it hard to compare them.
>
> We intentionally changed the common optimizer (SGD) for Coteaching+ for a more fair comparison. We also trained it with SGD but the results were overall consistently worse than training with Adam, as the authors explicitly recommend in their paper.
>
> **W.5.**
>
> (W.5.1.)
> > There is no explanation of when $\mathbb{E}_X[\rho_x] < 0.5$ is true.
>
> We argue that this is actually a standard assumption one needs to make in any noisy label learning setup: while the
> _identification_ of the clean labels is not assumed, their existence is. In fact, in a $K$-class scenario, one needs at
> least $\dfrac1K$ clean samples to exist so that learning is even possible [5].
>
> Nonetheless, we note that the machine would be indifferent to this assumption being invalidated, which amounts to the
> semantic flip of the positive and negative labels; its predictions on unseen data would simply need to be inverted. In
> other words, training is possible either way, e.g., as an extreme case of 100% label noise in binary classification, a
> (good enough) model's predictions on the unseen data will always be worse than random _until_ they are flipped, at which
> point the semantic labels are matched. So the assumption of high signal-to-noise ratio is to preserve consistency of the
> semantic meaning of the labels between training and testing sets. Furthermore, with our choice of $\rho_x = \sigma\left(-\beta |h(x)|\right)$, the range for $\rho_x$ is $(0, 0.5]$, satisfying the assumption.
>
> (W.5.2.)
> > There is no talk about how complex the calculations are or how long they take.
>
> We thank the reviewer for raising this point on computational costs. Indeed, with $\rho_x = \sigma\left(-\beta |h(x)|\right)$ in place, since $h(x)$ is already computed, our loss functional requires 1 absolute value call, 3 sigmoid calls, 4 subtractions and 14 multiplications only, which is a negligible overhead next to the training of $h$. In comparison, many of the methods from the literature we compared against are much more complex in nature (which is not necessarily a bad characteristic but for sure trades off the time complexity).
>
> (W.5.3.)
> > There is little error analysis: what if $\rho_x$ is estimated wrong? Sensitivity analysis is only done for hyperparameters $\lambda, \beta$, not for the noise model itself. There is no study of when the method fails or does worse than other methods.
>
> Since $\beta$ is the scale parameter of the sigmoid we use to model the transition rates as $\rho_x = \sigma\left(-\beta h(x)\right)$, we have indirectly also performed a (partial) sensitivity analysis on $\rho_x$. Since the possibilities for estimation of $\rho_x$ itself is endless (which, on the bright side, goes to show flexibility), we instead opt to battle-test our particular choice for $\rho_x$ in a wide range of experiment settings: synthetic label noise with image (CIFAR-10), audio (Speakers) and tabular datasets, and (now) real-world label noise with another image dataset (Clothing1M) with neural networks and gradient-boosting decision tree models as the underlying machines. The empirical results from these validate the promised generalization ability while showing insensitivity in needing to change the distance-based noise model.

---

> ### Author Response · Authors · 2025-11-20
> **Response to Reviewer Hbdw (9/12)**
>
> **W.6.**
>
> (W.6.1.)
> > Equation 4 is hard to understand and needs more steps to explain it.
>
> We thank the reviewer for this concern. Equation 4 is Equation 3 with unknowns concretely estimated, and in turn, Equation 3 is the solution of a 2x2 linear system, which indeed does not lend itself to immediate grasp; but it is what the mathematics results in. However, the proof presented in Appendix A.1. as well as the rationale for the choice of this modified loss function immediately afterwards might help for the origins of Equation 4. Furthermore, we present insights into the equation as to how it behaves in several edge cases in Section 3.2.3.
>
> (W.6.2.)
> > The main difference (IDN vs ILDN) is not highlighted early enough.
>
> We actually have spared a paragraph in Introduction (page 2, lines 070-079) specifically for the comparison of IDN and ILDN. We believe it is early enough but we are happy to get a further feedback on its placement.
>
> (W.6.3.)
> > Some important details (like warm-up and approximations) are mentioned briefly without explanation.
>
> We thank the reviewer for this concern. We updated the manuscript with our answers here to the 1st and 2nd subpoint of Weakness 1, i.e., W.1.1. and W.1.2 above. The corresponding changes are in Appendix A.4.1 (new section) and Appendix A.5.
>
> ### Questions
> ---------
> **Q.1.**
>
> (Q.1.1.)
> > (Section 3.2.1): - You mention that $h(x)$ models both $\mathbb{P}(Y|x)$ and $\mathbb{P}(\tilde{Y}|x)$. How can one function represent both clean and noisy label distributions? Can you give a theoretical or practical analysis of when this approximation works?
>
> We kindly refer to our response to the 1st subpoint of Weakness 1 above, i.e., W.1.1.
>
> (Q.1.2.)
> > What if you use the separate model for the whole training, not just the warm-up? Appendix A.3 shows worse results, but the separate model used a fixed $\mathbb{P}(\tilde{Y}|x)$ -- what if both networks are updated together?
>
> It is indeed possible to do this via, e.g., the (incomplete) likelihood of the noisy labels where it is demarginalized over the clean label probability as, e.g., $P(\widetilde Y \mid X) = P(\widetilde Y \mid Y,X) P(Y \mid X)$. A canonical example of such a model that uses the Expectation-Maximization to maximize the said likelihood is of Xiao et al. [1], where the authors treat the clean label (as well as the noise kind of an instance, named $Z$) as a hidden variable. This involves training two networks for $Y$ and $Z$, and learning the transition matrix (on a small dataset that assumes identified clean and noisy labels), all based on likelihood maximization. Apart from modeling the noise as class-only dependent, major limitations of this approach are twofold: (i) optimization gets complex as they learn the parameters of the said networks using EM (in fact, the algorithm is sensitive to initialization and to prevent the optimization from reaching "totally wrong posterior computations", they pretrain the networks on clean data first); (ii) they assume the existence and identifiability of the clean labels in the data (to pretrain the latent-variable modeling networks as well as the transition matrix). Our instance-only dependent label noise model, however, cannot be integrated into this framework because in $\rho_x = P(\widetilde Y \ne Y \mid X)$, $Y$ and $\widetilde Y$ are on the left side of the condition together, i.e., the clean label posterior cannot be factored as above using $\rho_x$. In other words, even one manages a simultaneous learning of $\widetilde Y$ and $Y$, our noise model does not lend itself to a mathematical connection between these objects.
>
> **Q.2.**
>
> (Q.2.1.)
> > Can you explain when $|h(x)|$ is a good way to measure distance to the decision boundary? - In neural networks with ReLU activations, the decision boundary can be made of straight-line segments. Does your method still work well in areas far from the training data?
>
> This approximation does not make an assumption on the topological structure of the boundary found in the last layer of the network, but on the power of the machine, e.g., it becomes exact if the machine is linear (e.g., logistic regression) or behaves linearly in its last layer (e.g., an interpolating neural network). For a detailed argument, we kindly refer to our response to the 3rd subpoint of Weakness 1 above, i.e., W.1.3.
>
> (Q.2.2.)
> > Have you thought about using real distance calculations (like DeepFool) for some samples to check how good the approximation is during training?
>
> We indeed have. While it was certainly a better approximation, it was a very slow process due to its iterative nature. We note that this distance computation needs to be done in every forward call to compute the loss suffered; DeepFool, even with a reasonable maximum iteration limit, proved to take intolerably long, hence our approximation, which is much cheaper to compute (and also has practical and theoretical support as discussed in W.1.3.).

---

> ### Author Response · Authors · 2025-11-20
> **Response to Reviewer Hbdw (10/12)**
>
> **Q.3.**
> > Why is the 4-epoch warm-up chosen? Does it affect performance? - Can you show results for 0, 2, 4, 8, and 16 warm-up epochs? - Is there a method to decide the warm-up length, like using the clean sample ratio or when $h$ converges?
>
> We kindly refer to our response to the 2nd subpoint of Weakness 1 above, i.e., W.1.2.
>
> **Q.4.**
>
> (Q.4.1.)
> > Your main experiments use ILDN noise injection. Can you also test with true IDN noise to check the model's assumptions?
>
> We actually tried this (Section 4.1.) to see if the model possesses the claimed generalization ability when the noise process is well-modeled by injecting the noise the way we are modeling it. Furthermore, we since also experimented with a naturally noisy dataset (Clothing1M) that is supposed to contain instance-dependent noise [1]; we kindly refer to our response to the 2nd subpoint of Weakness 4 above, i.e., W.4.2.
>
> (Q.4.2.)
> > When would IDN not be enough, and ILDN be needed? Can you describe the task or data features that decide this?
>
> Theoretically, we would argue that ILDN is performing an unnecessary conditioning on the (unknown) true label because it is an aggregated discrete statistic that is derived from what we argue the sufficient statistic, $X$. In other words, $\widetilde Y$ and $Y$ are conditionally independent given $X$. This sounds reasonable: when an annotator is given an image (i.e., an $X$) to label it (i.e., yield a $\widetilde Y$), the true label is not in the picture at all; it is only $X$ that annotator looks at to decide $\widetilde Y$. However, after the fact, when it is time to model this process, making use of the latent variable (and even only that) proved immensely useful as the literature's main focus has been on CCN models (and relatively recently ILDN), as we discuss in Related Work. In light of this, one setting where ILDN is favorable to IDN would be if one has access to a set of surely clean X-y pairs. For example, while [11], a prominent ILDN method, does not assume the availability of such a subset to widen its applicability (and instead uses anchor-like points), their theory is actually dependent on the anchor points (Step 4 in their Algorithm 1), i.e., those instances with almost surely clean labels, and therefore is expected to perform better than, e.g., IDN if such a subset is made available.
>
> (Q.4.3.)
> > Tables 1-3 show PTD and BLTM (ILDN methods) sometimes do better than NDX. Does this mean ILDN is better for these cases?
>
> We indeed do not expect our model (or any model for that matter) to always work better across all experiments. It is possible that another model proves more suitable for a given dataset with its (unknown) noise process. On average, however, we claim that the real-world noise is much more realistically modeled with IDN and ILDN, and similarly, on average, our model appears to outperform or maintain comparability with respect to all of the 12 baseline models we tried.
>
> **Q.5.**
>
> (Q.5.1.)
> > The conditions in Proposition 2 are hard to check in real situations. Can you give practical tips or simple methods for setting $\lambda$ other than grid search? - How does the best $\lambda$ depend on dataset features like size, noise, and dimension?
>
> We kindly refer to our response to the 1st subpoint of Weakness 2 above, i.e., W.2.1.
>
> (Q.5.2.)
> > Have you seen cases where the denominator in Eq. 3 gets close to zero during training even with regularization?
>
> The regularization trick to approximate division with a repeated subtraction is to avoid the cases where denominator gets close to 0; therefore in the regularized version there is no denominator to check if it gets close to zero. If you meant denominator getting close to 0 was something we actually observed: yes, indeed, our first experiments were with the original form of the loss function with division intact, and from time to time, the loss would get arbitrarily large, harming (terminating) the training. We would appreciate a clarification if we misunderstood the question.
>
> **Q.6.**
>
> (Q.6.1.)
> > How does the method work with larger datasets like full ImageNet and more complex problems?
>
> We kindly refer to our response to the 2nd subpoint of Weakness 4 above, i.e., W.4.2.
>
> (Q.6.2.)
> > Can you explain how to extend it to multi-class classification? Is it as simple as one-vs-rest, or are there major challenges?
>
> We kindly refer to our response to the 1st subpoint of Weakness 4 above, i.e., W.4.1.
>
> (Q.6.3.)
> > For the tabular LightGBM results, how does the method work with GBM's way of handling noisy data?
>
> To our knowledge, LightGBM's ability to handle noisy data lies in the _feature_ space rather than the label space, e.g., it has a "cat_smooth" parameter to smooth out possible noise in categorical features. We would appreciate a clarification if we miss the point.

---

> ### Author Response · Authors · 2025-11-20
> **Response to Reviewer Hbdw (11/12)**
>
> (Q.6.4.)
> > Are there times when NDX does much worse than the usual methods? What are these situations like? - What happens to performance when the noise model is very wrong (for example, the real noise depends on the class, but you think it is IDN)?
>
> One can come up with an artificial noise process, e.g., uniform label noise independent of instance and label, where the instance-dependence assumption is easily validated; in this case, a RCN-robust method is expected to perform much better. However, we argue that this is not a realistic case -- as shown in [4], the real-world noise is most likely dependent on the instance. For a class-only dependent scenario, [4] shows that such an assumption is not realistic from the noise process' point of view; what is realistic and has shown great improvements (as opposed to doing nothing special) is _assuming_ the noise process is as such and developing models to combat that. This approach, however, focuses on an aggregated discrete statistic that is the (unknown) true label $Y$, which is a derived quantity from what we argue the entire/sufficient statistic that is $X$. Therefore, it does not appear possible to generate $Y$-only dependent noise while $Y$ itself is already $X$-dependent.
>
> Furthermore, a notable scenario is a label noise model's performance on a curated, clean training set. We show this performance values in Figure 1 over the CIFAR-10 dataset, where the left-most data points in each subfigure corresponds to the test accuracies for models trained over the clean dataset, i.e., with 0% artificial label noise. Even in this case, our model does not show an unnecessary performance degradation but has essentially the same performance metrics as the Normal model.
>
> (Q.6.5.)
> > In Section 3.2.3, you show that $\tilde{\ell}(\cdot, \tilde{y})$ becomes ${\ell}(\cdot, -\tilde{y})$ when $\tilde{y} \cdot h(x)$ goes to negative infinity. This means the model trusts itself more than the label. Could this cause the model to repeat its mistakes? How does this compare to methods that reduce the importance of doubtful samples?
>
> We would like to make a distinction between two kinds of "doubtful" samples our model addresses:
>
> 1. The "obvious" mislabels where $\widetilde{y} \cdot h(x)$ is negatively very large. The only way this happens is if the sample is very far away from the decision boundary of the machine, and it cannot get this (oppositely) confident on a sample merely by repeating its mistake. We note that a random initialization gives on average a 0.5 confidence over all samples, and for the machine to achieve 0.99+ confidence on a given sample opposite to its given label, it must be an outlier: otherwise its neighborhood would also get parallel confidence with its original label; but then a decision boundary must be formed around it, contradicting the fact that it is very far away from the decision boundary. Therefore, this is the model's one way of detecting random label flips due to, e.g., misclicks or communication issues.
>
> 2. The "gray area" samples that lie around the decision boundary. As exemplified in the section you noted, loss correction performs a (nontrivial) weighted combination of the normal loss and the opposite one, and does not lean towards any side in the hope for robustness. Therefore, this is the model's another way of reducing the importance of doubtful samples.

---

> ### Author Response · Authors · 2025-11-20
> **Response to Reviewer Hbdw (12/12)**
>
> ### References
> --------------
> - [1] "Learning from Massive Noisy Labeled Data for Image Classification", Xiao et al.; IEEE CVPR, 2015.
> - [2] "Learning with Noisy Labels", Natarajan et al.; NIPS, 2013.
> - [3] "Making Deep Neural Networks Robust to Label Noise: a Loss Correction Approach", Patrini et al.; IEEE CVPR, 2017.
> - [4] "Beyond Class-conditional Assumption: A Primary Attempt to Combat Instance-Dependent Label Noise". Chen et al.; AAAI, 2021.
> - [5] "Learning from Binary Labels with Instance-dependent Noise", Menon et al.; Machine Learning, 2018/09.
> - [6] "On the Role of Label Noise in the Feature Learning Process", Han et al.; ICML, 2025.
> - [7] "Dividemix: Learning with noisy labels as semi-supervised learning", Li et al.; ICLR, 2020.
> - [8] "Estimating instance-dependent Bayes-label transition matrix using a deep neural network", Yang et al; ICML, 2022.
> - [9] "Learning with feature-dependent label noise: A progressive approach", Zhang et al.; ICLR, 2021.
> - [10] "On the decision boundary of deep neural networks", Li et al.; arXiv, 2019.
> - [11] "Part-dependent label noise: Towards instance-dependent label noise.", Xia et al.; NIPS, 2020.
> - [12] "Learning noisy linear threshold functions", Bylander; Technical Report, 1998.
> - [13] "Modelling class noise with symmetric and asymmetric distributions", Du and Cai; AAAI, 2015.
> - [14] "A rate of convergence for mixture proportion estimation, with application to learning from noisy labels", Scott; AISTATS, 2015.
> - [15] "Learning with symmetric label noise: the importance of being unhinged", van Rooyen et al.; NIPS, 2015.
> - [16] "Making risk minimization tolerant to label noise", Ghosh et al.; Neurocomputing, 2015:160.
> - [17] "Classification with noisy labels by importance reweighting", Liu and Tao; IEEE TPAMI, 2016.
> - [18] "Peer loss functions: learning from noisy labels without knowing noise rates", Liu and Guo; ICML, 2020.
> - [19] "Normalized loss functions for deep learning with noisy labels", Ma et al.; ICML, 2020.

---

### Official Review · Reviewer_joei · 2025-10-31

**Soundness:** 3
**Presentation:** 3
**Contribution:** 2
**Rating:** 4
**Confidence:** 4

**Summary:**

This paper investigates the label noise problem in binary classification. The authors revisit the **Instance-Only Dependent Noise (IDN)** model, a rarely explored framework where the label corruption depends solely on the instance features. The authors also propose an instance-aware loss correction method based on risk equivalence. At its core, the method models a per-instance noise rate $\rho_x = P(Y \neq \tilde{Y} \mid X=x)$ instead of a full transition matrix. Several ways for the estimation of transition rates are proposed. Experiments on diverse datasets (image, audio, and tabular) are conducted to verify the proposed methods.

**Strengths:**

1. The paper introduces an unbiased risk that incorporates the per-instance noise rate $\rho_x = P(Y \neq \tilde{Y} \mid X = x)$.
2. The authors conducted experiments on several datasets to verify the proposed method.

**Weaknesses:**

1. The proposed method is currently restricted to binary classification, and its extension to multi-class settings remains unexplored. This limits the applicability of the approach to broader real-world scenarios.
2. The paper does not include experiments on large-scale real-world noisy datasets (e.g., WebVision or Clothing1M). Evaluating on such datasets would provide stronger evidence of the method's scalability and practical robustness.

3. The estimation of the per-instance noise rate $\rho_x$ relies on heuristic proxies (e.g., distance to the decision boundary). The paper does not provide an in-depth analysis of how the choice of proxy or mapping function affects performance. Different proxies may exhibit different levels of robustness.

4. The paper lacks a clear discussion of its relationship with prior loss-correction methods, such as Natarajan et al. (2013). A more explicit theoretical analysis or empirical comparison would help clarify the novelty and contribution of the paper.

**Questions:**

Please see the weaknesses.

---

> ### Author Response · Authors · 2025-11-20
> **Response to Reviewer joei (1/4)**
>
> We thank you for your time and valuable feedback! Below we present our answers to the weakness points.
>
> ### Weaknesses
> ----------
> **W.1.**
> > The proposed method is currently restricted to binary classification, and its extension to multi-class settings remains unexplored. This limits the applicability of the approach to broader real-world scenarios.
>
> Apart from the naive extensions via, e.g., one-vs-one scheme, we argue that the extension would require nontrivial work. While one might try replacing the loss suffered w.r.t. to the "opposite label", i.e., $-\widetilde y$ in Equation 3 of the proposed loss function with, e.g., (i) an average of losses suffered among all the other labels with $\rho_x$ distributed, e.g., uniformly; or (ii) loss suffered w.r.t. to the "neighboring" label, i.e., the one at the other side of the (closest) decision boundary (which requires an extra decision if the network thinks the current label is not even on either side), they would be merely inspired solutions that lack theoretical grounds. In fact, with the risk equivalence formulation and our label noise model, the resulting linear system of equations is $K \times 2$ where $K$ is the number of classes: the system is simply overdetermined for $K > 2$, hence the nontriviality of the extension.
>
> That being said, we argue that a contribution on the binary classification is still valuable. Prevalent examples from the literature include [1], [2], [3], [4], [5], [6], [7] among others. Notably, the unbiased estimator idea in [1] has been later extended to multiclass classification by [8]; similarly, the symmetric loss idea in [5] has been generalized by [9] to the multiclass setting. Furthermore, we argue that bringing instance-dependency into the risk equivalence framework is nontrivial and offers a different perspective in instance-dependent label noise modeling with a concrete implementation on binary classification.
>
> In particular, while a much more powerful/natural [10] noise model, instance-dependence requires an estimation of a transition probability _per_ instance (which is a single value in our case and it is a matrix in ILDN). The prominent ILDN models (such as [11], [12]) are driven by the factorization $P(\widetilde Y | X) = P(\widetilde Y | X, Y) P(Y | X)$ which immediately poses a possible identifiability issue: while the product $P(\widetilde Y | X, Y) P(Y | X)$ may well-model the noisy labels, one needs to pay special care to recover a nontrivial $P(Y | X)$ out of it. On the other hand, our label noise formulation is drastically different: $P(\widetilde Y \ne Y | X)$ compared to $P(\widetilde Y | X, Y)$; the latent variable $Y$ is on the left side of the condition. Therefore, we argue that treating $X$ as a sufficient statistic for the mislabel probability is a statistical insight, which allows one to look at the instance-dependent label noise problem from a different perspective, i.e., in our case, achieving risk equivalence, which is unheard of through instance- and label-dependent noise model.
>
> Overall, while being limited to binary classification, we humbly argue that there is still theoretical and practical merit in our work.

---

> ### Author Response · Authors · 2025-11-20
> **Response to Reviewer joei (2/4)**
>
> **W.2.**
> > The paper does not include experiments on large-scale real-world noisy datasets (e.g., WebVision or Clothing1M). Evaluating on such datasets would provide stronger evidence of the method's scalability and practical robustness.
>
> Thank you for pointing out the need for an experiment on a naturally noisy dataset. We since performed an experiment on the Clothing1M dataset [13]; an 14-class classification problem with 1 million (noisy) instance-label pairs for training, and \~10,000 (clean) pairs for testing. While the dataset provideres also made cleaning training and validation subsets available, we discard these subsets in our experiments for all models. We take 10 binary subsets from the dataset, use a 6-layer CNN (slightly more complex than the one we used for CIFAR-10 as the resolution of images are greater; the details of the network are in Appendix A.8.) with the same validation and optimization techniques as in CIFAR-10. The test accuracy results are as follows where the header represents the selected binary classes, e.g., "6v8" means "Wind-breaker versus Down Coat", and the scores within 2% of the maximum (relative) are highlighted in bold.
>
> |             | 6v8       | 6v7       | 6v9       | 1v6       | 2v6       | 0v2       | 2v9       | 2v11      | 1v7       | 0v11      |
> |:------------|:----------|:----------|:----------|:----------|:----------|:----------|:----------|:----------|:----------|:----------|
> | Normal      | 62.79     | 74.85     | 69.03     | 73.53     | 67.10     | 65.67     | 68.42     | 71.97     | 72.88     | 71.46     |
> | BCN         | 63.24     | 73.10     | 61.56     | 69.60     | 60.46     | 61.59     | 55.42     | 69.50     | 70.20     | 67.21     |
> | UB          | 65.65     | 70.29     | 64.90     | 65.33     | 58.06     | 63.56     | 57.74     | 57.71     | 71.06     | 62.77     |
> | DMI         | 64.63     | 67.95     | 64.44     | 66.33     | 58.06     | 61.59     | 57.89     | 59.27     | 68.70     | 62.31     |
> | Peer        | 65.21     | 71.11     | 67.39     | 70.94     | 63.69     | 64.87     | 65.39     | 69.91     | 72.88     | 68.52     |
> | APL         | 76.06     | 78.95     | 76.73     | 79.73     | **80.37** | 69.97     | **77.35** | **79.23** | **79.42** | **77.34** |
> | PTD         | 66.35     | 78.95     | 68.40     | 77.39     | 71.89     | 58.82     | 62.60     | 77.49     | 75.35     | 72.24     |
> | BLTM        | 64.57     | 73.80     | 59.14     | 67.50     | 53.27     | 67.06     | 58.13     | 56.55     | 72.78     | 65.97     |
> | GCE         | 70.54     | 75.20     | 71.13     | 71.78     | 72.63     | 67.64     | 72.49     | 72.46     | 75.78     | 75.31     |
> | Coteaching+ | 75.17     | 78.36     | 66.54     | **82.83** | 78.25     | 67.78     | 75.12     | 77.82     | **79.53** | **77.20** |
> | Backward    | 69.59     | **80.70** | **77.43** | **83.08** | **80.83** | 65.96     | 74.80     | 76.83     | **78.56** | **76.88** |
> | PLC         | 66.22     | 74.50     | 64.75     | 70.69     | 61.29     | 65.89     | 67.07     | 75.52     | 73.95     | 68.13     |
> | NDX         | **78.10** | **79.88** | **78.37** | **81.57** | **80.92** | **72.96** | **78.71** | **79.80** | **78.78** | **78.25** |
>
> Results suggest that our model has also competitive performance in a real-world scenario by achieving comparable or mostly better testing accuracy metrics in comparison to the baseline models. (We amended the manuscript with a new section in Section 4 for this real-world dataset, thank you again.)

---

> ### Author Response · Authors · 2025-11-20
> **Response to Reviewer joei (3/4)**
>
> **W.3.**
> > The estimation of the per-instance noise rate $\rho_x$ relies on heuristic proxies (e.g., distance to the decision boundary). The paper does not provide an in-depth analysis of how the choice of proxy or mapping function affects performance. Different proxies may exhibit different levels of robustness.
>
> Estimation of $\rho_x$ is indeed a design choice; while we argue that it brings flexibility to the overall modeling process (as opposed to, say, an ILDN model where the transition matrix is estimated with a very specific way, e.g., by focusing on different parts of a given image in [11] or three stage training using distillation of the Bayes-optimal labels as in [12]), we also acknowledge that this freedom needs to be guided with sound design choices. The box we present in Section 3.2.2. offers several ways to model this quantity; but it is in no way an exhaustive list (as noted) and it is therefore hard to compare this large of a pool of choices.
>
> That being said, we would like to offer empirical and theoretical support for our choice in the paper for the experiments, i.e., distance-to-decision-boundary based modeling of $\rho_x = \sigma\left(-\beta |h(x)|\right)$.
>
> On the empirical side:
>
> - It is arguably a very intuitive one: an instance closer to the decision boundary tends to be more "ambiguous" as it carries features of more than one class, making it a plausible candidate for mislabeling. Moreover, historically, it found applications in [14] and [2] (albeit for linear machines).
>
> - It has a very low computational cost: $h(x)$ is already computed in a forward pass of the machine.
>
> - Across a wide spectrum of dataset-noise rate-machine experiment settings, i.e., (i) sanity check for the noise model in Figure 1; (ii) CIFAR-10 dataset's results in Table 1; (iii) Speakers dataset's results in Table 2; and (iv) tabular datasets' results in Table 3, our model manages to achieve comparable or (mostly) better test accuracy values.
>
> On the theoretical side, prompted by your comment (and also those of Reviewers nFiM and Hbdw), we provide a theoretical ground for this design choice, due to Menon et al. [15]. They show that a "boundary-consistent noise" model, of which our distance-to-decision-boundary design is a special case, is not only consistent between noisy and clean domains for AUROC maximization (Proposition 1 in [15]) but also lends itself to an explicit excess AUROC risk bound to quantify the consistency (Theorem 2 in [15]). Here, we restate the theorem by adapting to our notation for convenience:
>
> > Given $\rho_x = \phi\left(d_h(x)\right)$ where $\phi\colon\mathbb{R}\_0^+\to[0,1]$ is a monotonically decreasing function and $d_h(x)$ is the distance of $x$ to the decision boundary of $h$, suppose that $\rho_{\mathrm{max}} := \max_{x \in \mathcal{X}} \rho_x < \frac{1}{2}$. Then, for any scorer $h$,
> $$
> R_{\text{rank}}(h) - R^\*\_{\text{rank}} \leq \frac{\widetilde\pi \cdot (1 - \widetilde\pi)}{\pi \cdot (1 - \pi)} \cdot \frac{1}{1 - 2 \cdot \rho_{\text{max}}} \cdot (\widetilde R_{\text{rank}}(h) - \widetilde R^\*\_{\text{rank}}),
> $$
> where $\widetilde\pi = P(\widetilde Y = +1), \pi = P(Y = +1)$; $R_{\mathrm{rank}}(h)$ denotes the true clean ranking risk of $h$, i.e.,  $\mathbb{E}\_{X\mid Y=+1, X'\mid Y=-1}[\ell_{01}(h(X) - h(X'), 1)]$ and $\widetilde{R}_{\mathrm{rank}}(h)$ the true noisy ranking risk of $h$, defined similarly, and starred risks represent the Bayes optimal ones in the respective domains.
>
> This lays down a theoretical ground for this particular choice of $\rho_x$ along with its empirical support. We note that the assumption $\rho_\mathrm{max} < \frac12$ is satisfiable trivially via, e.g., halving the output of the sigmoid or using an exponential PDF with a rate $< 1/2$. We again thank the reviewer for this comment; we amended the manuscript to include this aspect in $\rho_x$'s modeling, in Section 3.2.2.

---

> ### Author Response · Authors · 2025-11-20
> **Response to Reviewer joei (4/4)**
>
> **W.4.**
> > The paper lacks a clear discussion of its relationship with prior loss-correction methods, such as Natarajan et al. (2013). A more explicit theoretical analysis or empirical comparison would help clarify the novelty and contribution of the paper.
>
> We have actually spared a paragraph at the end of the Related Work section (Section 2) to zoom into how our approach compares with that of Natarajan et al. [1] and Patrini et al. [8]. Furthermore, in Appendix A.1., immediately after the proof of Proposition 1, we present a theoretical contrast from Natarajan et al.'s work. To summarize these comparisons, we present a table below:
>
> |                                | Natarajan et al. [1] | Patrini et al. [8] | Ours              |
> |--------------------------------|------------------------------|--------------------------|-------------------|
> | Loss correction                | Yes                          | Yes                      | Yes               |
> | Classification Task            | Binary                       | Multiclass               | Binary            |
> | Instance-dependent label noise | No                           | No                       | Yes               |
> | Estimation of the noise rates  | Hyperparameters; validation  | Learned; disjointly      | Learned; jointly  |
> | Anchor points                  | No                           | Yes                      | No                |
> | Machine agnostic               | Yes                          | No*                      | Yes               |
> | Uncertainty on Y               | No                           | Yes                      | Yes               |
> | Risk equivalence               | No**                         | Yes                      | Yes               |
>
> *: In [8], authors require a machine that "can perfectly model the probability of the noisy labels" for a good estimation of the noise rates as a (disjoint) step before the actual training, and they advocate neural networks.
>
> **: In [1]'s "method of unbiased estimators", the latent variable $Y$ is treated deterministically, i.e., $P(Y\mid X) = 0$ or $P(Y\mid X) = 1$ instead of $P(Y\mid X) \in [0, 1]$. Therefore, it does not estimate $P(Y \mid X)$ in its formulation and is said to achieve risk equivalence only if no uncertainty on $Y$ is assumed.
>
> ### References
> -----------------
> - [1] "Learning with Noisy Labels", Natarajan et al.; NIPS, 2013.
> - [2] "Modelling class noise with symmetric and asymmetric distributions", Du and Cai; AAAI, 2015.
> - [3] "A rate of convergence for mixture proportion estimation, with application to learning from noisy labels", Scott; AISTATS, 2015.
> - [4] "Learning with symmetric label noise: the importance of being unhinged", van Rooyen et al.; NIPS, 2015.
> - [5] "Making risk minimization tolerant to label noise", Ghosh et al.; Neurocomputing, 2015:160.
> - [6] "Classification with noisy labels by importance reweighting", Liu and Tao; IEEE TPAMI, 2016.
> - [7] "eer loss functions: learning from noisy labels without knowing noise rates", Liu and Guo; ICML, 2020.
> - [8] "Making Deep Neural Networks Robust to Label Noise: a Loss Correction Approach", Patrini et al.; IEEE CVPR, 2017.
> - [9] "Normalized loss functions for deep learning with noisy labels", Ma et al.; ICML, 2020.
> - [10] "Beyond Class-conditional Assumption: A Primary Attempt to Combat Instance-Dependent Label Noise". Chen et al.; AAAI, 2021.
> - [11] "Part-dependent label noise: Towards instance-dependent label noise.", Xia et al.; NIPS, 2020.
> - [12] "Estimating instance-dependent Bayes-label transition matrix using a deep neural network", Yang et al; ICML, 2022.
> - [13] "Learning from Massive Noisy Labeled Data for Image Classification", Xiao et al.; IEEE CVPR, 2015.
> - [14] "Learning noisy linear threshold functions", Bylander; Technical Report, 1998.
> - [15] "Learning from Binary Labels with Instance-dependent Noise", Menon et al.; Machine Learning, 2018/09.

---

### Official Review · Reviewer_nFiM · 2025-11-01

**Soundness:** 3
**Presentation:** 3
**Contribution:** 3
**Rating:** 6
**Confidence:** 3

**Summary:**

This paper addresses learning under label noise in supervised binary classification by introducing a risk-equivalence-based loss correction that ensures robustness when label flips depend only on the input instance. The method derives a corrected loss so that minimizing the empirical noisy risk aligns with minimizing the true clean risk for any classification-calibrated base loss (e.g., logistic).

The authors propose efficient ways to estimate the per-instance noise rate $\eta_x$	​without requiring matrix-valued transition models, extending CCN and ILDN approaches with lower complexity. Experiments on image, audio, and tabular datasets show strong robustness to moderate and high noise, matching or exceeding prior methods.

Overall, the paper offers a principled and efficient framework for instance-dependent noise modeling. While conceptually incremental, it combines theoretical rigor and empirical breadth to provide a valuable step toward scalable, noise-tolerant learning.

**Strengths:**

- Provides a rigorous derivation of a corrected loss guaranteeing risk equivalence between noisy and clean risks, with solid theoretical grounding (Propositions 1, 2; Lemma 1).
- Introduces a stable regularized variant with Lipschitz and generalization-bound analysis.
- Revives the under-studied **instance-only dependent noise (IDN)** model, offering a simpler yet effective alternative to ILDN.
- Proposes an elegant, computationally efficient estimation of $ \rho_x$ using $ |h(x)| $ as a difficulty proxy, avoiding complex multi-stage procedures.
- Demonstrates machine-agnostic applicability across neural networks and gradient-boosted trees.
- Achieves strong and stable performance across image, audio, and tabular datasets, outperforming or matching competitive baselines under moderate and high noise.

**Weaknesses:**

- Theoretical contributions mainly extend prior loss-correction frameworks (e.g., Natarajan et al., Patrini et al.), with novelty centered on the instance-only dependency rather than new statistical insights.
- The warm-up assumption for accurate noise estimation is heuristic and may break under severe noise.
- The formulation $\rho_x = f(|h(x)|$ is empirically motivated without strong theoretical grounding.
- Experiments are limited to binary classification; multi-class extension is left for future work.
- Evaluation relies on relatively small or subsampled datasets, lacking results on large-scale real-world noise (e.g., WebVision).
- The drastic accuracy drop below 50% in Figure 1 suggests potential tuning or early-stopping issues in baseline comparisons

**Questions:**

- Can the proposed framework be extended to multi-class classification tasks?
- How sensitive is the method to the duration of the warm-up stage used for estimating ($P(Y|x)$?
- How does the estimation of $\eta_x$ influence performance when no label noise is present? Including results for a 0% noise setting (e.g., in Table 1) would clarify this.

---

> ### Author Response · Authors · 2025-11-20
> **Response to Reviewer nFiM (1/4)**
>
> We thank you for your time and valuable feedback! Below we present our answers to the weakness points and questions.
>
> ### Weaknesses
> ----------
> **W.1.**
> > Theoretical contributions mainly extend prior loss-correction frameworks (e.g., Natarajan et al., Patrini et al.), with novelty centered on the instance-only dependency rather than new statistical insights.
>
> While our work is indeed in a close relation to class-only dependent models of [1] and [2] (differences from which we discuss in detail at the end of the Related Work section (Section 2)), we argue that bringing instance-dependency into the equation is a nontrivial step. While a much more powerful/natural [3] noise model, it requires an estimation of a transition probability _per_ instance (which is a single value in our case and it is a matrix in ILDN). The prominent ILDN models (such as [4], [5]) are driven by the factorization $P(\widetilde Y | X) = P(\widetilde Y | X, Y) P(Y | X)$ which immediately poses a possible identifiability issue: while the product $P(\widetilde Y | X, Y) P(Y | X)$ may well-model the noisy labels, one needs to pay special care to recover a nontrivial $P(Y | X)$ out of it. On the other hand, our label noise formulation is drastically different: $P(\widetilde Y \ne Y | X)$ compared to $P(\widetilde Y | X, Y)$; the latent variable $Y$ is on the left side of the condition. Therefore, we argue that treating $X$ as a sufficient statistic for the mislabel probability is a statistical insight, which allows one to look at the instance-dependent label noise problem from a different perspective, i.e., in our case, achieving risk equivalence, which is unheard of through instance- and label-dependent noise model.
>
> Relatedly, we also list how our modified loss function behaves in several edge cases to draw insights as to how it works in comparison with a noise-intolerant loss in Section 3.2.3.
>
> **W.2.**
> > The warm-up assumption for accurate noise estimation is heuristic and may break under severe noise.
>
> 1. It is indeed a heuristic, albeit not an uncommon one among prominent label noise methods, e.g., DivideMix [6], BLTM [5] and PLC [7]. (In fact, in [7] (Lemma 2), authors theoretically justify the need for such a period early in the training, albeit for their specific noise model.) While these label noise models (and ours) are highly different in how the problem is addressed, the warm-up period is used for the unique aim of "attain[ing] a reasonable network" [7] for the "initial convergence of the algorithm" [6] to, e.g., "collect distilled examples" [5]. In other words, the warm-up period is used without any architectural modifications to learn a base network under the dominance of the clean labels [8] before it starts overfitting the label noise, at which point the proposed mechanisms start taking place. We also note that in some works that do not make use of an explicit warm-up period, the counterpart assumption of the identification of almost-surely clean samples (i.e., the anchor or anchor-like points) is in place, e.g., [9], [2].
>
> 2. Furthermore, as empirically shown across a wide spectrum of dataset-noise rate-machine settings, i.e., (i) sanity check for the noise model in Figure 1; (ii) CIFAR-10 dataset's results in Table 1; (iii) Speakers dataset's results in Table 2; and (iv) tabular datasets' results in Table 3, our model manages to achieve comparable or (mostly) better test accuracy values even under severe label noise, i.e., 44%.
>
> 3. Prompted by your comment (and also that of Reviewer Hbdw), we performed an experiment on the effect of the number of warm-up epochs by sweeping it to see the need for it as well as the sensitivity against it. We sweep the ratio "#Warm-Up Epochs / #Total Epochs" from 0 to 1. Ideally, we expect a reverse U-shape with a rather flat maxima w.r.t. test accuracy to represent (i) the trade-off: when the ratio is 1, model reduces to the "Normal" model, i.e., does nothing special about the label noise and has an inferior performance than the other rates; when the ratio is 0, dominance of the clean labels is not sufficiently utilized to form a reasonable baseline to correct; and (ii) the insensitivity against the number of warm-up epochs in the middle region as a rather flat maxima.
>
> We used the Tabular Adult dataset with 28% and 44% label noise as in Section 4. We train and validate our model as done in the usual experiments while sweeping the warm-up rate. We plotted the warm-up rate versus test accuracy with two different noise rates and the corresponding figure is placed in Appendix A.5 (here is [anonymous link](https://imgshare.cc/1lkkj020) to an online image server for your convenience). We roughly observe the mentioned trade-off and insensitivity over the number of warm-up epochs (while not being perfect U-shapes), supporting the warm-up strategy. The insensitivity also allows for not treating the number of warm-up epochs as a hyperparameter that needs heavy tuning (also the case in [5] - [7]).

---

> ### Author Response · Authors · 2025-11-20
> **Response to Reviewer nFiM (2/4)**
>
> **W.3.**
> > The formulation $\rho_x = f(|h(x)|$ is empirically motivated without strong theoretical grounding.
>
> It is indeed one design choice, arguably a very intuitive one: an instance closer to the decision boundary tends to be more "ambiguous" as it carries features of more than one class, making it a plausible candidate for mislabeling. Moreover, historically, it found applications in [10] and [11] (albeit for linear machines).
>
> Prompted by your comment (and also those of Reviewers joei and Hbdw), on the theoretical side, Menon et al. [12] shows that a "boundary-consistent noise" model, of which our distance-to-decision-boundary design, i.e., $\rho_x = \sigma(-\beta |h(x)|)$ is a special case, is not only consistent between noisy and clean domains for AUROC maximization (Proposition 1 in [12]) but also lends itself to an explicit excess AUROC risk bound to quantify the consistency (Theorem 2 in [12]). Here, we restate the theorem by adapting to our notation for convenience:
>
> > Given $\rho_x = \phi\left(d_h(x)\right)$ where $\phi\colon\mathbb{R}\_0^+\to[0,1]$ is a monotonically decreasing function and $d_h(x)$ is the distance of $x$ to the decision boundary of $h$, suppose that $\rho_{\mathrm{max}} := \max_{x \in \mathcal{X}} \rho_x < \frac{1}{2}$. Then, for any scorer $h$,
> $$
> R_{\text{rank}}(h) - R^\*\_{\text{rank}} \leq \frac{\widetilde\pi \cdot (1 - \widetilde\pi)}{\pi \cdot (1 - \pi)} \cdot \frac{1}{1 - 2 \cdot \rho_{\text{max}}} \cdot (\widetilde R_{\text{rank}}(h) - \widetilde R^\*\_{\text{rank}}),
> $$
> where $\widetilde\pi = P(\widetilde Y = +1), \pi = P(Y = +1)$; $R_{\mathrm{rank}}(h)$ denotes the true clean ranking risk of $h$, i.e.,  $\mathbb{E}\_{X\mid Y=+1, X'\mid Y=-1}[\ell_{01}(h(X) - h(X'), 1)]$ and $\widetilde{R}_{\mathrm{rank}}(h)$ the true noisy ranking risk of $h$, defined similarly, and starred risks represent the Bayes optimal ones in the respective domains.
>
> This lays down a theoretical ground for the model $\rho_x = \phi\left(-|h(x)|\right)$ along with its empirical support. We note that the assumption $\rho_\mathrm{max} < \frac12$ is satisfiable trivially via, e.g., halving the output of the sigmoid or using an exponential PDF with a rate $< 1/2$. We again thank the reviewer for this comment; we amended the manuscript to include this aspect in $\rho_x$'s modeling, in Section 3.2.2.
>
> **W.4.**
> > Experiments are limited to binary classification; multi-class extension is left for future work.
>
> Apart from the naive extensions via, e.g., one-vs-one scheme, we argue that the extension would require nontrivial work. While one might try replacing the loss suffered w.r.t. to the "opposite label", i.e., $-\widetilde y$ in Equation 3 of the proposed loss function with, e.g., (i) an average of losses suffered among all the other labels with $\rho_x$ distributed, e.g., uniformly; or (ii) loss suffered w.r.t. to the "neighboring" label, i.e., the one at the other side of the (closest) decision boundary (which requires an extra decision if the network thinks the current label is not even on either side), they would be merely inspired solutions that lack theoretical grounds. In fact, with the risk equivalence formulation and our label noise model, the resulting linear system of equations is $K \times 2$ where $K$ is the number of classes: the system is simply overdetermined for $K > 2$, hence the nontriviality of the extension.
>
> That being said, we argue that a contribution on the binary classification is still valuable. Prevalent examples from the literature include [1], [11], [13], [14], [15], [16], [17] among others. Notably, the unbiased estimator idea in [1] has been later extended to multiclass classification by [2]; similarly, the symmetric loss idea in [15] has been generalized by [18] to the multiclass setting. Furthermore, we argue that bringing instance-dependency into the risk equivalence framework is nontrivial and offers a different perspective in instance-dependent label noise modeling with a concrete implementation on binary classification; we kindly refer to our answer to the "Weakness 1" above for a detailed argument.

---

> ### Author Response · Authors · 2025-11-20
> **Response to Reviewer nFiM (3/4)**
>
> **W.5.**
> > Evaluation relies on relatively small or subsampled datasets, lacking results on large-scale real-world noise (e.g., WebVision).
>
> Thank you for pointing out the need for an experiment on a naturally noisy dataset. We since performed an experiment on the Clothing1M dataset [9]; an 14-class classification problem with 1 million (noisy) instance-label pairs for training, and \~10,000 (clean) pairs for testing. While the dataset provideres also made cleaning training and validation subsets available, we discard these subsets in our experiments for all models. We take 10 binary subsets from the dataset, use a 6-layer CNN (slightly more complex than the one we used for CIFAR-10 as the resolution of images are greater; the details of the network are in Appendix A.8.) with the same validation and optimization techniques as in CIFAR-10. The test accuracy results are as follows where the header represents the selected binary classes, e.g., "6v8" means "Wind-breaker versus Down Coat", and the scores within 2% of the maximum (relative) are highlighted in bold.
>
> |             | 6v8       | 6v7       | 6v9       | 1v6       | 2v6       | 0v2       | 2v9       | 2v11      | 1v7       | 0v11      |
> |:------------|:----------|:----------|:----------|:----------|:----------|:----------|:----------|:----------|:----------|:----------|
> | Normal      | 62.79     | 74.85     | 69.03     | 73.53     | 67.10     | 65.67     | 68.42     | 71.97     | 72.88     | 71.46     |
> | BCN         | 63.24     | 73.10     | 61.56     | 69.60     | 60.46     | 61.59     | 55.42     | 69.50     | 70.20     | 67.21     |
> | UB          | 65.65     | 70.29     | 64.90     | 65.33     | 58.06     | 63.56     | 57.74     | 57.71     | 71.06     | 62.77     |
> | DMI         | 64.63     | 67.95     | 64.44     | 66.33     | 58.06     | 61.59     | 57.89     | 59.27     | 68.70     | 62.31     |
> | Peer        | 65.21     | 71.11     | 67.39     | 70.94     | 63.69     | 64.87     | 65.39     | 69.91     | 72.88     | 68.52     |
> | APL         | 76.06     | 78.95     | 76.73     | 79.73     | **80.37** | 69.97     | **77.35** | **79.23** | **79.42** | **77.34** |
> | PTD         | 66.35     | 78.95     | 68.40     | 77.39     | 71.89     | 58.82     | 62.60     | 77.49     | 75.35     | 72.24     |
> | BLTM        | 64.57     | 73.80     | 59.14     | 67.50     | 53.27     | 67.06     | 58.13     | 56.55     | 72.78     | 65.97     |
> | GCE         | 70.54     | 75.20     | 71.13     | 71.78     | 72.63     | 67.64     | 72.49     | 72.46     | 75.78     | 75.31     |
> | Coteaching+ | 75.17     | 78.36     | 66.54     | **82.83** | 78.25     | 67.78     | 75.12     | 77.82     | **79.53** | **77.20** |
> | Backward    | 69.59     | **80.70** | **77.43** | **83.08** | **80.83** | 65.96     | 74.80     | 76.83     | **78.56** | **76.88** |
> | PLC         | 66.22     | 74.50     | 64.75     | 70.69     | 61.29     | 65.89     | 67.07     | 75.52     | 73.95     | 68.13     |
> | NDX         | **78.10** | **79.88** | **78.37** | **81.57** | **80.92** | **72.96** | **78.71** | **79.80** | **78.78** | **78.25** |
>
> Results suggest that our model has also competitive performance in a real-world scenario by achieving comparable or mostly better testing accuracy metrics in comparison to the baseline models. (We amended the manuscript with a new section in Section 4 for this real-world dataset, thank you again.)
>
> **W.6.**
> > The drastic accuracy drop below 50% in Figure 1 suggests potential tuning or early-stopping issues in baseline comparisons
>
> We argue that this is not an unexpected behavior for a model with a noise-intolerant loss function, i.e., logistic loss here. It is notoriously known to overfit to even random labels [19, 20]. A clean validation set would certainly help refrain it from such a drop, at least below 50%. However, the validation set itself is noisy in this case. Therefore, under severe noise, there is no guarantee for the noise-intolerant model trained and validated on a noisy data (which is the realistic case) to be at least better than random guess. In fact, a very similar graph is present in [21] (right-most subfigure of Figure 2, page 7) for the multiclass case (10 classes), where the Normal model's test accuracy drops below 10% under severe noise.

---

> ### Author Response · Authors · 2025-11-20
> **Response to Reviewer nFiM (4/4)**
>
> ### Questions
> ---------
> **Q.1.**
> > Can the proposed framework be extended to multi-class classification tasks?
>
> We kindly refer to our response to "Weakness 4" above.
>
> **Q.2.**
> > How sensitive is the method to the duration of the warm-up stage used for estimating ($P(Y \mid x)$?
>
> We kindly refer to our response to "Weakness 2" above.
>
> **Q.3.**
> > How does the estimation of $\eta_x$ influence performance when no label noise is present? Including results for a 0% noise setting (e.g., in Table 1) would clarify this.
>
> We actually present this for CIFAR-10. In Figure 1, the leftmost points of subfigures in Figure 1 correspond to 0% noise rate experiments on CIFAR-10. We observe that our model's performance is not unnecessarily degraded and practically the same as the Normal model's on all four subdatasets.
>
> ### References
> ----------
> - [1] "Learning with Noisy Labels", Natarajan et al.; NIPS, 2013.
> - [2] "Making Deep Neural Networks Robust to Label Noise: a Loss Correction Approach", Patrini et al.; IEEE CVPR, 2017.
> - [3] "Beyond Class-conditional Assumption: A Primary Attempt to Combat Instance-Dependent Label Noise". Chen et al.; AAAI, 2021.
> - [4] "Part-dependent label noise: Towards instance-dependent label noise.", Xia et al.; NIPS, 2020.
> - [5] "Estimating instance-dependent Bayes-label transition matrix using a deep neural network", Yang et al; ICML, 2022.
> - [6] "Dividemix: Learning with noisy labels as semi-supervised learning", Li et al.; ICLR, 2020.
> - [7] "Learning with feature-dependent label noise: A progressive approach", Zhang et al.; ICLR, 2021.
> - [8] "On the Role of Label Noise in the Feature Learning Process", Han et al.; ICML, 2025.
> - [9] "Learning from Massive Noisy Labeled Data for Image Classification", Xiao et al.; IEEE CVPR, 2015.
> - [10] "Learning noisy linear threshold functions", Bylander; Technical Report, 1998.
> - [11] "Modelling class noise with symmetric and asymmetric distributions", Du and Cai; AAAI, 2015.
> - [12] "Learning from Binary Labels with Instance-dependent Noise", Menon et al.; Machine Learning, 2018/09.
> - [13] "A rate of convergence for mixture proportion estimation, with application to learning from noisy labels", Scott; AISTATS, 2015.
> - [14] "Learning with symmetric label noise: the importance of being unhinged", van Rooyen et al.; NIPS, 2015.
> - [15] "Making risk minimization tolerant to label noise", Ghosh et al.; Neurocomputing, 2015:160.
> - [16] "Classification with noisy labels by importance reweighting", Liu and Tao; IEEE TPAMI, 2016.
> - [17] "eer loss functions: learning from noisy labels without knowing noise rates", Liu and Guo; ICML, 2020.
> - [18] "Normalized loss functions for deep learning with noisy labels", Ma et al.; ICML, 2020.
> - [19] "A closer look at memorization in deep networks", Arpit et al.; ICML, 2017.
> - [20] "Generalized cross entropy loss for training deep neural networks with noisy labels", Zhang and Sabuncu; NIPS, 2018.
> - [21] "L_dmi: a novel information-theoretic loss function for training deep nets robust to label noise", Xu et al.; NIPS, 2019.

---

### Official Review · Reviewer_5k6V · 2025-11-03

**Soundness:** 2
**Presentation:** 3
**Contribution:** 3
**Rating:** 4
**Confidence:** 2

**Summary:**

This paper tackles label noise in binary classification by reviving the instance-only dependent (IDN) noise model, where label corruption depends solely on the input instance. The authors derive a theoretically consistent loss correction scheme ensuring risk equivalence between noisy and clean risks, and stabilize it via a regularized variant with generalization guarantees. The instance-level noise rate ρₓ is efficiently estimated from model confidence (|h(x)|), enabling online learning. Experiments on image, audio, and tabular datasets show strong robustness under moderate-to-high noise.

**Strengths:**

The paper offers a solid theoretical foundation: Proposition 1 rigorously establishes risk equivalence, while Lemma 1 and Proposition 2 ensure stability and generalization.

Experiments across image, audio, and tabular domains with diverse models (CNN, MLP, LightGBM) confirm robustness and broad applicability.

**Weaknesses:**

Justification for using P(Y|X)=P(Y'|X) : This is a critical step in the method that needs justification. Without proper justification, it seems that there is a significant gap between the analysis and the actual proposed method.

**Questions:**

Have you tried to training two model approach, one for P(Y|X) and one for rho_x ?

---

> ### Author Response · Authors · 2025-11-20
> **Response to Reviewer 5k6V (1/3)**
>
> We thank you for your time and valuable feedback! Below we present our answers to the weakness point and the question.
>
> ### Weaknesses
> ----------
> **W.1.**
> > Justification for using $P(Y \mid X)=P(Y' \mid X)$ : This is a critical step in the method that needs justification. Without proper justification, it seems that there is a significant gap between the analysis and the actual proposed method.
>
> We first note that $P(Y \mid X)$'s estimation in statistically consistent (i.e., training as if with the clean labels as the sample size grows) or probabilistic models in general requires design choices. We exemplify from the literature as follows.
>   - "Learning from Massive Noisy Labeled Data for Image Classification" [1]: they concurrently fit two models to model $Y$ and $Z$ (the label noise *kind* latent variable) and maximize the likelihood of $\widetilde Y$ with Expectation-Maximization. That is, they exploit the factorization (of $\widetilde Y$ over $Y$) the class-only dependent label noise model allows, and maximize the (incomplete) likelihood. However, the resulting framework has 3 models in it (one for $Y$, one for $Z$ and one for the transition probabilities) and the optimization via EM gets complex (e.g., requires careful initialization) and requires an identified set of clean labels.
>   - "Learning with Noisy Labels" [2]: when they develop their "Method of Unbiased Estimators", they require $\mathbb{E}[\ell(\widetilde Y, f(X))] = \ell(Y, f(X))$ for all $Y$, $f(\cdot)$ values. Please note that there is no expectation on the right hand side. Therefore $P(Y \mid X)$ is nonexistent in their formulation, making it not achieve risk equivalence (we also point to this fact in the manuscript (end of related work)).
>   - "Making Deep Neural Networks Robust to Label Noise: a Loss Correction Approach" [3]: This is a multi-class generalization of the above work of [2]; this time we see the expectation on the right hand side. Building on the factorization $P(Y \mid X) = P(Y \mid \widetilde{Y},X)P(\widetilde{Y} \mid X)$ and their assumption of y-only dependent label noise, they aim to estimate $P(Y \mid \widetilde{Y})$, i.e., a $K\times K$ transition matrix ($K$: number of classes). Their (one) design choice is to train a separate neural network to model this matrix by assuming the existence of perfect samples (i.e., those having almost surely clean labels). Therefore, they first learn a transition matrix, freeze it and then use it in the second phase of learning a new machine on $\widetilde{Y}$s to uncover $P(Y \mid X)$. This modeling of the transition matrix turned out to perform really well, as Forward & Backward loss correction methods from this paper is still of high relevance in label noise research (theory- and performance-wise).
>
> Our instance-dependent label noise model, while more natural/powerful [4], does not lend itself to the factorization above because in $\rho_x = P(\widetilde{Y} \ne Y \mid X)$, $Y$ and $\widetilde{Y}$ are on the left side of the condition together. Nevertheless, we could still learn the "annotator's brain" in a separate machine, i.e., model $P(\widetilde{Y} \mid X)$, then freeze its in-sample predictions, and then in a second phase, while training an $h$ with $\tilde\ell$ (on the same dataset still), use the frozen predictions of the former machine as a proxy for $P(\widetilde{Y} \mid X)$, and the current output of $h$ as a proxy for $P(Y \mid X)$, as with $\tilde\ell$, this phase's machine $h$ is expected to uncover $P(Y \mid X)$ by Proposition 1. Actually, we tried this. In Section 3.2.1, we mention this way of modeling quantities, and in Appendix A.4, we present a detailed empirical comparison of this disjoint approach with what we instead do -- the difference in performance (and naturally also the computation time) was significantly worse in this disjoint way of modeling those probabilities, providing an empirical evidence for our design choice.

---

> ### Author Response · Authors · 2025-11-20
> **Response to Reviewer 5k6V (2/3)**
>
> While that empirical evidence suggests one can do better than disjoint modeling, we directly justify our assumption as follows. As noted above, in Patrini et al.'s loss correction design [3], they assume the existence of perfectly clean examples, on/with which they first train a network to model $p(Y \mid \widetilde{Y})$ (and similarly, also in [1]). In our setup (or in any noisy label learning setup for that matter), while the *identification* of the clean labels is not assumed, their existence is, e.g., our $\mathbb{E}[\rho_x] < 0.5$ assumption. In fact, in a $K$-class scenario, one needs at least $\dfrac1K$ clean samples to exist so that learning is even possible [5]. (Actually, training is possible either way, e.g., as an extreme case of 100% label noise in binary classification, a (good enough) model's predictions on the unseen data will always be worse than random *until* they are flipped, at which point the semantic labels are matched. So the assumption of high signal-to-noise ratio is to preserve consistency of the semantic meaning of the labels between training and testing sets.) What's more, it has been demonstrated by [6] that the neural network first "focuses" on these clean samples in the early stages of the training. But this means we can elevate the disjoint modeling idea of [3] of $P(Y \mid X)$ and $P(\widetilde{Y} \mid X)$ with the dominance of clean samples in early stages of training to make the modeling joint: a number of warm-up epochs at start with the normal (uncorrected) loss. Once the "groundwork" of establishing a decision boundary by the machine in the warm-up period is done with the usual loss function through the dominant clean samples, our loss correction mechanism kicks in to make the machine more aware of the pitfalls due to label noise (this "awareness" is examined mathematically on several edge cases in Section 3.2.3, where we present similarities and differences between $\tilde\ell$ and $\ell$).
>
> ### Questions
> ---------
> **Q.1.**
> > Have you tried to training two model approach, one for $P(Y \mid X)$ and one for $\rho_x$ ?
>
> We interpret the question in two ways and answer them both as follows.
>
> 1. Training a model for the clean label posterior $P(Y \mid X)$ and another one for $\rho_x$ disjointly/simultaneously
>
> In any label noise model, learning a machine (from possibly noisy labels) that effectively models the clean label posterior is the ultimate goal. Therefore, while modeling $P(Y \mid X)$ cannot directly be a part of a given label noise learning algorithm, it is possible to model it via, e.g., the (incomplete) likelihood of the noisy labels where it is demarginalized over the clean label probability as, e.g., $P(\widetilde Y \mid X) = P(\widetilde Y \mid Y,X) P(Y \mid X)$. A canonical example of such a model that uses the Expectation-Maximization to maximize the said likelihood is of Xiao et al. [1], where the authors treat the clean label (as well as the noise kind of an instance, named $Z$) as a hidden variable. This involves training two networks for $Y$ and $Z$, and learning the transition matrix (on a small dataset that assumes identified clean and noisy labels), all based on likelihood maximization. Apart from modeling the noise as class-only dependent, major limitations of this approach are twofold: (i) optimization gets complex as they learn the parameters of the said networks using EM (in fact, the algorithm is sensitive to initialization and to prevent the optimization from reaching "totally wrong posterior computations", they pretrain the networks on clean data first); (ii) they assume the existence and identifiability of the clean labels in the data (to pretrain the latent-variable modeling networks as well as the transition matrix). Our instance-only dependent label noise model, however, cannot be integrated into this framework because in $\rho_x = P(\widetilde Y \ne Y \mid X)$, $Y$ and $\widetilde Y$ are on the left side of the condition together, i.e., the clean label posterior cannot be factored as above using $\rho_x$. In other words, even one manages a simultaneous learning of $\widetilde Y$ and $Y$, our noise model does not lend itself to a mathematical connection between these objects.
>
> 2. Training a model for the noisy label posterior $P(\widetilde Y \mid X)$ and another one for $P(Y \mid X)$ and $\rho_x$ disjointly/simultaneously
>
> In this two-model scenario, our approach can be employed. In fact, we did try this setting on the Speakers dataset and the results are in Appendix A.4. We observed that joint training is significantly and consistently better-performing while, as expected, being much more time-efficient. We justify this positive result in our answer above to your weakness point W.1.

---

> ### Author Response · Authors · 2025-11-20
> **Response to Reviewer 5k6V (3/3)**
>
> ### References
> ----------
> - [1] "Learning from Massive Noisy Labeled Data for Image Classification", Xiao et al.; IEEE CVPR, 2015.
> - [2] "Learning with Noisy Labels", Natarajan et al.; NIPS, 2013.
> - [3] "Making Deep Neural Networks Robust to Label Noise: a Loss Correction Approach", Patrini et al.; IEEE CVPR, 2017.
> - [4] "Beyond Class-conditional Assumption: A Primary Attempt to Combat Instance-Dependent Label Noise". Chen et al.; AAAI, 2021.
> - [5] "Learning from Binary Labels with Instance-dependent Noise", Menon et al.; Machine Learning, 2018/09.
> - [6] "On the Role of Label Noise in the Feature Learning Process", Han et al.; ICML, 2025.

---

### Author Response · Authors · 2025-11-20
**Summary of the Revision upon Feedback**

Dear Reviewers,

We thank you all for your valuable time and feedback! We have responded to each and every weakness point raised and question asked to our best ability. We have revised our manuscript accordingly, mentioned the relevant sections prompted by your reviews in individual responses, and uploaded a highlighted PDF.

Here is the summary of the major changes:

- We experimented with a real-world dataset that is naturally noisy, Clothing1M [1]. A new section in Experiments is spared for this (Section 4.4.).
- We have added a theoretical support for our distance-based modeling of the transition probabilities [2]. We were not aware of this result while writing the manuscript (though, we were already citing the source it comes from! [2]), and prompted by your comments, we compiled it while rechecking the said literature [2]. This is now in the amended Section 3.2.2.
- We detailed our argument on modeling $\widetilde Y$ and $Y$ together. This resulted in a new section, Appendix A.4.1.
- Relatedly, we added support for our warm-up strategy; we experimented with one of our datasets to see the need for warm-up as well as insensitivity against it. This resulted in a new section, Appendix A.5.
- For space constraints, we moved our Sanity Check experiment to be Appendix A.2 to make space for the experiment with Clothing1M.

The minor changes include adding the details of the new dataset Clothing1M [1]; adding the details of the CNN we used for its experiments; adding the test set distribution of the subsets of this dataset; and renaming the subsections of Section 4 (Experiments) for a better exposition.

We thank you again and we are looking forward to hearing from you!

Best Regards,
&nbsp;

Authors


### References
----------
[1] "Learning from Massive Noisy Labeled Data for Image Classification", Xiao et al.; IEEE CVPR, 2015.

[2] "Learning from Binary Labels with Instance-dependent Noise", Menon et al.; Machine Learning, 2018/09.

---

### Author Response · Authors · 2025-11-30
**Summary of our responses to the Reviewers**

Dear Area Chair,

We would like to summarize the major concerns the Reviewers raised and our responses to them.

> Experiment with a real-world, i.e., a naturally noisy dataset

We have since performed an experiment with Clothing1M [1], and spared a new section for it in the revised manuscript (Section 4.4.). The results were similarly positive as for the other dataset settings, showing practical robustness with a naturally noisy large dataset.

> Support for the way we are modeling $P(Y \mid X)$

We argued that in any probabilistic model on label noise, one needs to make a design choice for modeling $P(Y\mid X)$. We exemplified this through 3 prominent models of the literature: in [1], a separate network was trained to maximize the incomplete likelihood of $\widetilde Y$; in [2], it is treated deterministically; in [3], a two-stage (disjoint) training is employed together with the assumption of anchor points. What we do is spare a few "warm-up" epochs in the beginning of training without loss correction, which is essentially the _joint_ version of [3]. We argued that
  - warm-up epochs are a common strategy [7], [8], [9] and even theoretically shown to be necessary [9] (albeit for their specific noise model) and
  - during the warm-up, dominance of clean labels was recently proven by [6].

Furthermore, we performed an experiment on the effect of warm-up epochs by sweeping it from 0 to full number of epochs. When plotted it against test accuracy over two noise rates, we roughly observed a reverse U-shape with a rather flat maxima, which is the desired behaviour: there is a _need_ for warmup (left, increasing part of $\cap$); there is _insensitivity_ against it in the middle region (rather flat maxima of $\cap$); and it drops down to the Normal method's performance (right, decreasing part of $\cap$). These supportive arguments resulted in two new sections (Appendix A.4.1 and Appendix A.5) in the revised manuscript.

> Support for the distance-based approximation of $\rho_x$, the transition probability

We argued that our approximation has merits in three aspects:
  - it is intuitive: the closer an instance to the decision boundary, the more error prone it must have been to mislabeling as it contains features of more than one class;
  - it has empirical support: among a wide spectrum of experiments (image, audio, tabular; synthetic, real-life; moderate and high noise; NNs and GBMs), it shows comparable or mostly better generalization when compared to 12 methods from the literature;
  - it has theoretical support: due to Menon et al. [5], our noise model is AUROC-consistent (in achieveing true AUROC with clean labels from noisy labels). We amended Section 3.2.2. with this argument.

> Limited to binary classification

While we acknowledge this limitation (in the manuscript also), we humbly argue that there is still theoretical and practical merit in our work. We have resurfaced the underutilized instance-only dependent label noise model in a risk-equivalence framework through loss correction with theoretical support; provided a unifying & flexible probability model; and showed strong empirical support against 12 models of the literature. To the best of our knowledge, there were previusly only two works on this: [12] from 1998 and [13] from 2015, both of which focused on linear machines with a specific choice for $\rho_x$. We believe that our work brings a new perspective to the instance-dependent label noise problem where the entirety of the recent works (say, since 2019) uses ILDN, i.e., instance _and_ label dependent noise.

Furthermore, there are many prevalent examples from the label noise literature addressing binary classification, e.g., [2], [13], [14], [15], [16], [17], [18] among others. Notably, the unbiased estimator idea in [2] has been later extended to multiclass classification by [3]; similarly, the symmetric loss idea in [16] has been generalized by [19] to the multiclass setting. It is in our future agenda to extend to the multiclass setting, which we argue is a nontrivial work, e.g., in our risk equivalence formulation, the linear system is $K \times 2$ in general for $K$ classes, which is simply overdetermined for $K > 2$ (which is not to mean this is the only way to use instance-only dependent noise but to emphasize the nontriviality of the extension).

----------------------------------------

Overall, we would like to reiterate that we have responded to each and every weakness point raised and question asked by the Reviewers to our best ability. We have revised our manuscript accordingly, mentioned the relevant sections prompted by the Reviewers in individual responses, and uploaded a highlighted PDF.

Thank you for your consideration and valuable time, and we would be happy to answer any questions you might have.

Best Regards,

Authors

(For the references here, please refer to our [comment](https://openreview.net/forum?id=tuvkrivvbG&noteId=6wwVJtPahz) at the very end of this page.)

---

### Meta-Review · Area_Chair_rY88 · 2025-12-26

**Summary:**

This paper tackles label noise in binary classification by reviving the instance-only dependent (IDN) noise model, where label corruption depends solely on the input instance.


The scores received for this submission 4466.  There are various concerns from the reviewers but from my perspective, they are addressed in detail from the authors' response and final paper summization message. In addition, one reviewer voted for 4 has a very low confidence level (2).

based on the overall recommendation and the authors' response, I would recommend its acceptance. I would strongly recommend the authors to include all the constructive suggestions from the reviewers.

**Reviewer Concerns:**

I belive there are no ourstanding issues as far as I can read from the very detailed response from the authors.

**Reviewer Scores:**

NA

---

### Decision · Program_Chairs · 2026-01-26

Accept (Poster)